# Matchings Under Biased and Correlated Evaluations

**Amit Kumar**
IIT Delhi

**Nisheeth K. Vishnoi**
Yale University

## Abstract

We study a two-institution stable matching model in which candidates from two distinct groups are evaluated using partially correlated signals that are group-biased. This extends prior work (which assumes institutions evaluate candidates in an identical manner) to a more realistic setting in which institutions rely on overlapping, but independently processed, criteria. These evaluations could consist of a variety of informative tools such as standardized tests, shared recommendation systems, or AI-based assessments with local noise. Two key parameters govern evaluations: the bias parameter $\beta \in (0, 1]$, which models systematic disadvantage faced by one group, and the correlation parameter $\gamma \in [0, 1]$, which captures the alignment between institutional rankings. We study the representation ratio $\mathcal{R}(\beta, \gamma)$, i.e., the ratio of disadvantaged to advantaged candidates selected by the matching process in this setting. Focusing on a regime in which all candidates prefer the same institution, we characterize the large-market equilibrium and derive a closed-form expression for the resulting representation ratio. Prior work shows that when $\gamma = 1$, this ratio scales linearly with $\beta$. In contrast, we show that $\mathcal{R}(\beta, \gamma)$ increases nonlinearly with $\gamma$ and even modest losses in correlation can cause sharp drops in the representation ratio. Our analysis identifies critical $\gamma$-thresholds where institutional selection behavior undergoes discrete transitions, and reveals structural conditions under which evaluator alignment or bias mitigation are most effective. Finally, we show how this framework and results enable interventions for fairness-aware design in decentralized selection systems.

## 1 Introduction

Stable matching mechanisms are a cornerstone of allocation theory, extensively studied in game theory, economics, and, more recently, machine learning [48, 49, 50, 8, 21, 41, 32, 33, 39]. These mechanisms assign agents (e.g., students, workers, users) to institutions (e.g., schools, employers, content slots) under preference and capacity constraints, ensuring that no unmatched agent-institution pair would mutually prefer each other over their assigned match. Such matching systems are widely deployed in admissions practices, labor markets, and digital platforms.

Often, preferences across candidates are determined using evaluations such as standardized tests, interviews, or aggregate reviews. However, such evaluations can exhibit *group-dependent bias*—systematically disadvantaging candidates from certain demographic groups [47, 28, 46, 22, 40, 55, 36]. Further, institutions often rely on overlapping signals (e.g., standardized test scores or resumes), yielding *inter-institutional correlation* in how candidates are assessed. These structural properties of bias and correlation can jointly skew allocation outcomes, compounding access gaps across education, employment, and economic opportunity [31, 30, 56, 26, 10].

The rise of AI-driven evaluation tools has further amplified these concerns. Predictive scoring models, automated interviews, and recommender systems are now widely used to rank candidates [43]. While scalable and efficient, these systems often inherit biases from historical data and reinforce alignment through shared features or pretraining [44, 54]. As a result, modern evaluations may not only be biased but also correlated across institutions in opaque or unintended ways.

39th Conference on Neural Information Processing Systems (NeurIPS 2025).

A growing body of work has studied the impact of biased evaluations on allocation. For example, [15] analyzes a centralized matching model where a common evaluation multiplicatively downweights scores for one group. They show that representation degrades linearly with bias and propose fairness-preserving mechanisms. In contrast, many real-world systems are decentralized: institutions make decisions independently, often using correlated but locally processed signals. For instance, school choice programs use structured tie-breaking rules that induce inter-institutional correlation [1, 4]. Even in the absence of bias, such correlation can generate disparities across groups [12]. These insights motivate our central question: *How do group-dependent bias and inter-institutional correlation jointly shape group-level representation in decentralized matching markets?*

**Our contributions.** We build on and generalize the centralized model of [15] by introducing a decentralized stable matching framework that jointly models group-dependent bias and inter-institutional correlation. We consider a two-institution setting where candidates belong to one of two groups: $G_1$ (*advantaged*) and $G_2$ (*disadvantaged*). A total fraction $c$ of candidates must be matched across both institutions. Each candidate possesses two latent attributes. Institution 1 ranks candidates solely by the first attribute, while Institution 2 uses a weighted combination of both attributes. The *correlation parameter* $\gamma \in [0, 1]$ controls the degree of alignment between the evaluations of the two institutions. As in [15], candidates from $G_2$ face a multiplicative bias $\beta \in (0, 1]$ applied to both attributes. For our main results, we assume that all candidates prefer Institution 1, reflecting prestige-driven preferences in many real-world settings (e.g., centralized college admissions [2, 42, 35, 53]). We extend our analysis to general preference distributions in Section L. Our main contributions are:

1. **Equilibrium characterization.** We first show that, in the infinite population limit, the stable matching is governed by two thresholds, $s_1^\star$ and $s_2^\star$, which depend on $\beta$, $\gamma$, and $c$. The threshold $s_1^\star$ admits a straightforward expression; see Equation (4). In contrast, deriving $s_2^\star$ requires resolving interdependence between institutions' rankings and cross-group orderings. We introduce a regime-reduction technique that collapses sixteen potential case distinctions into three interpretable $\gamma$-based regions, determined by analytically derived thresholds $\gamma_1, \gamma_2, \gamma_3$ (Theorem 3.1). Using this structure, we obtain a closed-form expression for $s_2^\star$ (Theorem 3.2) and prove that it varies *unimodally* with respect to $\gamma$ (Theorem E.1), capturing a subtle trade-off: increasing correlation can both enhance evaluator alignment and reduce access to high-scoring candidates due to intensified competition with Institution 1.

2. **Fairness metric and monotonicity.** Building on the equilibrium thresholds, we derive a piecewise closed-form expression for the representation ratio $\mathcal{R}(\beta, \gamma)$, which measures the relative selection rates between the two groups (Theorem 3.3). We introduce a normalized variant, $\mathcal{N}(\beta, \gamma)$, which quantifies fairness relative to the ideal setting with full evaluator alignment. Unlike $\mathcal{R}$, previously studied in the centralized model of [15], the normalized ratio isolates the effect of evaluator misalignment and enables scale-free comparisons across regimes. Because both $\mathcal{R}$ and $\mathcal{N}$ depend on regime-specific thresholds, direct algebraic analysis is intractable; instead, we establish their monotonicity in $\beta$ and $\gamma$ by examining the equilibrium-defining equations themselves.

3. **Studying fairness interventions.** Leveraging the structure of $\mathcal{N}(\beta, \gamma)$, we identify the Pareto frontier of minimal interventions that achieve a desired fairness level $\tau$. Specifically, we characterize the combinations of evaluator bias $\beta$ and alignment $\gamma$ that satisfy $\mathcal{N}(\beta, \gamma) \geq \tau$, and visualize these trade-offs via contour plots. This enables system designers—given current evaluation parameters $(\beta_0, \gamma_0)$ and a target $\tau$—to determine whether fairness can be achieved by adjusting either parameter and to select cost-effective strategies accordingly; see Section 4.

Together, these results provide a structural and quantitative foundation for fairness-aware design in decentralized selection systems with biased and partially aligned evaluations.

**Related work.** Stable matching has been extensively studied in game theory, economics, and machine learning, with foundational contributions on stability, truthfulness, and decentralized evaluation [24, 48, 49, 50, 21, 8, 32, 33, 39]. Our work builds on [15], who studied fairness in centralized matching under group-biased evaluations. We generalize their setting to allow partially correlated, institution-specific evaluations in decentralized markets. This aligns with recent work highlighting the role of inter-institutional correlation in shaping outcomes [12, 4, 1]. While prior models focus on either bias or correlation, we analyze their joint effect on representation. We also connect to literature on algorithmic fairness in subset selection, ranking, and admissions [29, 16, 25], and to empirical studies showing persistent disparities in algorithmic selection systems [26, 31, 30]. A full discussion of related work is deferred to Appendix A.

## 2 Model

We consider a two-institution matching setting where Institution $i$ has capacity $c_i n$ for $i \in \{1, 2\}$. Candidates belong to an *advantaged group* $G_1$ or a *disadvantaged group* $G_2$, with group sizes $|G_i| = \nu_i n$. We focus on the demand-exceeding-supply regime, where $c_1 + c_2 < \nu_1 + \nu_2$.

**Correlated evaluations.** Each candidate $i$ has two independent attributes, $v_{i1}$ and $v_{i2}$, drawn i.i.d. from the uniform distribution on $[0, 1]$. Institution 1 evaluates candidates using $v_{i1}$, while Institution 2 uses a convex combination:

$$u_{i1} = v_{i1} \qquad \text{and} \qquad u_{i2} = \gamma v_{i1} + (1 - \gamma) v_{i2},$$

where $\gamma \in [0, 1]$ controls the degree of correlation between the two institutions' evaluations. This model captures, for instance, settings where Institution 2 relies partly on a common criterion and partly on institution-specific judgments.

**Bias in evaluations.** We model bias using a parameter $\beta \in (0, 1]$, following the framework of [29, 15]. The estimated utility $\hat{u}_{i\ell}$ of a candidate from group $G_2$ for Institution $\ell$ is downscaled relative to their true utility:

$$\hat{u}_{i\ell} = u_{i\ell} \text{ if } i \in G_1 \quad \text{and} \quad u_{i\ell} = \beta u_{i\ell} \text{ if } i \in G_2.$$

This captures settings where disadvantaged candidates are systematically undervalued in evaluations, for example, due to differences in interview performance or standardized tests. Our analysis focuses on $\beta \leq 1$, though it extends naturally to $\beta > 1$ to model overestimation, to asymmetric bias across institutions; see Appendix I), or to additive bias (Appendix J). Both institutions evaluate and select candidates from *both* groups; the parameter $\beta$ only affects how candidates from $G_2$ are evaluated.

**Stable matching.** A matching $M$ assigns each candidate to an institution, subject to institutional capacity constraints. The matching $M$ is *stable* if no candidate $i$ and institution $\ell$ form a blocking pair: that is, if $i$ prefers $\ell$ to their assigned institution $M(i)$, and there exists a candidate $j$ currently matched to $\ell$ such that $\hat{u}_{i\ell} > \hat{u}_{j\ell}$, then $i$ must already be assigned to $\ell$. Stable matchings can be computed via the Deferred Acceptance algorithm [24, 50]. We consider a model in which each candidate prefers Institution 1 over Institution 2 with probability $p$. The main body focuses on the case $p = 1$, where all candidates prefer Institution 1. Extensions to general $p$ are discussed in Appendix L.

**Scaling limit.** We analyze the model in the large-market regime, where the number of candidates $n \to \infty$. This is a standard approach in stable matching theory for capturing micro-to-macro behavior. In the finite setting, it is known that stable matchings are characterized by threshold-based rules; see Proposition B.1. Specifically, for any realization of the evaluation scores $\hat{u}$, the unique stable assignment is determined by two cutoffs $S_1$ and $S_2$ such that $M^{-1}(1) = \{i : \hat{u}_{i1} \geq S_1\}$, $M^{-1}(2) = \{i : \hat{u}_{i1} < S_1 \text{ and } \hat{u}_{i2} \geq S_2\}$. Taking expectations, normalizing by $n$, and letting $n \to \infty$, we obtain the *scaling limit*, where the stable matching is described by deterministic thresholds $s_1$ and $s_2$ satisfying:

$$\nu_1 \Pr[\hat{u}_{i1} \geq s_1] + \nu_2 \Pr[\hat{u}_{i'1} \geq s_1] = c_1, \tag{1}$$

$$\nu_1 \Pr[\hat{u}_{i1} < s_1, \hat{u}_{i2} \geq s_2] + \nu_2 \Pr[\hat{u}_{i'1} < s_1, \hat{u}_{i'2} \geq s_2] = c_2. \tag{2}$$

Here, $i$ denotes a candidate from group $G_1$ and $i'$ from group $G_2$. These equations admit a *unique* solution $(s_1^\star, s_2^\star)$; see Appendix B for formal details. We show in Appendix B.3 that the finite thresholds $(S_1, S_2)$ converge to their deterministic limits $(s_1^\star, s_2^\star)$ as $n \to \infty$.

**Metrics.** To quantify fairness in selection outcomes, we use the *representation ratio* introduced by [15]. Let $\rho_j$ denote the fraction of selected candidates from group $G_j$, and define: $\mathcal{R}(M) := \frac{\min(\rho_1, \rho_2)}{\max(\rho_1, \rho_2)}$. This metric captures disparity in group-level representation, with $\mathcal{R}(M) = 1$ indicating perfect proportionality across groups. In the scaling limit, the representation ratio can be expressed in closed form in terms of the stable matching thresholds $(s_1^\star, s_2^\star)$ derived from equations (1)–(2):

$$\mathcal{R}(\beta, \gamma) = \frac{\Pr[\hat{u}_{i'1} \geq s_1^\star] + \Pr[\hat{u}_{i'1} < s_1^\star, \hat{u}_{i'2} \geq s_2^\star]}{\Pr[\hat{u}_{i1} \geq s_1^\star] + \Pr[\hat{u}_{i1} < s_1^\star, \hat{u}_{i2} \geq s_2^\star]}. \tag{3}$$

While $\mathcal{R}(\beta, \gamma)$ is useful for quantifying raw disparity, it is not always scale-free: for fixed $\beta$, it may attain a maximum below 1 depending on the alignment parameter $\gamma$. This complicates comparisons across systems with different baseline levels of bias. To address this, we introduce the *normalized representation ratio* $\mathcal{N}(\beta, \gamma) := \frac{\mathcal{R}(\beta, \gamma)}{\max_{\gamma' \in [0,1]} \mathcal{R}(\beta, \gamma')}$, which measures the fraction of ideal representation achieved at the current alignment level. Unlike $\mathcal{R}$, this normalized metric allows us to isolate the impact of misalignment independently of the underlying bias and supports principled intervention analysis.

# 3 Main results

We characterize the *representation ratio* $\mathcal{R}(\beta, \gamma)$ (and thus $\mathcal{N}(\beta, \gamma)$), which quantifies how equitably candidates from advantaged and disadvantaged groups are selected under a stable matching. Under the scaling limit introduced in Section 2, the matching is determined by two thresholds $s_1^\star$ and $s_2^\star$, which are solutions to Equations (1) and (2). For clarity, we focus on the symmetric case $c_1 = c_2 = c$ and $\nu_1 = \nu_2 = 1$, providing a complete piecewise characterization of $s_1^\star$, $s_2^\star$, and the resulting $\mathcal{R}(\beta, \gamma)$ as functions of the bias parameter $\beta$ and the correlation parameter $\gamma$. We focus on the demand-exceeding-supply regime $c < 1$, where the thresholds determine partial selection. Our analysis centers on the regime $\beta \geq 1 - c$, where both groups are eligible for selection by Institution 1—capturing cases where bias is moderate and fairness is nontrivial. Extensions to $\beta < 1 - c$ and to more general preference models are deferred to Appendix K and Appendix L respectively.

**Step 1: Computing the threshold $s_1^\star$.** We begin by solving equation (1), which determines the threshold $s_1^\star$ used by Institution 1. Since Institution 1 evaluates candidates using their first attribute $v_{i1}$, and a bias factor of $\beta$ is applied to candidates from group $G_2$, assuming $\nu_1 = \nu_2 = 1$ and $c_1 = c_2 = c$, the left-hand side of (1) becomes the average admission probability across both groups: $\frac{1}{2}\Pr[\hat{u}_{i1} \geq s_1] + \frac{1}{2}\Pr[\hat{u}_{i'1} \geq s_1] = \frac{1}{2}(1 - s_1) + \frac{1}{2}\left(1 - \frac{s_1}{\beta}\right)$. Setting this to $c$ yields:

$$s_1^\star = \frac{2-c}{1+1/\beta}. \tag{4}$$

This expression shows that $s_1^\star$ increases with $\beta$: as bias decreases and evaluations of group $G_2$ improve, the institution can raise its threshold while maintaining its target capacity. For this threshold to admit candidates from both groups, it must satisfy $s_1^\star \leq \beta$, ensuring that the effective threshold for group $G_2$ does not exceed their maximum possible score. Moreover, when $\beta \geq 1 - c$, one can show that $s_2^\star \leq s_1^\star$ (Proposition C.2). This yields the condition $\beta \geq 1 - c$, which defines our main regime of interest. When $\beta < 1 - c$, Institution 1 selects only from group $G_1$, resulting in a degenerate matching; we defer this case to Appendix K. Finally, note that $s_1^\star$ depends only on $\beta$ and $c$, and is independent of $\gamma$, which influences only the threshold $s_2^\star$ used by Institution 2.

**Step 2: Computing the threshold $s_2^\star$.** The threshold $s_2^\star$ satisfies the second matching equation (2), which captures the fraction of candidates not admitted by Institution 1 but accepted by Institution 2. That is, $s_2^\star$ defines the acceptance region for Institution 2 over candidates rejected by Institution 1, based on a potentially different evaluation criterion. As a warm-up, consider the case $\gamma = 1$, where both institutions evaluate candidates using the same biased attribute $v_{i1}$. This corresponds to a centralized setting previously studied in [15]. In this case, Institution 2 selects candidates whose (biased) utility falls in the interval $[s_2^\star, s_1^\star]$, so the selection probabilities for each group are: $\Pr[\hat{u}_{i1} \in [s_2, s_1]] = s_1 - s_2$, $\Pr[\hat{u}_{i'1} \in [s_2, s_1]] = \frac{s_1 - s_2}{\beta}$. Using equation (2), the capacity constraint becomes: $\frac{1}{2}(s_1 - s_2) + \frac{1}{2}\left(\frac{s_1 - s_2}{\beta}\right) = c$. Solving yields the closed-form expression: $s_2^\star = s_1^\star - \frac{c}{1+1/\beta}$. This expression serves as a benchmark and reveals a useful structure: when both institutions are aligned in evaluation, $s_2^\star$ is simply a shifted version of $s_1^\star$.

In the general case $\gamma \in (0, 1)$, Institution 2 evaluates a convex combination of the two attributes: $u_{i2} = \gamma v_{i1} + (1 - \gamma)v_{i2}$. A candidate is selected if they are rejected by Institution 1 ($\hat{u}_{i1} < s_1$) and accepted by Institution 2 ($\hat{u}_{i2} \geq s_2$). The key term in (2) is the joint probability $\Pr[\hat{u}_{i2} \geq s_2 \wedge \hat{u}_{i1} < s_1]$, which corresponds to the area above the line $L(\gamma, s_2) := \gamma x + (1 - \gamma)y = s_2$ inside the rectangle $[0, s_1] \times [0, 1]$. Depending on how this line intersects the rectangle, we distinguish four cases:

*Case I:* $s_2 \leq \min(s_1\gamma, 1 - \gamma)$. The line intersects the $y$-axis at $\frac{s_2}{1-\gamma} \leq 1$ and the $x$-axis at $\frac{s_2}{\gamma} \leq s_1$, yielding $\Pr[\hat{u}_{i1} < s_1, \hat{u}_{i2} \geq s_2] = s_1 - \frac{s_2^2}{2\gamma(1-\gamma)}$.

*Case II:* $s_2 \geq \max(s_1\gamma, 1 - \gamma)$. The line intersects the $y$-axis at $\frac{s_2}{1-\gamma} > 1$ and the $x$-axis at $\frac{s_2}{\gamma} > s_1$, so $\Pr[\hat{u}_{i1} < s_1, \hat{u}_{i2} \geq s_2] = \frac{(1-\gamma-s_2+\gamma s_1)^2}{2\gamma(1-\gamma)}$.

*Case III:* $s_1\gamma < s_2 < 1 - \gamma$. The line intersects the $y$-axis at $\frac{s_2}{1-\gamma} \leq 1$ and the line $x = s_1$ at $\frac{s_2-\gamma s_1}{1-\gamma}$, leading to $\Pr[\hat{u}_{i1} < s_1, \hat{u}_{i2} \geq s_2] = \frac{s_1}{2}\left(2 - \frac{2s_2-\gamma s_1}{1-\gamma}\right)$.

*Case IV:* $1 - \gamma < s_2 < s_1\gamma$. The line intersects the $y$-axis at $\frac{s_2}{1-\gamma} > 1$ and the $x$-axis at $\frac{s_2}{\gamma} \leq s_1$, giving $\Pr[\hat{u}_{i1} < s_1, \hat{u}_{i2} \geq s_2] = s_1 - \frac{s_2}{\gamma} + \frac{1-\gamma}{2\gamma}$.

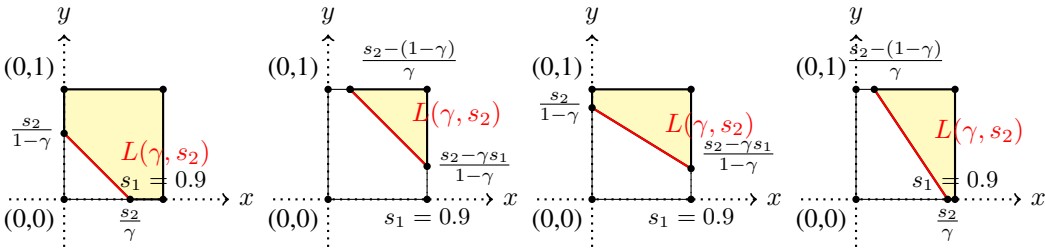

Figure 1: (Left to Right) Cases I, II, III, IV that arise in the probability computation.

We now repeat the analysis for group $G_2$, whose evaluations are rescaled by $\beta$. That is, the decision boundary becomes $L_\beta(\gamma, s_2) := \gamma x + (1-\gamma)y = s_2/\beta$, and the relevant domain is $[0, s_1/\beta] \times [0, 1]$. The same four cases apply, denoted I'–IV', depending on how the line intersects this rectangle. A brute-force approach would guess the correct pair of cases for groups $G_1$ and $G_2$ (16 combinations total), and solve (2) under that assumption. However, this yields little structural insight into the behavior of $s_2^\star$ or the resulting representation ratio $\mathcal{R}(\beta, \gamma)$. In the next step, we characterize this structure piecewise by regime.

**Step 3: Identifying structural regimes via $\gamma$-thresholds.** Remarkably, our next result shows that only four of the sixteen case pairs (I–IV $\times$ I'–IV') arise in equilibrium, and that each occurs over a contiguous sub-interval of $[0, 1]$ characterized by three values $0 \leq \gamma_1 \leq \gamma_2 \leq \gamma_3 \leq 1$.

**Theorem 3.1 ($\gamma$-thresholds).** *For any fixed $\beta \geq 1 - c$, there exist unique values $\gamma_1 \leq \gamma_2 \leq \gamma_3 \in [0, 1]$ such that: (i) $s_2^\star(\gamma)/\beta \leq 1 - \gamma$ if and only if $\gamma \leq \gamma_1$; (ii) $s_2^\star(\gamma) \leq 1 - \gamma$ if and only if $\gamma \leq \gamma_2$; (iii) $s_2^\star(\gamma) \geq \gamma s_1^\star$ if and only if $\gamma \leq \gamma_3$. Moreover, these thresholds are given by: (i) $\gamma_1 = \frac{2s_1^\star(\beta+1/\beta)-4(1-c)}{2s_1^\star(\beta+1/\beta)-4(1-c)+(s_1^\star)^2(1+1/\beta^2)}$; (ii) $\gamma_2 = \frac{x}{x+1}$, where $x$ is the unique root in $(\gamma_1, \gamma_3)$ of $\left(1 + \frac{1}{\beta^2}\right)\frac{x^2(s_1^\star)^2}{2} + x\left(\frac{s_1^\star}{\beta} - \frac{s_1^\star}{\beta^2} - c\right) + \frac{1}{2}\left(1 - \frac{1}{\beta}\right)^2 = 0$; (iii) $\gamma_3 = \frac{1}{1+c}$.*

At a high level, the reason why only four out of the possible sixteen cases arise is as follows. When $\gamma = 0$, $L(\gamma, s_2)$ is horizontal and hence, we are in Case III (see Figure 1). We show that $s_2^\star(\gamma)/1 - \gamma$ and $s_2^\star(\gamma)/\gamma$ are monotonically increasing and decreasing functions of $\gamma$ respectively. Thus, as we raise $\gamma$, the $y$-intercept of $L(\gamma, s_2)$ increases and its $x$-intercepts decreases. Therefore, we transition from case III to case II and eventually to case IV (when $\gamma = 1$, $L(\gamma, s_2)$ is vertical) – note that case I never happens because in this case, the relevant area above $L(\gamma, s_2)$ is more than $1/2$, but the capacity at Institution 2 is at most $1/2$. The line $L_\beta(\gamma, s_2)$ also goes through these transitions as we raise $\gamma$. Since these two lines remain parallel (and $L_\beta$ lies above $L$) as we change $\gamma$, their transitions happen in a coordinated manner. Theorem 3.1 shows that the pair of cases corresponding to these two lines goes through the following transitions: (III, III'), (III, II'), (II, II'), (IV, IV'), and these regimes are separated by three critical thresholds—$\gamma_1$, $\gamma_2$, and $\gamma_3$.

These thresholds also mark qualitative changes in how the institutions evaluate candidates and balance the two attributes. For instance, when $\gamma < \gamma_1$, disadvantaged candidates can qualify based solely on their second attribute $v_{i2}$, but for $\gamma > \gamma_1$, they must also meet performance on $v_{i1}$, making admission more stringent. Likewise, above $\gamma_3$, both groups may qualify based on $v_{i1}$ alone. Between these values, the selection regime reflects blended dependence on both attributes, with differential effects across groups due to bias.

The proof of the first part of Theorem 3.1 relies on monotonicity of the scaled thresholds $\frac{s_2^\star(\gamma)}{\gamma}$ and $\frac{s_2^\star(\gamma)}{1-\gamma}$ that arises due to the structure of the equilibrium condition and the convex combination of attributes. The values of the $\gamma$-thresholds are derived as follows: for each $i \in \{1, 2, 3\}$, the definition of $\gamma_i$ also yields an expression for $s_2^\star(\gamma_i)$. For example, $s_2^\star(\gamma_1) = \beta(1 - \gamma_1)$. Since the threshold $s_2^\star(\gamma)$ also satisfies the capacity constraint (2), we get equations for solving $\gamma_i$.

Note that, while $\gamma_3 = \frac{1}{1+c}$ remains fixed for all $\beta$, we show in Proposition D.1 that the threshold functions $\gamma_1(\beta)$ and $\gamma_2(\beta)$ are strictly increasing and decreasing in $\beta$, respectively, and both converge to a common limit as $\beta \to 1$. In particular, $\lim_{\beta \to 1} \gamma_1(\beta) = \lim_{\beta \to 1} \gamma_2(\beta) = \frac{c}{1+c/2+c^2/4} < \gamma_3$. This implies that even in the absence of bias ($\beta = 1$), structural phase transitions in the equilibrium threshold $s_2^\star$ (and hence the representation ratio $\mathcal{R}$) still occur as evaluator alignment $\gamma$ increases. Our

framework thus provides insights not only into the compounding effects of bias and misalignment, but also into the residual impact of evaluator disagreement in the fully unbiased regime.

Finally, when $\beta < 1 - c$, the equilibrium admits a richer structure: there is an additional threshold $\gamma_4$ where $\gamma_4 = s_2^\star(\gamma_4)/\beta$. Further, there is a *critical* value $\beta_c \in [0, 1 - c]$ such that if $\beta < \beta_c$, the relative ordering of these thresholds is $\gamma_1, \gamma_3, \gamma_2, \gamma_4$; otherwise it is $\gamma_1, \gamma_2, \gamma_3, \gamma_4$. At this critical value of $\beta$, $\gamma_2$ equals $\gamma_3$, i.e., both the groups never satisfy the conditions in case II for a given value of $\gamma$. We defer a full treatment of that regime to Appendix K. We now use the threshold structure above to derive explicit expressions for $s_2^\star(\gamma)$ in each regime.

**Step 4: Solving for $s_2^\star(\gamma)$ in each regime.** The thresholds $\gamma_1, \gamma_2, \gamma_3$ partition $[0, 1]$ into four intervals, each corresponding to a fixed pair of geometric cases for groups $G_1$ and $G_2$, namely $(III, III'), (III, II'), (II, II'), (IV, IV')$. In each such regime, the expression for $\Pr[\hat{u}_{i2} \geq s_2 \wedge \hat{u}_{i1} < s_1]$ is known from Step 2, and we can solve equation (2) to obtain $s_2^\star(\gamma)$.

**Theorem 3.2** (**Equations for $s_2^\star(\gamma)$**)**.** *Assume $\beta \geq 1 - c$ and $c < \frac{1}{2}$. Then $s_2^\star(\gamma)$ satisfies:*

- $[0, \gamma_1]$: $s_1^\star - \frac{2s_1^\star s_2^\star - \gamma(s_1^\star)^2}{2(1-\gamma)} + \frac{s_1^\star}{\beta} - \frac{2s_1^\star s_2^\star - \gamma(s_1^\star)^2}{2\beta^2(1-\gamma)} = c$

- $[\gamma_1, \gamma_2]$: $s_1^\star - \frac{2s_1^\star s_2^\star - \gamma(s_1^\star)^2}{2(1-\gamma)} + \frac{(1-\gamma-s_2^\star/\beta+\gamma s_1^\star/\beta)^2}{2\gamma(1-\gamma)} = c$

- $[\gamma_2, \gamma_3]$: $\frac{(1-\gamma-s_2^\star+\gamma s_1^\star)^2}{2\gamma(1-\gamma)} + \frac{(1-\gamma-s_2^\star/\beta+\gamma s_1^\star/\beta)^2}{2\gamma(1-\gamma)} = c$

- $[\gamma_3, 1]$: $s_1^\star(1 + 1/\beta) - \frac{s_2^\star}{\gamma} - \frac{s_2^\star}{\beta\gamma} + \frac{1-\gamma}{\gamma} = c$

The piecewise equations above yield a unique value of $s_2^\star$ for each $\gamma$. In particular, even when the defining equation is quadratic, the interval-specific constraints on $s_2^\star$ (e.g., bounds derived from geometry or feasibility) ensure a single consistent solution. As $\gamma$ varies, $s_2^\star$ transitions through qualitatively distinct behaviors—decreasing, convex, and increasing—reflecting a subtle interplay between evaluator alignment and group-based access. We show in Theorem E.1 that $s_2^\star(\gamma)$ is unimodal (and, hence, non-monotone w.r.t. $\gamma$) and continuous, with matching slopes at regime boundaries. Note that when $\beta < 1 - c$, due to the emergence of an additional threshold $\gamma_4$, Theorem K.5 shows that $s_2^\star(\gamma)$ can now have several local minima or maxima (see Appendix K.3). These characterizations of $s_2^\star$ now allow us to derive the representation ratio $\mathcal{R}(\beta, \gamma)$, which we analyze in the next step.

**Step 5: Closed-form expression for representation ratio $\mathcal{R}(\beta, \gamma)$.** Using the expressions for $s_2^\star$ derived in Step 4, we now give piecewise closed-form expressions for $\mathcal{R}(\beta, \gamma)$.

**Theorem 3.3** (**Representation ratio**)**.** *Fix $c < \frac{1}{2}$ and $\beta \geq 1 - c$.*

*Define* $\Delta(\beta, \gamma) := -\beta s_1^\star + \sqrt{(s_1^\star)^2(\beta^2 + 1) - \frac{2(1-\gamma)((1-\beta)s_1^\star - c)}{\gamma}}$, *and* $\theta(\beta, \gamma) :=$ $\frac{-\beta(1-\beta)+\sqrt{-(1-\beta)^2 + \frac{2c\gamma(1+\beta^2)}{1-\gamma}}}{1+\beta^2}$. *Then:*

$$\mathcal{R}(\beta, \gamma) = \begin{cases} \frac{(\beta-(1-c))(\beta+1-c)+c^2}{1-\beta^2(1-2c)} & \text{if } \gamma \in [0, \gamma_1], & \frac{1-\frac{s_1^\star}{\beta}+\frac{\gamma\Delta^2(\beta,\gamma)}{2(1-\gamma)}}{1-s_1^\star+c-\frac{\gamma\Delta^2(\beta,\gamma)}{2(1-\gamma)}} & \text{if } \gamma \in [\gamma_1, \gamma_2], \\ \frac{1-\frac{s_1^\star}{\beta}+\frac{(1-\gamma)\theta^2(\beta,\gamma)}{2\gamma}}{1-s_1^\star+c-\frac{(1-\gamma)\theta^2(\beta,\gamma)}{2\gamma}} & \text{if } \gamma \in [\gamma_2, \gamma_3], & \frac{-(1-\beta)(1+\gamma)+4\gamma c}{(1+\gamma)(1-\beta)+4\gamma\beta c} & \text{if } \gamma \in [\gamma_3, 1]. \end{cases}$$

The proof of this theorem uses the expression (3) for $\mathcal{R}(\beta, \gamma)$ in terms of $s_1^\star$ and $s_2^\star$. For each of the regimes defined by $\gamma_1, \gamma_2, \gamma_3$, we use (3.2) to eliminate the explicit dependence of this expression on $s_2^\star$. The monotonicity of $\mathcal{R}(\beta, \gamma)$ with respect to $\gamma$ follows from analyzing these closed-form expressions. Details are given in Appendix F. As an example, consider the regime $\gamma \in [\gamma_2, \gamma_3]$. Using the equation for $s_2^\star$ in this interval (Step 4), we define $\Delta := \frac{s_2^\star - \gamma s_1^\star}{1-\gamma}$, and observe that the mass of group $G_1$ candidates selected by Institution 2 is $\frac{(1-\gamma)\Delta^2}{2\gamma}$. This expression, together with its analog for group $G_2$, yields an explicit formula for $\mathcal{R}(\beta, \gamma)$ in each regime.

We present a summary of the results in this section in Table 1. While our main focus is on representation, the framework also supports closed-form expressions of institutional utilities (see Appendix H). In the next section, we leverage these results to examine how $\mathcal{R}(\beta, \gamma)$ behaves jointly with $\beta$ and $\gamma$, and explore design strategies that mitigate bias while maintaining institutional efficiency.

| Structural property | Result and location | |
|---|---|---|
| Closed-form expression for $s_1^\star$ | $s_1^\star = \frac{2-c}{1+1/\beta}$ | (Eq. (4)) |
| Monotonicity of $s_1^\star$ w.r.t. $\beta$ | increasing in $\beta$ | (follows directly from Eq. (4)) |
| Four regimes of $\gamma$ | $\gamma_1, \gamma_2, \gamma_3$ | (Theorem 3.1) |
| piecewise expression for $s_2^\star$ | via characterization via $\gamma_1, \gamma_2, \gamma_3$ | (Theorem 3.2) |
| Unimodality of $s_2^\star$ w.r.t. $\gamma$ | $s_2^\star$ is continuous with at most 1 minimum | (Theorem E.1) |
| Monotonicity of $s_2^\star$ w.r.t. $\beta$ | increasing in $\beta$ | (Proposition C.1) |
| piecewise expression for $\mathcal{R}$ | 4-case formula | (Theorem 3.3) |
| Monotonicity of $\mathcal{R}$ in $\beta$ | increasing in $\beta$ for fixed $\gamma$ | (Proposition F.1) |

Table 1: Summary of structural properties of thresholds and representation metrics for $\beta \geq 1 - c$.

**Extension to general preferences.** We also extend our analysis to the *general preference setting*, where candidates prefer Institution 1 with probability $p \in [0, 1]$, as described in Appendix L. In this setting, the equilibrium thresholds $s_1^\star$ and $s_2^\star$ satisfy coupled nonlinear equations that lack closed-form solutions. In Appendix L.1, we derive these equilibrium conditions and prove the existence and uniqueness of the solution, ensuring that the model remains well-posed for all $p$. Using the probability expressions developed earlier, we show in Appendix L.2 that these thresholds can be computed efficiently through numerical evaluation. Building on these results, Appendix L.3 provides closed-form formulas for key outcome metrics—the representation ratio and institutional utilities—expressed directly in terms of $s_1^\star$ and $s_2^\star$. Analytically, we establish in Appendix L.4 that both thresholds vary monotonically with $p$, although not necessarily strictly: $s_1^\star(p)$ remains constant up to a critical point $p^\star$ and increases thereafter, corresponding to a regime where the selection probability $\Pr[u_{i1} < s_1^\star(p) \wedge u_{i2} \geq s_2^\star(p)]$ is zero. We formalize this structural break in Appendix L.4.1 and confirm it numerically. Consequently, both $\mathcal{R}$ and $\mathcal{N}$ decrease with $p$ and exhibit a clear *phase transition*: they remain constant for $p \leq p^\star$ and decline sharply thereafter. Finally, numerical experiments in Appendix L.5 and Figures 8–14 reveal how thresholds, representation ratios, and utilities co-evolve as $p$ and the correlation parameter $\gamma$ vary. The qualitative behavior of $\mathcal{R}(\beta, \gamma)$ and $\mathcal{N}(\beta, \gamma)$ remains monotone in $\beta$ and $\gamma$ even under general preferences, while varying $p$ introduces the new structural effects described above—monotone yet piecewise regimes and distinct phase transitions that extend the continuum-limit analysis.

# 4 Analysis and intervention

A central goal of this work is to guide interventions that improve representation for disadvantaged groups. Toward this, we study how fairness metrics vary with two systemic levers: evaluator bias ($\beta$) and institutional correlation ($\gamma$). While our main structural result yields closed-form expressions for the representation ratio $\mathcal{R}(\beta, \gamma)$ (Theorem 3.3), its policy relevance depends on understanding how $\mathcal{R}$ and its normalized counterpart $\mathcal{N}$ evolve with these parameters. This section is organized into four parts. We begin by analyzing how $\mathcal{R}(\beta, \gamma)$ varies with $\beta$ and $\gamma$, highlighting regimes of monotonicity, sharp transitions, and diminishing returns. We then study the variation of normalized ratio $\mathcal{N}(\beta, \gamma)$ with respect to $\beta$ and $\gamma$. Building on these insights, we formulate a framework for intervention planning under fairness targets and show how to use it.

**Monotonicity, transitions, and concavity of the representation ratio.** We begin by asking: How does $\mathcal{R}(\beta, \gamma)$ vary with the bias level $\beta$ and evaluator correlation $\gamma$? For any fixed $\gamma$, the dependence on $\beta$ is predictable: as bias decreases (i.e., $\beta$ increases), selection thresholds $s_1^\star$ and $s_2^\star$ become stricter for the advantaged group, yielding a higher representation ratio (see Appendix F.2). The variation of $\mathcal{R}(\beta, \gamma)$ with $\gamma$ (for a fixed $\beta \geq 1 - c$), however, is more subtle. Since $\gamma$ controls how correlated the two institutions are in their evaluation criteria, one might expect that decreasing $\gamma$ allows for more diverse evaluation and improves access for disadvantaged candidates. While $s_1^\star$ does not change with $\gamma$, $s_2^\star(\gamma)$ can increase or decrease with $\gamma$: it follows a U-shaped trajectory as $\gamma$ increases (see Theorem E.1). This raises a natural question: can $\mathcal{R}(\beta, \gamma)$ be non-monotonic with $\gamma$ as well? The answer is no in the regime $c < 1/2$ and $\beta \geq 1 - c$.

**Corollary 4.1.** *Fix $c < 1/2$ and $\beta \geq 1 - c$. Then $\mathcal{R}(\beta, \gamma)$ increases monotonically in $\gamma$.*

In particular, $\mathcal{R}(\beta, \gamma)$ is maximized when $\gamma = 1$: $\mathcal{R}(\beta, \gamma) \leq \mathcal{R}(\beta, 1)$ for all $\gamma \in [0, 1]$. The proof heavily relies on the piecewise formulas derived in Theorem 3.3 and is deferred to Appendix F.1. Figure 2 illustrates the contrast: while $s_2^\star$ dips and then rises with $\gamma$, the representation ratio $\mathcal{R}(\beta, \gamma)$ increases steadily—more sharply so for larger values of $\beta$. Note, however, that this monotonicity

may not hold outside the (less relevant) regime $\beta \geq 1 - c$. In particular, when $\beta < 1 - c$, no one from $G_2$ can be admitted to Institution 1, and this equilibrium structure can reverse the fairness trend with increasing $\gamma$; see Proposition K.8 for details.

Importantly, using the piecewise characterization from Theorem 3.3, we can generalize prior results by [14] on representation ratio. [14] considered the case $\gamma = 1$ and showed that $\mathcal{R}(\beta, 1)$ varies approximately linearly with $\beta$. Theorem 3.3 shows that the behavior of $\mathcal{R}$ w.r.t. $\beta$ can be highly non-linear in the presence of $\gamma$. We highlight this with phenomena with two examples.

**Example 1 (Quadratic decay).** Let $c = 1/2 - \varepsilon$. On the one hand, when $\gamma = 1$, the last case of Theorem 3.3 implies $\mathcal{R}(\beta, 1) = \frac{-(1-\beta)(1+1)+4(1)(1/2-\varepsilon)}{(1+1)(1-\beta)+4(1)\beta(1/2-\varepsilon)} = \frac{\beta - 2\varepsilon}{1 - 2\beta\varepsilon}$, which scales like $\beta$ for small enough $\varepsilon$. On the other hand, when $\gamma \leq \gamma_1$, from the first case in Theorem 3.3, we obtain that $\mathcal{R}(\beta, \gamma) = \frac{\beta^2 + 2\varepsilon}{1 + 2\varepsilon\beta^2} \approx \beta^2$ for small enough $\varepsilon$. Thus, $\mathcal{R}$ degrades from being linear in $\beta$ when $\gamma \approx 1$ to quadratic in $\beta$ when $\gamma \approx 0$.

**Example 2 (Amplification in low-selectivity regime).** Consider a highly selective regime, where $c$ is small and $\beta \approx 1 - c$. For $\gamma \leq \gamma_1$, Theorem 3.3 yields: $\mathcal{R}(1 - c, \gamma) = \frac{c^2}{1-(1-c)^2(1-2c)} \approx c/4$. In contrast, when $\gamma = 1$, using Theorem 3.3, we obtain: $\mathcal{R}(1 - c, 1) = \frac{-2c+4c}{2c+4c(1-c)} = \frac{c}{3c-2c^2} \approx \frac{1}{3}$ for small enough $c$. Thus, while $\mathcal{R}(1, \gamma) = 1$ for all $\gamma$, a slight decrease in $\beta$ (from 1 to $1 - c$) results in a huge gap in representation ratio near the two extremes of $\gamma$ in a highly-selective setup.

To summarize, Theorem 3.3 and Corollary 4.1 imply that $\mathcal{R}(\beta, \gamma)$ is piecewise-defined across four structural regimes, separated by thresholds $\gamma_1, \gamma_2, \gamma_3$. It remains flat in the low-$\gamma$ regime ($\gamma \leq \gamma_1$), non-linearly increases through $[\gamma_1, 1]$. Taken together, these insights underscore that the representation ratio may not be uniformly responsive to interventions changing $\beta$ and $\gamma$ due to the compounding effect of these parameters on $\mathcal{R}$.

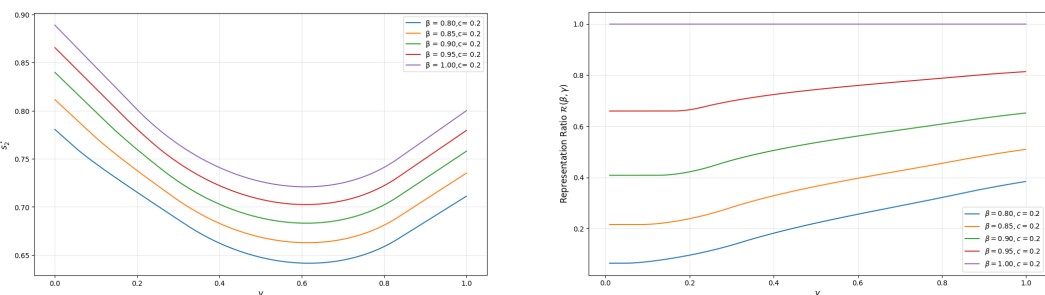

Figure 2: Variation of (left) selection threshold $s_2^\star$ and (right) representation ratio $\mathcal{R}(\beta, \gamma)$ as a function of $\gamma$, for fixed $c = 0.2$.

**Measuring fairness loss: Normalized representation and its response to interventions.** Recall that the normalized representation ratio is defined as $\mathcal{N}(\beta, \gamma) := \frac{\mathcal{R}(\beta, \gamma)}{\max_{\gamma' \in [0,1]} \mathcal{R}(\beta, \gamma')}$. Since $\mathcal{R}(\beta, \gamma) \leq \mathcal{R}(\beta, 1)$ for $\gamma \in [0, 1]$, this simplifies to $\mathcal{R}(\beta, \gamma)/\mathcal{R}(\beta, 1)$. Note that $0 \leq \mathcal{N}(\beta, \gamma) \leq 1$ quantifies the fraction of the maximum achievable fairness retained for a fixed $\beta$. There are two key intervention levers: increasing $\beta$ (reducing bias) and increasing correlation ($\gamma$) to increase $\mathcal{N}$. However, these interventions, in the least, require that $\mathcal{N}$ is monotonically non-decreasing in $\beta$ and $\gamma$. From Corollary 4.1, we already know that $\mathcal{R}(\beta, \gamma)$, and hence $\mathcal{N}(\beta, \gamma)$, increases monotonically with $\gamma$. What is less obvious—but follows from the closed-form expressions in Theorem 3.3 is that $\mathcal{N}(\beta, \gamma)$ is also monotonically increasing in $\beta$ when $c \leq 1/2$; see Theorem F.2. Figure 3 (left) illustrates this trend. The growth of $\mathcal{N}(\beta, \gamma)$ is phase-dependent: flat in the low-alignment regime ($\gamma \leq \gamma_1$), sharp in intermediate phases, and nearly linear for $\gamma \in [\gamma_3, 1]$. The most severe degradation occurs when both $\beta$ and $\gamma$ are small—highlighting the importance of addressing both alignment and bias in fairness interventions. Extending Example 1 above (the case $c \approx 1/2$. When $\gamma = 1$, we have $\mathcal{N}(\beta, \gamma) = 1$ by definition. But as $\gamma \to 0$, $\mathcal{R}(\beta, \gamma) \approx \beta^2$, while $\mathcal{R}(\beta, 1) \approx \beta$, implying $\mathcal{N}(\beta, \gamma) \approx \beta$. In summary, even after normalization, fairness never deteriorates when bias is reduced or alignment improves, and $\beta$ and $\gamma$ jointly exert a compounded effect on $\mathcal{N}$. These properties imply that the normalized representation ratio is both stable and directionally reliable—ensuring that interventions targeting either parameter will monotonically improve fairness.

**Target-based intervention planning.** We can now leverage the structure of $\mathcal{N}(\beta, \gamma)$ to design and evaluate intervention strategies that improve representation. One may consider two classes of interventions: (1) continuous interventions that incrementally reduce bias or increase correlation, and (2) structural interventions such as capacity reservations. In the fully aligned setting ($\gamma = 1$), [15] shows that reserving seats in each institution proportional to group sizes, and applying stable matching within each group, guarantees perfect representation. A similar strategy can be shown (by extending their approach) to achieve a representation ratio of exactly 1 for arbitrary values of $\beta$ and $\gamma$. This yields a non-parametric, structure-based intervention that ensures fairness independently of bias mitigation or evaluator alignment.

In contrast, interventions that modify the evaluation process, by increasing ($\beta$) or increasing ($\gamma$), may provide additional flexibility in real-world selection systems. For instance, in AI-mediated settings, $\beta$ can be increased through debiasing or anonymized scoring; $\gamma$ can be increased using standardized models, calibration protocols, or shared training data. In human-mediated settings, increasing $\beta$ might involve structured rubrics, blind evaluation, or bias training; increasing $\gamma$ could be achieved by coordinated evaluation guidelines, common score sheets, or centralized vetting frameworks.

We now show how a system designer, given current parameters $(\beta_0, \gamma_0)$ and a fairness target $\tau \in [0, 1]$, can use our framework to plan interventions. Since $\mathcal{N}(\beta, \gamma)$ is increasing in both $\beta$ and $\gamma$, any target $\tau \leq 1$ is attainable by increasing either parameter. We compute the *Pareto frontier*—the set of minimal $(\beta, \gamma)$ pairs that just achieve $\mathcal{N}(\beta, \gamma) \geq \tau$, using Theorem 3.3. This frontier reveals all efficient tradeoffs between bias mitigation and evaluator alignment that meet the target. Figure 3 (right) illustrates this frontier for $\tau = 0.8$, with a heatmap of $\mathcal{N}(\beta, \gamma)$. The red contour shows the threshold $\mathcal{N} = \tau$; points above meet the goal, while those below do not. The shape of the frontier reveals which direction—$\beta$ or $\gamma$—yields more efficient gains. Notably, the frontier becomes vertical when $\gamma \leq \gamma_1(\beta)$, reflecting the fact that $\mathcal{N}$ is flat in this region. Since $\gamma_1$ increases with $\beta$, this insensitivity band widens as bias decreases. Overall, the frontier offers practical guidance: starting from $(\beta_0, \gamma_0)$, one can locate the nearest feasible point above the contour to meet the fairness target.

We illustrate this with a concrete example (Appendix G). Fix $\beta_0 = 0.85$, $\gamma_0 = 0.40$, and $c = 0.20$. We first compute $s_1^\star \approx 0.8276$ using Equation (4), and then use Theorem 3.1 to identify the applicable regime for $\gamma$. Applying Theorem 3.3, we obtain the values $\mathcal{R}(\beta_0, \gamma_0) \approx 0.329$, $\mathcal{R}(\beta_0, 1) \approx 0.510$, and $\mathcal{N}(\beta_0, \gamma_0) \approx 0.644$. From Figure 3 (right), we observe that achieving a fairness target of $\tau = 0.80$ requires either $\gamma \approx 0.640$ or $\beta \approx 0.911$. This demonstrates that modest improvements in either evaluator alignment or bias reduction are sufficient to meet the desired fairness level.

Finally, note that the baseline parameters $(\beta_0, \gamma_0)$ may be estimated directly from observed data. The bias parameter $\beta$ can be recovered from the observed threshold used by Institution 1 via $\hat{\beta}_0 = \frac{\hat{s}_1^\star}{2 - c - \hat{s}_1^\star}$ as follows from Equation (4). Holding $\beta = \hat{\beta}_0$ fixed, the correlation parameter $\gamma$ can be estimated by numerically inverting the monotone relation $\gamma \mapsto \mathcal{R}(\hat{\beta}_0, \gamma)$ (or equivalently $\mathcal{N}$): $\hat{\gamma}_0 = \arg\min_{\gamma \in [0,1]} |\mathcal{R}(\hat{\beta}_0, \gamma) - \hat{\mathcal{R}}|$ which admits a unique solution by Theorem 3.3.

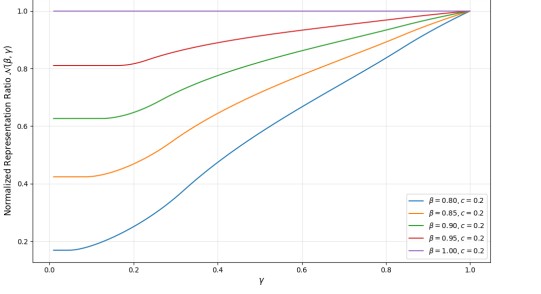 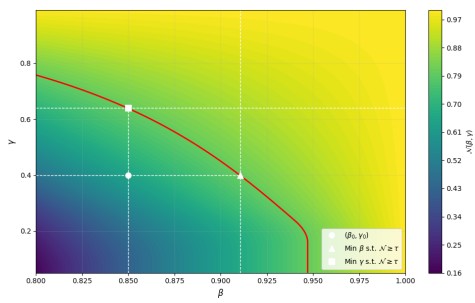

Figure 3: (Left) $\mathcal{N}(\beta, \gamma)$ and (right) Pareto frontier of $(\beta, \gamma)$ pairs achieving $\mathcal{N}(\beta, \gamma) \geq \tau = 0.8$, for $c = 0.2$. Starting from $(\beta_0 = 0.85, \gamma_0 = 0.4)$, the minimum interventions required to exceed the threshold are: $\beta \geq 0.911$ (with $\gamma_0 = 0.4$) or $\gamma \geq 0.640$ (with $\beta_0 = 0.85$).

# 5 Conclusion, extensions, limitations, and future work

We introduced a mathematical framework to study the structure of stable matchings in a two-institution setting where candidate evaluations are partially correlated and one group experiences group-level bias. Our focus was on how fairness—measured via the representation ratio—evolves with evaluator bias ($\beta$) and correlation ($\gamma$). Our key technical contributions include: (i) a closed-form characterization of the equilibrium thresholds $s_1^\star$ and $s_2^\star(\gamma)$ that govern the stable matching (Equation (4), Theorem 3.2); (ii) a piecewise expression for the representation ratio $\mathcal{R}(\beta, \gamma)$ (Theorem 3.3); and (iii) an analytic description of the structural $\gamma$-thresholds $\gamma_1, \gamma_2, \gamma_3$ that govern transitions in selection (Theorem 3.1).

Despite the non-monotonicity of $s_2^\star(\gamma)$, we show that $\mathcal{R}(\beta, \gamma)$ and its normalized variant $\mathcal{N}(\beta, \gamma)$ increase monotonically in $\gamma$ (Corollary 4.1). These findings support the design of fairness interventions: we use $\mathcal{N}(\beta, \gamma)$ to compute the Pareto frontier of bias–correlation combinations that achieve a target representation level $\tau$. Our main analysis focuses on the regime $\beta \geq 1 - c$ and full candidate preference for Institution 1. In Appendices K and L, we extend the model to handle stronger bias ($\beta < 1 - c$) and partial preferences, and show that several key structural properties persist.

The model makes several simplifying assumptions to ensure analytical tractability. Candidate attributes are drawn independently and uniformly from $[0, 1]$, following standard assumptions in prior work on fairness in matching and screening [18, 29, 15]. While this enables clean derivations and actionable intervention design, it does not capture the full heterogeneity or noise present in real-world evaluations. Some generalizations are straightforward. For example, our techniques extend naturally to settings with asymmetric capacities ($c_1 \neq c_2$) or candidate masses ($\nu_1 \neq \nu_2$). Additionally, results from [4] can be used to transfer our continuum-based analysis to finite $n$, with approximation error $O(1/\sqrt{n})$; see Appendix B. The framework and techniques also extend to several richer settings beyond the symmetric, independent case. First, allowing asymmetric bias across institutions ($\beta_1 \neq \beta_2$) shows that greater evaluator alignment ($\gamma$) continues to improve representation, as less-biased institutions can "rescue" disadvantaged candidates who narrowly miss selection elsewhere (Appendix I). Second, in the additive-bias formulation (Appendix J), where disadvantaged candidates face a fixed evaluation penalty rather than a multiplicative discount, the equilibrium structure and existence of $\gamma$-thresholds persist with only quantitative changes to the cutoff equations. Third, when candidate attributes are drawn from a smooth but non-uniform distribution with cumulative distribution function $\Phi$, the equilibrium thresholds satisfy $\Phi(s_1^\star) + \Phi\left(\frac{s_1^\star}{\beta}\right) = 2(1 - c_1)$, generalizing the uniform case (4). Although closed-form expressions are no longer available, the expressions used to compute probabilities such as $\Pr[u_{i1} < s_1, u_{i2} \geq s_2]$ can be written in terms of $\Phi$, and the resulting thresholds and fairness metrics remain numerically tractable. We omit the details. That said, if candidate attributes exhibit strong dependence—for example, if all candidates have identical or highly similar profiles—the concentration results needed for the finite-to-infinite reduction may not apply. In such cases, convergence to a deterministic limit could fail, and our monotonicity guarantees may no longer hold. Nonetheless, we expect the qualitative trend—that increasing $\gamma$ improves $\mathcal{R}$—to persist under weaker assumptions, such as mild correlations or clustered attribute structures. Extending the convergence analysis to such settings remains an open and valuable direction for future work.

Several directions remain open for future work. First, incorporating strategic behavior—such as candidates adjusting effort or signaling based on perceived bias—would enable a game-theoretic analysis of incentives. Second, extending the evaluation model to include multi-attribute correlation, group-dependent score distributions, or asymmetric institutional preferences could better capture practical complexity. Beyond the static analysis presented here, a natural next step is to examine how the representation ratio $\mathcal{R}$ might evolve over time in systems that repeat or adapt across cycles—such as admissions, hiring, or promotions. In such dynamic environments, past matching outcomes could shape future candidate preferences, group participation, or institutional strategies. This feedback could lead to rich temporal effects: higher representation might enhance perceived fairness or institutional reputation, creating a virtuous cycle, whereas persistent underrepresentation might discourage participation and reinforce disparities. Capturing these dynamics would require an explicitly game-theoretic or behavioral model with endogenous feedback, a direction we see as particularly promising for future work. Finally, integrating this framework with empirical data to estimate or learn ($\beta, \gamma$), and adapting interventions based on observed outcomes, could support adaptive fairness mechanisms in real-world deployments.

## Acknowledgments

We thank L. Elisa Celis for useful comments on the paper. This work was funded by NSF Award CCF-2112665, and in part by grants from Tata Sons Private Limited, Tata Consultancy Services Limited, Titan.

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

# Contents

# A Detailed related work

The game-theoretic study of assignment and selection problems, initiated by [24], has expanded into a broad literature exploring stability, truthfulness, incomplete information, and machine learning approaches to preference inference [48, 49, 50, 21, 45, 34, 8, 32, 33, 39]. Comprehensive overviews are provided in [51, 37].

Several empirical studies examine biases in algorithmic assignment and selection systems across real-world contexts, including centralized admissions and online labor markets [56, 26, 31, 30].

Our work builds on and extends the framework of [15], which studies a centralized multi-institution matching model where a single entity evaluates candidates. Their model assumes a uniform evaluation process across institutions and systematic bias against disadvantaged candidates, leading to unfair and inefficient outcomes. They propose fairness-aware selection algorithms to mitigate these issues. We generalize this setting by introducing *correlated institutional evaluations* and analyzing how *bias and correlation jointly shape fairness and representation*.

A related line of research examines the role of correlation in candidate evaluations (e.g., [6, 5, 3, 12]). [12] studies a model where institutions assign correlated scores while maintaining unchanged marginal distributions. Unlike their work, which focuses on emergent statistical discrimination due to varying correlation across demographic groups, we analyze *explicit biases* in evaluations and their compounded effect with correlation on stable matching

A parallel and complementary line of work studies correlated evaluations in continuum matching markets using stylized models of noisy signal generation. In particular, [57] and [58] analyze how shared or idiosyncratic noise across institutions shapes diversity, efficiency, and learning dynamics. While their models abstract away from institutional selection rules and stability constraints, they highlight how evaluation noise and correlation can lead to monoculture outcomes or suppress informative distinctions between candidates. Our work differs in modeling two-sided stable matchings with explicit score-based thresholds, but shares a broader motivation: understanding how evaluation design influences representational and welfare outcomes.

Other studies explore decentralized selection systems where institutions evaluate candidates using multiple criteria. [18] examines decentralized matching but focus on institution-specific fit measures and strategic behavior, whereas our model incorporates both *correlated evaluations and biases*. Despite these differences, both approaches highlight inefficiencies in decentralized allocation.

Our work also connects to research using a continuum model of students and institutions, which simplifies analysis by treating admissions as a supply-and-demand problem. [17] study noisy priorities, while [7] develop a framework for characterizing stable matchings. [4] refine these models to improve approximations in finite markets. We apply similar techniques to analyze equilibrium thresholds under bias and correlation.

The role of correlation in student priorities has been examined in school choice, particularly regarding tie-breaking lotteries and welfare outcomes. [1, 4] compare settings where institutions share a common lottery versus independent lotteries, corresponding to $\gamma = 1$ (fully correlated) and $\gamma = 0$ (independent evaluations) in our model.

Beyond school choice, other studies analyze the correlation between student preferences and priorities. [11] examines one-to-one matching mechanisms that avoid systematic disadvantages, while [19] study how correlation affects the stability-efficiency trade-off when comparing Deferred Acceptance and Top Trading Cycles.

Our work also relates to empirical studies on biases in real-world allocation systems, including school admissions and online labor markets [56, 26, 31, 30]. [12] theoretically analyzes selection under differential noise, showing that increased noise for disadvantaged groups can harm both groups by reducing the probability of securing a preferred match. In contrast, we model bias as a systematic downscaling of scores and analyze its effect on *representation and fairness metrics*.

Finally, we discuss related work on bias in algorithmic decision-making and corresponding interventions. Several studies investigate core algorithmic problems when inputs are subject to bias, including subset selection [29, 23, 52, 13, 25, 38], ranking [16], and classification [9]. Another line of research focuses on fairness constraints in selection problems, such as ensuring proportional representation [29, 20, 27], with the goal of mitigating bias through explicit constraints.

# B   From finite to infinite: A continuum stable matching game

**Equations for thresholds.** We consider a two-institution matching setting, where the institutions have capacities $c_1 n$ and $c_2 n$, respectively. Candidates belong to either an advantaged group $G_1$ or a disadvantaged group $G_2$, with group sizes $|G_1| = \nu_1 n$ and $|G_2| = \nu_2 n$. Here, $c_1, c_2, \nu_1, \nu_2$ are constants. Recall that every candidate prefers Institution 1 to Institution 2, and that the observed utilities $\hat{u}_{i1}$ and $\hat{u}_{i2}$ are generated as described in Section 2. In a deterministic setting, the following result holds. The proof appears in Appendix B.1.

**Proposition B.1** (**Thresholds in the deterministic case**). *Consider a deterministic setting with two institutions and two groups such that the observed utilities $\hat{u}_{i\ell}$ are all distinct. Then, there is a unique stable matching $M$ that can be described by two thresholds $s_1$ and $s_2$ as follows: $M^{-1}(1) = \{i : \hat{u}_{i1} \geq s_1\}$, and $M^{-1}(2) = \{i : \hat{u}_{i1} < s_1 \text{ and } \hat{u}_{i2} \geq s_2\}$.*

The key intuition behind the proof is that since all candidates strictly prefer Institution 1, any stable assignment must first allocate the top $c_1 n$ candidates—ranked by $\hat{u}_{i1}$—to Institution 1. Among the remaining candidates, Institution 2 then selects the top $c_2 n$ candidates based on their $\hat{u}_{i2}$ values. The thresholds $s_1$ and $s_2$ correspond to the lowest $\hat{u}_{i1}$ and $\hat{u}_{i2}$ values among candidates assigned to Institution 1 and Institution 2, respectively.

In the stochastic setting, the observed utilities are random, and so are the thresholds $S_1$ and $S_2$ that determine the matching. The stability conditions in the finite setting require that exactly $c_2 n$ candidates are assigned to Institution 1 and $c_2 n$ candidates to Institution 2. That is, conditioning on the thresholds $S_1$ and $S_2$, we have

$$\sum_{i \in G_1} \mathbb{1}(\hat{u}_{i1} \geq S_1) + \sum_{i \in G_2} \mathbb{1}(\hat{u}_{i1} \geq S_1) = c_1 n, \tag{5}$$

$$\sum_{i \in G_1} \mathbb{1}(\hat{u}_{i2} \geq S_2, \hat{u}_{i1} < S_1) + \sum_{i \in G_2} \mathbb{1}(\hat{u}_{i2} \geq S_2, \hat{u}_{i1} < S_1) = c_2 n. \tag{6}$$

Taking expectations conditioned on $S_1, S_2$ (and then over the joint distribution of $(S_1, S_2)$), we obtain

$$\mathbb{E}\left[ \sum_{i \in G_1} \Pr[\hat{u}_{i1} \geq S_1 \mid S_1, S_2] + \sum_{i \in G_2} \Pr[\hat{u}_{i1} \geq S_1 \mid S_1, S_2] \right] = c_1 n, \tag{7}$$

$$\mathbb{E}\left[ \sum_{i \in G_1} \Pr[\hat{u}_{i2} \geq S_2, \hat{u}_{i1} < S_1 \mid S_1, S_2] + \sum_{i \in G_2} \Pr[\hat{u}_{i2} \geq S_2, \hat{u}_{i1} < S_1 \mid S_1, S_2] \right] = c_2 n. \tag{8}$$

Dividing by $n$, we have

$$\nu_1 \mathbb{E}\left[ \Pr[\hat{u}_{i1} \geq S_1 \mid S_1, S_2] \right] + \nu_2 \mathbb{E}\left[ \Pr[\hat{u}_{i'1} \geq S_1 \mid S_1, S_2] \right] = c_1, \tag{9}$$

$$\nu_1 \mathbb{E}\left[ \Pr[\hat{u}_{i2} \geq S_2, \hat{u}_{i1} < S_1 \mid S_1, S_2] \right] + \nu_2 \mathbb{E}\left[ \Pr[\hat{u}_{i'2} \geq S_2, \hat{u}_{i'1} < S_1 \mid S_1, S_2] \right] = c_2, \tag{10}$$

where $i \in G_1, i' \in G_2$. [7] show that as $n \to \infty$ the random thresholds $S_1$ and $S_2$ concentrate and converge almost surely to deterministic limits $s_1$ and $s_2$ respectively. Therefore, in the large-market (mean-field) limit, the conditional probabilities can be replaced by the probabilities evaluated at the limits, so that the above equations reduce to

$$\nu_1 \Pr[\hat{u}_{i1} \geq s_1] + \nu_2 \Pr[\hat{u}_{i'1} \geq s_1] = c_1, \tag{11}$$

$$\nu_1 \Pr[\hat{u}_{i2} \geq s_2, \hat{u}_{i1} < s_1] + \nu_2 \Pr[\hat{u}_{i'2} \geq s_2, \hat{u}_{i'1} < s_1] = c_2. \tag{12}$$

Equations (1) and (12) are the continuum or mean-field equations for the thresholds $s_1$ and $s_2$. [4] extends the mean-field analysis by establishing finite-sample concentration bounds for the thresholds. From his result, one can deduce that for finite $n$ the random thresholds $S_1$ and $S_2$ approximate the deterministic limits $s_1$ and $s_2$ with high probability, with all errors being of order $1/\sqrt{n}$. We now give the conditions under which these equations have a unique solution.

**Proposition B.2** (**Existence and uniqueness of thresholds**). *Assume that $c_1 + c_2 \leq \nu_1 + \nu_2$. Then, a solution $(s_1^\star, s_2^\star)$ satisfying (11)-(12) exists. Moreover, the solution is unique, provided that the probability distributions of $\hat{u}_{i1}$ and $\hat{u}_{i2}$ are strictly decreasing and continuous.*

The conditions of this proposition hold when the underlying distribution is uniform on $[0, 1]$, ensuring the existence of a unique solution $s_1^\star, s_2^\star$ as functions of the model parameters. The proof appears in Appendix B.2. For a fixed $\beta$, we also define $s_{1,\beta}^\star = \min(1, s_1^\star/\beta)$ and $s_{2,\beta}^\star = \min(1, s_2^\star/\beta)$.

**Expressions for metrics in terms of thresholds.** Given thresholds $s_1^\star$ and $s_2^\star$ for the two institutions respectively, the ratio of the measure of candidates from $G_1$ that are assigned to one of these

institutions to the measure of $G_1$ is given by (here $i \in G_1$): $\Pr[\widehat{u}_{i1} \geq s_1^\star] + \Pr[\widehat{u}_{i2} \geq s_2^\star, \widehat{u}_{i1} < s_1^\star]$. The corresponding quantity for $G_2$ can be expressed similarly. Thus, if $i \in G_1, i' \in G_2$, then representation ratio is equal to

$$\mathcal{R} = \frac{\Pr[\widehat{u}_{i'1} \geq s_1^\star] + \Pr[\widehat{u}_{i'2} \geq s_2^\star, \widehat{u}_{i'1} < s_1^\star]}{\Pr[\widehat{u}_{i1} \geq s_1^\star] + \Pr[\widehat{u}_{i2} \geq s_2^\star, \widehat{u}_{i1} < s_1^\star]}. \tag{13}$$

Given the threshold $s_1^\star$, the (observed) utility derived by Institution 1 from a candidate $i$ is 0 if $\widehat{u}_{i1} < s_1^\star$; $\widehat{u}_{i1}$ otherwise. Therefore, the utility of Institution 1 is given by (here $i \in G_1, i' \in G_2$):

$$\mathcal{U}_1 = \nu_1 \int_0^1 s \mathbb{1}[s \geq s_1^\star] d\mu_s + \nu_2 \int_0^1 s \mathbb{1}[s \geq s_1^\star] d\mu_s', \tag{14}$$

where $\mu_s$ is the measure induced by the distribution of $\widehat{u}_{i1}$ for $i \in G_1$ (and similarly for $\mu_s'$). The utility of Institution 2 can be expressed similarly.

**Continuum game interpretation.** We end this section by interpreting equations (11) and (12) as stability conditions in an infinite matching game, where $G_1$ and $G_2$ represent continuous populations rather than discrete candidates. Consider a setting where candidates belong to two groups, $G_1$ and $G_2$, with population measures $\nu_1$ and $\nu_2$, respectively, so that the total population is $\nu_1 + \nu_2 = 1$. Institutions 1 and 2 have fractional capacities $c_1$ and $c_2$, satisfying the feasibility constraint: $c_1 + c_2 \leq \nu_1 + \nu_2 = 1$. Admissions follow a threshold-based rule:

- Candidates with $\hat{u}_{i1} \geq s_1$ are assigned to Institution 1.

- Among the remaining candidates, those with $\hat{u}_{i2} \geq s_2$ are assigned to Institution 2.

- All others remain unmatched.

The thresholds $(s_1, s_2)$ must satisfy Equations (11) and (12). Equation (11) ensures Institution 1 selects exactly $c_1$ candidates, while Equation (12) ensures Institution 2 fills exactly $c_2$ positions from candidates not admitted to Institution 1.

In finite matching models, stability requires that no candidate–institution pair forms a *blocking coalition*, where both would prefer to be matched to each other over their current assignment. In the continuum setting, stability is implicitly enforced by the threshold structure and capacity constraints. Candidates assigned to Institution 2 or left unmatched cannot move to Institution 1 unless $s_1$ decreases. However, decreasing $s_1$ would admit more than $c_1$ candidates, violating capacity constraints. Similarly, Institution 1 cannot replace a lower-ranked candidate with a higher-ranked one without exceeding $c_1$ or displacing an admitted candidate, contradicting its ranking rule. Candidates assigned to Institution 1 or left unmatched cannot move to Institution 2 unless $s_2$ decreases, which would admit more than $c_2$ candidates, again violating capacity constraints. Since each institution fills to capacity with the highest-ranked available candidates, and no candidate can move to a more preferred institution without violating constraints, the resulting assignment is *stable by construction*. The equilibrium equations (11)–(12) thus encode the continuum equivalent of a stable matching.

## B.1 Proof of Proposition B.1

*Proof.* Since the number of candidates is more than the total available capacity, it is easy to check that any stable assignment must fill all the available capacity, i.e., it must assign $c_1 n$ candidates to Institution 1 and $c_2 n$ candidates to Institution 2. Let $S_1$ be the top $c_1 n$ candidates according to the utility value $\widehat{u}_{i1}$ (observe that we are using the fact that observed utilities are distinct and hence this set is well defined). We first argue that any stable assignment $M$ must assign $S_1$ to Institution 1. Indeed, suppose $i \in S_1$ is not assigned to Institution 1 by $M$. Then there must be a candidate $i' \notin S_1$ that gets assigned to Institution 1. This creates an instability: Institution 1 prefers $i$ to $i'$, and $i$ prefers its assignment in $M$ (which could be Institution 2 or unassigned) to Institution 1. Thus, $M$ must assign exactly $S_1$ to Institution 2.

Similarly, if $G$ denotes the set of all candidates and $S_2$ is the subset of $G \setminus S_1$ containing the top $c_2 n$ candidates according to $\widehat{u}_{i2}$ values, then any stable assignment must assign $S_2$ to Institution 2. This establishes the uniqueness of a stable assignment. The thresholds $s_1$ is the minimum $\widehat{u}_{i1}$ value of a candidate in $S_1$, and similarly for $s_2$. $\qquad\square$

## B.2 Proof of Proposition B.2

*Proof.* **Step 1: Continuity and monotonicity of allocation functions.** Define the allocation functions:

$$A_1(s_1) = \nu_1 \Pr[\hat{u}_{i1} \geq s_1] + \nu_2 \Pr[\hat{u}_{i'1} \geq s_1],$$
$$A_2(s_1, s_2) = \nu_1 \Pr[\hat{u}_{i2} \geq s_2, \hat{u}_{i1} < s_1] + \nu_2 \Pr[\hat{u}_{i'2} \geq s_2, \hat{u}_{i'1} < s_1].$$

Since $\Pr[\hat{u}_{ij} \geq s]$ is the survival function $1 - F_{ij}(s)$, where $F_{ij}(s)$ is the cumulative distribution function (CDF), it follows that:

- $A_1(s_1)$ is a continuous, strictly decreasing function of $s_1$.

- $A_2(s_1, s_2)$ is continuous in both $s_1$ and $s_2$.

**Step 2: Boundary conditions.** Consider the extreme cases:

- If $s_1 = 1$, then $\Pr[\hat{u}_{i1} \geq 1] = 0$, so $A_1(1) = 0$.

- If $s_1 = 0$, then $\Pr[\hat{u}_{i1} \geq 0] = 1$, so $A_1(0) = \nu_1 + \nu_2$.

- If $s_2 = 1$, then $A_2(s_1, 1) = 0$.

- If $s_2 = 0$, then $A_2(s_1, 0)$ reaches its maximum possible value $\nu_1 + \nu_2$.

Thus, for any feasible capacities $c_1, c_2$ satisfying $c_1 + c_2 \leq \nu_1 + \nu_2$, the thresholds $s_1, s_2$ can be adjusted to ensure feasibility.

**Step 3: Existence of a solution.** Define the function:

$$F(s_1, s_2) = (A_1(s_1) - c_1, A_2(s_1, s_2) - c_2).$$

Since $A_1(s_1)$ is strictly decreasing and continuous, and $A_2(s_1, s_2)$ is continuous in both variables, the Intermediate Value Theorem (applied in $\mathbb{R}^2$) guarantees the existence of a solution $(s_1^\star, s_2^\star)$ satisfying $F(s_1^\star, s_2^\star) = (0, 0)$.

**Step 4: Uniqueness of the solution.** To prove uniqueness, assume two solutions $(s_1', s_2')$ and $(s_1'', s_2'')$. Since $A_1(s_1)$ is strictly decreasing, it follows that:

$$A_1(s_1') = A_1(s_1'') \Rightarrow s_1' = s_1''.$$

Similarly, for a fixed $s_1^\star$, $A_2(s_1^\star, s_2)$ is strictly decreasing in $s_2$, implying that:

$$A_2(s_1^\star, s_2') = A_2(s_1^\star, s_2'') \Rightarrow s_2' = s_2''.$$

Thus, $(s_1^\star, s_2^\star)$ is unique. □

## B.3 Thresholds in the finite-population setting

In the finite model, each candidate $i$ in a group $G_j$, where $j \in \{1, 2\}$, has latent attributes $(v_{i1}, v_{i2})$ drawn independently and uniformly from $[0, 1]^2$. These attributes are transformed into observed utilities $\hat{u}_{ij}$, which determine candidate preferences and institutional choices. A realization $\omega$ corresponds to a specific draw of all $(v_{i1}, v_{i2})$ pairs in the population. Once $\omega$ is fixed, the utilities $\hat{u}_{ij}$ are fixed as well, and the resulting stable matching is deterministic.

In this setting, Proposition B.1 establishes that for every realization $\omega$, the stable matching is characterized by thresholds $S_1^\star(\omega)$ and $S_2^\star(\omega)$, such that Institution 1 selects candidates with $\hat{u}_{i1} \geq S_1^\star(\omega)$ and Institution 2 selects those with $\hat{u}_{i2} \geq S_2^\star(\omega)$.

Assuming candidate attributes are i.i.d., the fraction of candidates in each group satisfying these threshold conditions concentrates around their expectations by standard concentration inequalities (e.g., Chernoff bounds). Consequently, the empirical thresholds $S_1^\star(\omega)$ and $S_2^\star(\omega)$ converge to deterministic limits $s_1^\star$ and $s_2^\star$ as $n \to \infty$, satisfying the equations derived in the infinite-population model.

While our main analysis characterizes the equilibrium thresholds $(s_1^\star, s_2^\star)$ in the continuum limit, it is natural to ask how close the thresholds in a finite population are to their limiting values. The following result shows that these thresholds concentrate sharply around their deterministic counterparts.

| $\gamma$ | Representation Ratio (mean $\pm$ std) | $\gamma$ | Representation Ratio (mean $\pm$ std) |
|---|---|---|---|
| 0.0 | $0.48 \pm 0.16$ | 0.0 | $0.44 \pm 0.10$ |
| 0.1 | $0.44 \pm 0.14$ | 0.1 | $0.45 \pm 0.10$ |
| 0.2 | $0.44 \pm 0.15$ | 0.2 | $0.41 \pm 0.10$ |
| 0.3 | $0.49 \pm 0.16$ | 0.3 | $0.44 \pm 0.06$ |
| 0.4 | $0.53 \pm 0.25$ | 0.4 | $0.53 \pm 0.09$ |
| 0.5 | $0.55 \pm 0.19$ | 0.5 | $0.56 \pm 0.10$ |
| 0.6 | $0.61 \pm 0.19$ | 0.6 | $0.58 \pm 0.10$ |
| 0.7 | $0.64 \pm 0.28$ | 0.7 | $0.58 \pm 0.12$ |
| 0.8 | $0.64 \pm 0.16$ | 0.8 | $0.63 \pm 0.12$ |
| 0.9 | $0.62 \pm 0.17$ | 0.9 | $0.67 \pm 0.16$ |
| 1.0 | $0.73 \pm 0.42$ | 1.0 | $0.63 \pm 0.13$ |

$n = 100$, $c = 0.2$, $\beta = 0.9$, $|G_1| = |G_2| = 100$.    $n = 250$, $c = 0.2$, $\beta = 0.9$, $|G_1| = |G_2| = 250$.

Table 2: Representation ratios for different values of $\gamma$ and finite $n$. The standard deviation decreases as $n$ increases, indicating closer agreement with the continuum predictions.

**Proposition B.3** (**Concentration of finite-population thresholds**). *Given parameters $c$, $\beta$, and $\gamma$, let $(s_1^\star, s_2^\star)$ denote the unique solution to* (11)–(12). *Consider a finite stochastic instance $\mathcal{I}$ with $|G_1| = |G_2| = n$, and two institutions of capacities $cn$ each, bias parameter $\beta$, and correlation parameter $\gamma$. Then, with high probability, there exist thresholds $s_1, s_2$ governing the stable assignment in $\mathcal{I}$ such that*

$$|s_\ell - s_\ell^\star| = O\left(\sqrt{\tfrac{\log n}{n}}\right) \quad \text{for each } \ell \in \{1, 2\}.$$

*Proof.* The proof follows by a standard Chernoff-bound argument. Given $c, \beta, \gamma$, Proposition B.2 ensures a unique solution $(s_1^\star, s_2^\star)$ to (11)–(12). In the finite instance $\mathcal{I}$, random thresholds $s_1, s_2$ define the unique stable assignment as in Proposition B.1.

For each candidate $i \in G_1 \cup G_2$, define the indicator $X_i = \mathbb{1}\{\hat{u}_{i1} \geq s_1^\star - \varepsilon\}$, where $\varepsilon > 0$ is small enough that $s_1^\star - \varepsilon \geq 0$. Let $X = \sum_i X_i$. If $i \in G_1$, then $\mathbb{E}[X_i] = \Pr[u_{i1} \geq s_1^\star] - \varepsilon$, and if $i \in G_2$, then $\mathbb{E}[X_i] \geq \Pr[u_{i1} \geq s_1^\star]$. Hence,

$$\mathbb{E}[X] \geq n(\Pr[\hat{u}_{i1} \geq s_1^\star] + \Pr[\hat{u}_{i'1} \geq s_1^\star]) - n\varepsilon = nc_1 - n\varepsilon,$$

where $i \in G_1$, $i' \in G_2$, and the last equality follows from (11). Let $\mathcal{E}$ denote the event that $X < nc_1$. By the Chernoff bound, $\Pr[\mathcal{E}] \leq e^{-\varepsilon^2 n/4}$. Setting $\varepsilon = O(\sqrt{\tfrac{\log n}{n}})$ gives $\Pr[\mathcal{E}] \leq n^{-a}$ for some constant $a > 0$. If $\mathcal{E}$ does not occur, then $s_1 \geq s_1^\star - \varepsilon$; otherwise, $s_1 < s_1^\star - \varepsilon$ would violate the capacity constraint at Institution 1. A symmetric argument yields $s_1 \leq s_1^\star + \varepsilon$.

An analogous argument for $s_2$ (using expressions for $\Pr[\hat{u}_{i2} \geq s_2 \wedge \hat{u}_{i1} < s_1]$ from Section 3) shows that $|s_2 - s_2^\star| \leq c\varepsilon$ with high probability, for some constant $c$ depending on $c, \beta, \gamma$. This completes the proof. $\qquad\square$

**Finite-sample validation.** To corroborate the theoretical results and illustrate their practical relevance, we implemented our model for finite sample sizes ($n = 100$ and $n = 250$ per group) and conducted 30 independent trials for each value of $\gamma$. The results, summarized in Table 2, show that even for moderately sized systems (100–250 candidates per group), the continuum thresholds provide an excellent approximation to the empirical representation ratios. As expected, the standard deviation of the observed representation ratios decreases with increasing $n$, indicating convergence toward the continuum-limit predictions.

# C Monotonicity of thresholds with respect to $\beta$

In this section, we prove the monotonicity of $s_1^\star$ and $s_2^\star$ with respect to the bias parameter $\beta$. This also implies that the representation ratio is monotone with respect to $\beta$ (Proposition F.1). We restrict the discussion to the symmetric setting $\nu_1 = \nu_2 = 1$ and $c_1 = c_2 = c$. We further prove a useful result showing that the threshold $s_2^\star$ always remains at most $s_1^\star$.

**Proposition C.1 (Monotonicity of $s_1^\star$ and $s_2^\star$ w.r.t. $\beta$).** *For any fixed value of $\gamma$, the thresholds $s_1^\star$ and $s_2^\star$ are non-decreasing functions of the bias parameter $\beta$.*

An easy proof of the monotonicity of $s_1^\star$ with respect to $\beta$ considers the expression for $s_1^\star$. Recall from Section 3 that $s_1^\star = \frac{2-c}{1+1/\beta}$ if $\beta \geq 1 - c$, and $1 - c$ otherwise. This expression is clearly monotone in $\beta$. However, we provide a more intuitive proof that also extends to the monotonicity of $s_2^\star$.

Suppose we increase $\beta$ while keeping $s_1^\star$ fixed. The measure of selected candidates from $G_2$ increases (while that for $G_1$ remains unchanged). Since the capacity of Institution 1 is fixed, $s_1^\star$ must also increase. The argument for the monotonicity of $s_2^\star$ is a more involved version of this idea. As a corollary, we find that the representation ratio is monotone in $\beta$: increasing $\beta$ raises both thresholds, thereby reducing the admission probability for candidates in $G_1$.

*Proof.* Given a value $\beta$ and threshold $s_1$, let $f(\beta, s_1)$ denote $\Pr[\hat{u}_{i1} \geq s_1] + \Pr[\hat{u}_{i'1} \geq s_1]$, where $i \in G_1, i' \in G_2$. Then $f(\beta, s_1)$ is a non-decreasing function of $\beta$ and a decreasing function of $s_1$. Further, Equation (11) implies that $f(\beta, s_1^\star(\beta)) = c$.

Now consider two values $\beta_1 < \beta_2$, and assume for contradiction that $s_1^\star(\beta_1) > s_1^\star(\beta_2)$. Then:

$$c = f(\beta_1, s_1^\star(\beta_1)) \leq f(\beta_2, s_1^\star(\beta_1)) < f(\beta_2, s_1^\star(\beta_2)) = c,$$

which is a contradiction. Therefore, $s_1^\star(\beta_2) \geq s_1^\star(\beta_1)$.

We now prove the monotonicity of $s_2^\star(\beta)$. Rewriting Equation (12), we have:

$$\Pr[\gamma X + (1-\gamma)Y \geq s_2^\star \mid X < s_1^\star]\Pr[X < s_1^\star] + \Pr[\gamma\beta X + \beta(1-\gamma)Y \geq s_2^\star \mid \beta X < s_1^\star]\Pr[\beta X < s_1^\star] = c,$$

where $X, Y$ are independent and uniformly distributed on $[0, 1]$.

Let $Z_\beta$ be a uniform random variable on $[0, s_1^\star]$. Then we can rewrite the equation as:

$$\Pr[\gamma Z_\beta + (1 - \gamma)Y \geq s_2^\star]\Pr[X < s_1^\star] + \Pr[\gamma Z_\beta + \beta(1 - \gamma)Y \geq s_2^\star]\Pr[\beta X < s_1^\star] = c.$$

Let: $A(\beta) := \Pr[\gamma Z_\beta + (1-\gamma)Y \geq s_2^\star]$, $B(\beta) := \Pr[\gamma Z_\beta + \beta(1-\gamma)Y \geq s_2^\star]$, $p(\beta) := \Pr[X < s_1^\star]$, $q(\beta) := \Pr[\beta X < s_1^\star]$.

Suppose $\beta_1 < \beta_2$ and assume $s_2^\star(\beta_1) > s_2^\star(\beta_2)$. Then using monotonicity of $s_1^\star(\beta)$, we verify: (i) $A(\beta_1) < A(\beta_2)$, (ii) $B(\beta_1) < B(\beta_2)$, (iii) $p(\beta_1) < p(\beta_2)$, (iv) $A(\beta) \geq B(\beta)$ for all $\beta \in [0, 1]$.

Since $A(\beta)p(\beta) + B(\beta)q(\beta) = c$, the difference becomes:

$$A(\beta_1)p(\beta_1) + B(\beta_1)q(\beta_1) - A(\beta_2)p(\beta_2) - B(\beta_2)q(\beta_2) = 0.$$

This difference can be rewritten as:

$$A(\beta_1)(p(\beta_1) - p(\beta_2)) + p(\beta_2)(A(\beta_1) - A(\beta_2)) + B(\beta_1)(q(\beta_1) - q(\beta_2)) + q(\beta_2)(B(\beta_1) - B(\beta_2)).$$

From Equation (11), we have $p(\beta) + q(\beta) = c$, so $p(\beta_1) - p(\beta_2) = q(\beta_2) - q(\beta_1)$. Substituting and simplifying, we get:

$$(A(\beta_1) - B(\beta_1))(p(\beta_1) - p(\beta_2)) + p(\beta_2)(A(\beta_1) - A(\beta_2)) + q(\beta_2)(B(\beta_1) - B(\beta_2)) < 0,$$

which contradicts the assumption. Therefore, $s_2^\star(\beta_1) \leq s_2^\star(\beta_2)$. $\qquad\square$

The following result shows that when the two groups have equal size and the capacities at both institutions are equal, the threshold $s_2^\star$ is always at most $s_1^\star$. The proof follows from the observation that if $s_2^\star > s_1^\star$, then a candidate in $G_1$ must satisfy $v_{i2} \geq s_2^\star$ to be assigned to Institution 2, effectively reducing the problem to a single-institution setting, which leads to a contradiction.

**Proposition C.2** ($s_1^\star$ **dominates** $s_2^\star$ **in the symmetric case**). *Suppose $\nu_1 = \nu_2 = \nu$ and $c_1 = c_2 = c\nu$. Then, for any fixed $\beta$, $c$, and $\gamma$, we have $s_2^\star \leq s_1^\star$.*

*Proof.* Let $X$ and $Y$ be i.i.d. uniform random variables on $[0, 1]$. The threshold $s_1^\star$ satisfies:

$$\Pr[X \geq s_1^\star] + \Pr[X \geq s_{1,\beta}^\star] = c,$$

where $s_{1,\beta}^\star := \min(1, s_1^\star/\beta)$.

The threshold $s_2^\star$ satisfies:

$$\Pr[\gamma X + (1-\gamma)Y \geq s_2^\star, X < s_1^\star] + \Pr[\gamma X + (1-\gamma)Y \geq s_{2,\beta}^\star, X < s_{1,\beta}^\star] = c,$$

where $s_{2,\beta}^\star := \min(1, s_2^\star/\beta)$.

Assume for contradiction that $s_2^\star > s_1^\star$. Then:

$$\Pr[Y > s_2^\star] \geq \Pr[\gamma X + (1-\gamma)Y \geq s_2^\star, X < s_1^\star],$$

and similarly for $s_{2,\beta}^\star$. Adding both terms yields:

$$\Pr[Y > s_2^\star] + \Pr[Y > s_{2,\beta}^\star] \geq c.$$

But since $s_1^\star < s_2^\star$, this implies:

$$\Pr[Y > s_1^\star] + \Pr[Y > s_{1,\beta}^\star] > c,$$

contradicting the definition of $s_1^\star$. Therefore, $s_2^\star \leq s_1^\star$. $\qquad\square$

# D    Proofs of results for $\gamma$-thresholds

In this section, we present the proof of Theorem 3.1 that establishes the existence of the $\gamma$-thresholds $\gamma_1$, $\gamma_2$, and $\gamma_3$ when $\beta \geq 1 - c$. By definition, these thresholds characterize the transitions in the evaluation probabilities $\Pr[\gamma X + (1 - \gamma)Y \geq s_2]$ and $\Pr[\gamma X + (1 - \gamma)Y \geq s_2/\beta]$ as $\gamma$ increases from 0 to 1. Specifically, the system passes through the regime transitions: (III,III'), (III,II'), (II,II'), and (IV,IV').

We also describe how to compute the values of $\gamma_1$, $\gamma_2$, and $\gamma_3$, and establish monotonicity properties of these thresholds with respect to the bias parameter $\beta$. We restate the theorem here for convenience.

**Theorem 3.1** ($\gamma$-**thresholds**). *For any fixed $\beta \geq 1 - c$, there exist unique values $\gamma_1 \leq \gamma_2 \leq \gamma_3 \in [0, 1]$ such that: (i) $s_2^\star(\gamma)/\beta \leq 1 - \gamma$ if and only if $\gamma \leq \gamma_1$; (ii) $s_2^\star(\gamma) \leq 1 - \gamma$ if and only if $\gamma \leq \gamma_2$; (iii) $s_2^\star(\gamma) \geq \gamma s_1^\star$ if and only if $\gamma \leq \gamma_3$. Moreover, these thresholds are given by: (i) $\gamma_1 = \frac{2s_1^{\star}(\beta+1/\beta)-4(1-c)}{2s_1^{\star}(\beta+1/\beta)-4(1-c)+(s_1^{\star})^2(1+1/\beta^2)}$; (ii) $\gamma_2 = \frac{x}{x+1}$, where $x$ is the unique root in $(\gamma_1, \gamma_3)$ of $\left(1 + \frac{1}{\beta^2}\right)\frac{x^2(s_1^{\star})^2}{2} + x\left(\frac{s_1^{\star}}{\beta} - \frac{s_1^{\star}}{\beta^2} - c\right) + \frac{1}{2}\left(1 - \frac{1}{\beta}\right)^2 = 0$; (iii) $\gamma_3 = \frac{1}{1+c}$.*

We break the proof into two parts. In the first part, we show the existence of thresholds. Subsequently, we derive expressions for them.

*Proof of existence of $\gamma$-thresholds.* We show that $\frac{s_2^\star(\gamma)}{\gamma}$ is a decreasing function of $\gamma$. Consider values $\gamma_1 < \gamma_2$ where $\gamma_1, \gamma_1 \in (0, 1)$. Our goal is to show that $\frac{s_2^\star(\gamma_2)}{\gamma_2} < \frac{s_2^\star(\gamma_1)}{\gamma_1}$. By definition of $s_2^\star(\gamma_1)$,

$$\Pr[\gamma_1 X + (1 - \gamma_1)Y \geq s_2^\star(\gamma_1), X < s_1^\star] + \Pr[\gamma_1 X + (1 - \gamma_1)Y \geq s_{2,\beta}^\star(\gamma), X < s_{1,\beta}^\star] = c,$$

where $X$ and $Y$ are independent random variables with values in $[0, 1]$. The above condition can be equivalently written as

$$\Pr\left[X + (1/\gamma_1 - 1)Y \geq \frac{s_2^\star(\gamma_1)}{\gamma_1}, X < s_1^\star\right] + \Pr\left[X + (1/\gamma_1 - 1)Y \geq \frac{s_{2,\beta}^\star(\gamma_1)}{\gamma_1}, X < s_{1,\beta}^\star\right] = c.$$

Now observe that $x + (1/\gamma_1 - 1)y > x + (1/\gamma_2 - 1)y$ for any $x, y \in (0, 1)$. Thus, a simple coupling argument shows that

$$\Pr\left[X + (1/\gamma_2 - 1)Y \geq \frac{s_2^\star(\gamma_1)}{\gamma_1}, X < s_1^\star\right] + \Pr\left[X + (1/\gamma_2 - 1)Y \geq \frac{s_{2,\beta}^\star(\gamma_1)}{\gamma_1}, X < s_{1,\beta}^\star\right] < c.$$

Assume for the sake of contradiction that $\frac{s_2^\star(\gamma_1)}{\gamma_1} \leq \frac{s_2^\star(\gamma_2)}{\gamma_2}$. Then, $\frac{s_{2,\beta}^\star(\gamma_1)}{\gamma_1} \leq \frac{s_{2,\beta}^\star(\gamma_2)}{\gamma_2}$ as well. Therefore, the above inequality implies that

$$\Pr\left[X + (1/\gamma_2 - 1)Y \geq \frac{s_2^\star(\gamma_2)}{\gamma_2}, X < s_1^\star\right] + \Pr\left[X + (1/\gamma_2 - 1)Y \geq \frac{s_{2,\beta}^\star(\gamma_2)}{\gamma_2}, X < s_{1,\beta}^\star\right] < c.$$

Rearranging terms, we get

$$\Pr\left[\gamma_2 X + (1 - \gamma_2)Y \geq s_2^\star(\gamma_2), X < s_1^\star\right] + \Pr\left[\gamma_2 X + (1 - \gamma_2)Y \geq s_{2,\beta}^\star(\gamma_2), X < s_{1,\beta}^\star\right] < c.$$

But this contradicts the definition of $s_2^\star(\gamma_2)$. Thus, we see that $\frac{s_2^\star(\gamma)}{\gamma}$ is a decreasing function of $\gamma$. In a similar manner, we can show that $\frac{s_{2,\beta}^\star(\gamma)}{\gamma}$ is a decreasing function of $\gamma$, and $\frac{s_2^\star(\gamma)}{1-\gamma}$ and $\frac{s_{2,\beta}^\star(\gamma)}{1-\gamma}$ are increasing functions of $\gamma$. Since the distribution from which the attributes of a candidate are drawn (in this case, the uniform distribution on $[0,1]$) is continuous, it follows that $s_2^\star(\gamma)$ is a continuous function of $\gamma$.

When $\gamma$ approaches $0$, $\frac{s_{2,\beta}^\star(\gamma)}{1-\gamma}$ approaches $0$ and when $\gamma$ approaches $1$, $\frac{s_{2,\beta}^\star(\gamma)}{1-\gamma}$ approaches $\infty$. Since $\frac{s_{2,\beta}^\star(\gamma)}{1-\gamma}$ is a monotonically increasing continuous function of $\gamma$, there is a unique value $\gamma$, call it $\gamma_1$, such that $\frac{s_{2,\beta}^\star(\gamma_1)}{1-\gamma_1} = 1$. This first statement in the lemma now follows from this observation and the monotonicity of $\frac{s_{2,\beta}^\star(\gamma)}{1-\gamma}$. Other statements in the lemma can be shown similarly. $\square$

*Proof of expressions for $\gamma$-thresholds.* In the regime $\beta \leq 1 - c$, we know that $s_1^\star \leq \beta$. We also know that $s_2^\star \leq s_1^\star$ (Proposition C.2). Thus, $s_1^\star, s_2^\star \leq \beta$ for all $\gamma \in [0,1]$. Hence, $s_{j,\beta}^\star = \frac{s_j^\star}{\beta}$ for $j \in \{1,2\}$. It now follows from the definition of $\gamma_3$ and $\gamma_4$ that $\gamma_3 = \gamma_4$. The monotonicity of $\frac{s_2^\star}{1-\gamma}$ shows that $\gamma_1 < \gamma_2$. We now show that $\gamma_2 \leq \gamma_3$.

We first argue that $\gamma_1 \leq \gamma_3$. Suppose not, i.e., $\gamma_3 < \gamma_1$. For $\gamma \in [0, \gamma_3]$, we are in case $III$ and $III'$. Therefore, $s_2^\star$ satisfies:

$$s_1^\star - \frac{2s_1^\star s_2^\star - \gamma(s_1^\star)^2}{2(1-\gamma)} + \frac{s_1^\star}{\beta} - \frac{2s_1^\star s_2^\star - \gamma(s_1^\star)^2}{2\beta^2(1-\gamma)} = c. \tag{15}$$

By definition of $\gamma_3$, $s_2^\star = \gamma_3 s_1^\star$ when $\gamma = \gamma_3$. Substituting this above, we see that

$$s_1^\star(1 + 1/\beta) - \frac{\gamma_3(s_1^\star)^2}{2(1-\gamma_3)}\left(1 + 1/\beta^2\right) = c.$$

Since $\gamma_3 \leq \gamma_1$, we know by monotonicity of $\frac{s_2^\star(\gamma)}{1-\gamma}$ that $s_1^\star \gamma_3 = s_2^\star \leq \beta(1 - \gamma_3)$. Thus, we get

$$c \geq s_1^\star(1 + 1/\beta) - \frac{\beta s_1^\star}{2}\left(1 + 1/\beta^2\right) = s_1^\star\left(1 - \frac{\beta}{2} + \frac{1}{2\beta}\right) \geq s_1^\star,$$

where the last inequality follows from the fact that $\beta \leq 1$. But we know that $s_1^\star = \frac{2-c}{1+1/\beta} \geq \frac{2-c}{2} > c$, if $c < 1/2$. Thus, we cannot have $\gamma_3 \leq \gamma_1$. Thus $\gamma_1 = \min(\gamma_1, \gamma_2, \gamma_3)$ and hence, (15) holds for all $\gamma \in [0, \gamma_1]$. Substitution $s_2^\star = \beta(1 - \gamma_1)$ for $\gamma = \gamma_1$ in (15) yields the desired expression for $\gamma_1$.

Now we show that $\gamma_2 \leq \gamma_3$. Suppose not, i.e., $\gamma_3 \in (\gamma_1, \gamma_2)$. We are in cases $III$ and $II'$ in $[\gamma_1, \gamma_3]$. Therefore, $s_2^\star$ satisfies:

$$s_1^\star - \frac{2s_1^\star s_2^\star - \gamma(s_1^\star)^2}{2(1-\gamma)} + \frac{(1 - \gamma - s_2^\star/\beta + \gamma s_1^\star/\beta)^2}{2\gamma(1-\gamma)} = c. \tag{16}$$

For $\gamma = \gamma_3$, we know that $s_2^\star = \gamma_3 s_1^\star$. Substituting this above, we get:

$$s_1^\star - \frac{\gamma_3(s_1^\star)^2}{2(1-\gamma_3)} + \frac{1 - \gamma_3}{2\gamma_3} = c.$$

Since $\gamma_3 < \gamma_2$, $s_1^\star \gamma_3 = s_2^\star < (1 - \gamma_3)$. Using this, we see that $c > s_1^\star$, which is a contradiction (as argued in the previous case above). Thus, $\gamma_2 \leq \gamma_3$. The expression for evaluating $\gamma_2$ in the statement of the lemma follows from substituting $s_2^\star = 1 - \gamma_2$ for $\gamma = \gamma_2$ in (16).

Finally, we show the desired expression for $\gamma_3$. In the interval $\gamma \in [\gamma_2, \gamma_3]$, we are in case $II$ and $II'$. Thus, $s_2^\star$ satisfies:

$$(1 - \gamma - s_2^\star + \gamma s_1^\star)^2 + (1 - \gamma - s_2^\star/\beta + \gamma s_1^\star/\beta)^2 = 2c\gamma(1 - \gamma).$$

At $\gamma = \gamma_3$, $s_2^\star = \gamma_3 s_1^\star$. Substituting this above, we get the desired expression for $\gamma_3$. $\qquad\square$

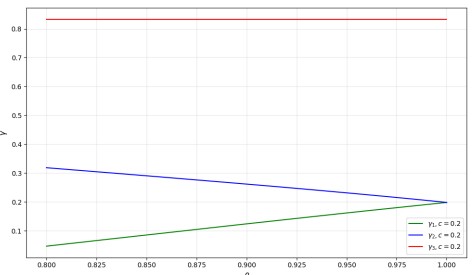 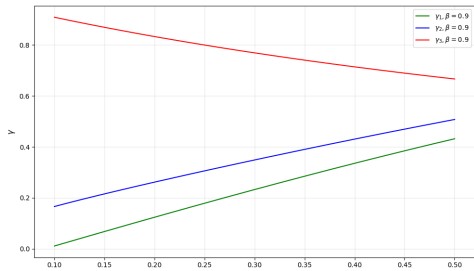

Figure 4: Variation of thresholds $\gamma_1, \ldots, \gamma_3$ with $\beta$ for a fixed value of $c$ (Left) and with $c$ for a fixed value of $\beta$ (Right). Note that $\beta \geq 1 - c$.

We now show monotonicity of $\gamma_1$ and $\gamma_2$ with respect to $\beta$ – observe that $\gamma_3$ is independent of $\beta$.

**Proposition D.1 (Monotonicity of $\gamma$-thresholds w.r.t. $\beta$).** *Assume $c < 1/2$. Then $\gamma_1(\beta)$ is monotonically increasing and $\gamma_2(\beta)$ is a monotonically decreasing function of $\beta$ as $\beta$ varies from $1 - c$ to $1$.*

*Proof.* We first consider $\gamma_1$. Using Theorem 3.1, we know that $\gamma_1$ is equal to

$$\frac{2s_1^\star(\beta + 1/\beta) - 4(1 - c)}{\underbrace{2s_1^\star(\beta + 1/\beta) - 4(1 - c)}_{a(\beta)} + \underbrace{(s_1^\star)^2(1 + 1/\beta^2)}_{b(\beta)}}.$$

The above can be written as $\frac{a(\beta)}{a(\beta) + b(\beta)}$. The sign of the derivative of $\gamma_1$ is given by $a'(\beta)b(\beta) - a(\beta)b'(\beta)$, where $a'(\beta)$ and $b'(\beta)$ denote the derivative of $a(\beta)$ and $b(\beta)$ with respect to $\beta$ respectively. It is easy to verify that $a(\beta) \geq 0$: indeed, this is same as verifying $2s_1^\star(\beta + 1/\beta) \geq 2(1 - c)$. Using $s_1^\star = \frac{2-c}{1+1/\beta}$, this is same as verifying

$$(2 - c)(\beta + 1/\beta) - 2(1 - c)(1 + 1/\beta) \geq 0.$$

Simplifying, the above is the same as verifying

$$2(c + \beta - 1) + c/\beta - c\beta \geq 0,$$

which is true because $\beta + c \geq 1$ and $\beta \leq 1$.

Now recall that the sign of the derivative of $\gamma_1$ with respect to $\beta$ is same as that of $a'(\beta)b(\beta) - a(\beta)b'(\beta)$. Since $a(\beta), b(\beta) \geq 0$, it suffices to show that $a'(\beta) > 0$ and $b'(\beta) < 0$. Using the definition of $s_1^\star$, $a'(\beta)$ is $2(2 - c)$ times

$$\frac{d}{d\beta}\left(\frac{\beta + 1/\beta}{1 + 1/\beta}\right) = \frac{\beta^2 + 2\beta - 1}{(1 + \beta)^2} = \frac{(1 + \beta)^2 - 2}{(1 + \beta)^2} > 0,$$

because $1 + \beta \geq \sqrt{2}$ (recall that $\beta \geq 1 - c \geq 1/2$). Thus, $a'(\beta) > 0$. We consider $b'(\beta)$. Now, $b'(\beta) = 2s_1^\star \frac{ds_1^\star}{d\beta} - 2(s_1^\star)^2/\beta^3$. Thus, $b'(\beta) < 0$ if $\frac{ds_1^\star(\beta)}{d\beta} < \frac{s_1^\star}{\beta^3}$. Using $s_1^\star = \frac{2-c}{1+1/\beta}$, this inequality is equivalent to $1/\beta < 1 + 1/\beta$, which is clearly true. Thus, $\gamma_1(\beta)$ is an increasing function of $\beta$.

We now show the monotonicity of $\gamma_2(\beta)$. Suppose for the sake of contradiction that there are values $\beta < \beta'$ such that $\gamma_2(\beta) \geq \gamma_2(\beta')$. Now, by definition of $\gamma_2$, we know that $\frac{s_2^\star(\gamma_2)}{1-\gamma_2} = 1$. Now, we have the following sequence of inequalities:

$$1 = \frac{s_2^\star(\gamma_2(\beta), \beta))}{1 - \gamma_2(\beta)} \leq \frac{s_2^\star(\gamma_2(\beta'), \beta)}{1 - \gamma_2(\beta')} < \frac{s_2^\star(\gamma_2(\beta'), \beta')}{1 - \gamma_2(\beta')} = 1.$$

Here the first inequality follows from the fact that for a fixed $\beta$, $\frac{s_2^\star(\gamma)}{1-\gamma}$ is an increasing function of $\gamma$, and the second inequality follows from the fact that for a fixed $\gamma$, $s_2^\star(\beta)$ is a monotonically increasing function of $\beta$. This leads to a contradiction, and hence $\gamma_2(\beta)$ is a monotonically decreasing function of $\beta$. $\qquad\square$

# E  Proofs of results for $s_2^\star$

In this section, we present the proof of Theorem 3.2 and analyze the behavior of the selection threshold $s_2^\star$ as $\gamma$ varies. Our approach is to examine the governing equation for $s_2^\star$ in each sub-interval of $[0, 1]$ defined by $\gamma_1, \gamma_2, \gamma_3$, where the selection dynamics transition. Using these equations, we then study the non-monotonic variation of $s_2^\star$ with $\gamma$: as $\gamma$ increases, $s_2^\star$ initially decreases but eventually starts to rise again. The proof of this result relies on analyzing the equations governing $s_2^\star$ from Theorem 3.2. Unless this turns out to be a linear equation, analyzing a closed-form solution for $s_2^\star$ is highly non-trivial. Our approach involves studying the equations satisfied by the first and second derivatives of $s_2^\star$ with respect to $\gamma$. This enables us to infer its behavior across different regions of $\gamma$. We now prove Theorem 3.2. We restate it here for convenience.

**Theorem 3.2 (Equations for $s_2^\star(\gamma)$).** *Assume $\beta \geq 1 - c$ and $c < \frac{1}{2}$. Then $s_2^\star(\gamma)$ satisfies:*

- $[0, \gamma_1]$: $s_1^\star - \frac{2s_1^\star s_2^\star - \gamma(s_1^\star)^2}{2(1-\gamma)} + \frac{s_1^\star}{\beta} - \frac{2s_1^\star s_2^\star - \gamma(s_1^\star)^2}{2\beta^2(1-\gamma)} = c$

- $[\gamma_1, \gamma_2]$: $s_1^\star - \frac{2s_1^\star s_2^\star - \gamma(s_1^\star)^2}{2(1-\gamma)} + \frac{(1-\gamma-s_2^\star/\beta+\gamma s_1^\star/\beta)^2}{2\gamma(1-\gamma)} = c$

- $[\gamma_2, \gamma_3]$: $\frac{(1-\gamma-s_2^\star+\gamma s_1^\star)^2}{2\gamma(1-\gamma)} + \frac{(1-\gamma-s_2^\star/\beta+\gamma s_1^\star/\beta)^2}{2\gamma(1-\gamma)} = c$

- $[\gamma_3, 1]$: $s_1^\star(1 + 1/\beta) - \frac{s_2^\star}{\gamma} - \frac{s_2^\star}{\beta\gamma} + \frac{1-\gamma}{\gamma} = c$

*Proof.* These equations follow by definition of $\gamma_1, \ldots, \gamma_4$ and the corresponding cases described in Section 3. For instance, when $\gamma \in [0, \gamma_1]$, we are in case $III$ and $III'$. Other cases can be argued similarly. $\qquad\square$

We now show that $s_2^\star(\gamma)$ varies in a non-monotone manner, but has only one local minimum.

**Theorem E.1 (Variation of $s_2^\star$ with $\gamma$).** *Assume that the bias parameter $\beta$ satisfies $\beta \geq 1 - c$ and $c < 1/2$. Then, $s_2^\star$ exhibits the following behavior as $\gamma$ varies from 0 to 1:*

  *(i) For $\gamma \in [0, \gamma_1]$, $s_2^\star$ decreases linearly with $\gamma$.*

  *(ii) For $\gamma \in [\gamma_1, \gamma_2]$, $s_2^\star$ is unimodal: it initially decreases and may subsequently increase.*

  *(iii) For $\gamma \in [\gamma_2, \gamma_3]$, $s_2^\star$ is convex.*

  *(iv) For $\gamma \in [\gamma_3, 1]$, $s_2^\star$ is increasing.*

*Furthermore, the slope of $s_2^\star$ at $\gamma_2^-$ matches that at $\gamma_2^+$. Thus, $s_2^\star$ is unimodal.*

*Proof.* **Case $\gamma \in [0, \gamma_1]$:** Here, Theorem 3.2 shows that $s_2^\star$ varies linearly with $\gamma$. In fact, $s_2^\star$ is equal to

$$\frac{1}{2(s_1^\star + s_1^\star/\beta^2)}\left(2s_1^\star(1+1/\beta)(1-\gamma) - 2c(1-\gamma) + \gamma(s_1^\star)^2(1+1/\beta^2)\right).$$

The coefficient of $\gamma$ is equal to $\frac{1}{2s_1^\star(1+1/\beta^2)}$ times

$$2c + (s_1^\star)^2(1+1/\beta^2) - 2s_1^\star(1+1/\beta) = 4c - 4 + \left(\frac{2-c}{1+1/\beta}\right)^2\left(1+\frac{1}{\beta^2}\right) \leq 4c - 4 + (2-c)^2/2.$$

The above is negative if $c < 1/2$. Thus, $s_2^\star$ is a decreasing function of $\gamma$ in $[0, \gamma_1]$.

**Case $\gamma \in [\gamma_1, \gamma_2]$:** We rewrite the constraint in Theorem 3.2 for this case as follows:

$$2\gamma(1-\gamma)(s_1^\star - c) - 2\gamma s_1^\star s_2^\star + \gamma^2(s_1^\star)^2 + (1 - \gamma - s_2^\star/\beta + \gamma s_1^\star/\beta)^2 = 0. \tag{17}$$

Differentiating this, we get (here $(s_2^\star)'$ denotes the derivative of $s_2^\star$ w.r.t. $\gamma$):

$$2(1 - 2\gamma)(s_1^\star - c) - 2s_1^\star s_2^\star + 2\gamma(s_1^\star)^2$$
$$- 2\left(1 - \gamma - \frac{s_2^\star}{\beta} + \frac{\gamma s_1^\star}{\beta}\right)\left(1 - \frac{s_1^\star}{\beta}\right)$$
$$- (s_2^\star)'\left(2\gamma s_1^\star + \frac{2}{\beta}\left(1 - \gamma - \frac{s_2^\star}{\beta} + \frac{\gamma s_1^\star}{\beta}\right)\right) = 0.$$

We first show that $(s_2^\star)' < 0$ at $\gamma = \gamma_1$. For this, we substitute $s_2^\star = \beta(1 - \gamma_1)$ in the equation above. Since the coefficient of $(s_2^\star)'$ is negative, it suffices to show that the remaining terms not involving $(s_2^\star)'$ is negative, i.e., we need to show:

$$(1 - 2\gamma_1)(s_1^\star - c) - 2s_1^\star\beta(1 - \gamma_1) + 2\gamma(s_1^\star)^2 - \frac{2\gamma s_1^\star}{\beta} < 0.$$

The l.h.s. is a linear function of $\gamma_1$. Since $\gamma_1 \geq 0$ and $\beta(1 - \gamma_1) \geq s_1^\star\gamma_1$ (because $\gamma_1 \leq \gamma_3$), it suffices to check the above condition for $\gamma_1 = 0$ and $\gamma_1 = \frac{\beta}{\beta + s_1^\star}$. For $\gamma_1 = 0$, the above condition holds because $s_1^\star \leq \beta$. When $\gamma_1 = \frac{\beta}{\beta + s_1^\star}$, $\gamma_1 \geq 1/2$ because $s_1^\star \leq \beta$. Therefore, $(1 - 2\gamma_1)(s_1^\star - c) < 0$. Using this value of $\gamma_1$, $\beta(1 - \gamma_1) = \gamma s_1^\star$. Thus, the above condition holds here as well. Thus, we have shown that $s_2^\star < 0$ at $\gamma = \gamma_1$. It remains to show that $s_2^\star$ is unimodal. This follows from the following general fact:

**Fact E.2.** *Let $f(\gamma)$ be a twice differentiable function of $\gamma$ on a closed interval $[\gamma_1, \gamma_2]$ such that the following condition holds:*

$$A(\gamma)f''(\gamma) + B(f'(\gamma))^2 + Cf'(\gamma) + D = 0,$$

*where $B, C, D$ are independent of $\gamma$, $C \neq 0$, and $A(\gamma)$ is a continuous function of $\gamma$ and there is a positive constant $E$ such that $A(\gamma) \geq E$. Then $f(\gamma)$ is unimodal on $[\gamma_1, \gamma_2]$, i.e., if $f'(\gamma_a) = f'(\gamma_b) = 0$, then $f'(\gamma) = 0$ for all $\gamma \in [\gamma_a, \gamma_b]$.*

*Proof.* First, consider the case when $D > 0$ (the case $D < 0$ is analogous). Let $u(\gamma)$ denote the $f'(\gamma)$. Assume that there are two distinct values $\gamma_a, \gamma_b \in [\gamma_1, \gamma_2]$, $\gamma_a < \gamma_b$ such that $u(\gamma_a) = u(\gamma_b) = 0$, but $u(\gamma_0) \neq 0$ for some $\gamma_0 \in [\gamma_a, \gamma_b]$. Since $u(\gamma)$ is continuous, we can assume, by suitably selecting $\gamma_a$ and $\gamma_b$, that $u(\gamma)$ is non-negative in $[\gamma_a, \gamma_b]$. By the equation stated in the condition above, we see that there is a constant $\varepsilon > 0$ such that if $|u(\gamma)| < \varepsilon$, then $u'(\gamma) < 0$. By continuity of $u(\gamma)$, there exists a $\delta > 0$ such that for all points $\gamma \in I := [\gamma_a, \min(\gamma_a + \delta, \gamma_0)]$, $0 < u(\gamma) < \varepsilon$. Now, consider a value $\gamma \in I$, where $\gamma \neq \gamma_a$. By mean value theorem, there exists a value $\gamma_c \in [\gamma_a, \gamma]$ such that $u'(\gamma_c) = \frac{u(\gamma) - u(\gamma_a)}{\gamma - \gamma_a} = \frac{u(\gamma)}{\gamma - \gamma_a} > 0$. But this is a contradiction, because $u(\gamma_c) < \varepsilon$ and hence, $u'(\gamma_c)$ must be negative.

Finally, consider the case where $D = 0$. Assume that $C > 0$; the other case can be handled similarly. It follows that there is a constant $\varepsilon$ such that if $0 < u(\gamma) \leq \varepsilon$, then $u'(\gamma) < 0$. The above argument now carries over analogously. $\square$

Let us verify that the conditions stated in the above fact hold in our setting. Differentiating (17) twice and grouping terms, we get (here $(s_2^\star)'$ and $(s_2^\star)''$ denote the derivative and the second derivative of $s_2^\star$ w.r.t. $\gamma$ respectively):

$$\left(\frac{2\Delta}{\beta} + 2s_1^\star\gamma\right)(s_2^\star)'' - \frac{2}{\beta^2}((s_2^\star)')^2 + \left(4s_1^\star - \frac{4}{\beta}(1 - s_1^\star/\beta)\right)s_1^\star - 2(s_1^\star - c) + 2(s_1^\star)^2 + 2(1 - s_1^\star/\beta)^2 = 0.$$

Here $\Delta$ denotes $1 - \gamma + \frac{\gamma s_1^\star}{\beta} - \frac{s_2^\star}{\beta}$. Note that $\Delta \geq 0$ because $s_2^\star \leq s_1^\star$, and hence,

$$\Delta \geq 1 - \gamma + \frac{\gamma s_1^\star}{\beta} - \frac{s_1^\star}{\beta} = (1 - \gamma)(1 - s_1^\star/\beta) \geq 0.$$

We also note that

$$4s_1^\star - \frac{4}{\beta}(1 - s_1^\star/\beta) \neq 0.$$

Indeed, if the l.h.s. is equal to 0, then $\beta s_1^\star = 1 - s_1^\star/\beta$, i.e., $s_1^\star(\beta + 1/\beta) = 1$. Now, $\beta + 1/\beta \geq 2$ and $s_1^\star > 1/2$. Therefore, this cannot happen. Thus, the conditions stated in Fact E.2 hold with $f(\gamma)$ denoting $s_2^\star(\gamma)$, $a(\gamma) = \frac{2\Delta}{\beta} + 2s_1^\star\gamma$, $b = \frac{-2}{\beta^2}$, $c = \left(4s_1^\star - \frac{4}{\beta}(1 - s_1^\star/\beta)\right)$ and $d = -2(s_1^\star - c) + 2(s_1^\star)^2 + 2(1 - s_1^\star/\beta)^2$.

We now claim that $(s_2^\star(\gamma))'$ can be 0 for at most one $\gamma \in [\gamma_1, \gamma_2]$. Indeed, suppose it is 0 at two distinct points $\gamma_a$ and $\gamma_b$, where $\gamma_a < \gamma_b$. Then, Fact E.2 shows that $s_2^\star(\gamma)$ is a constant during the entire interval $[\gamma_a, \gamma_b]$. But we argue that this cannot happen. Indeed, suppose $s_2^\star(\gamma) = K$ during this interval. Substituting this value in (35) and multiplying both sides by $2\gamma(1 - \gamma)$, we see that $\gamma$ satisfies a non-zero quadratic polynomial with constant coefficients. But this can have at most two roots. This is a contradiction because all $\gamma \in [\gamma_a, \gamma_b]$ satisfy this equation. Thus, we see that $s_2^\star(\gamma)$ has at most one local maximum or local minimum in $[\gamma_1, \gamma_2]$. Since $(s_2^\star)' < 0$ at $\gamma_1$, we conclude that $s_2^\star$ can have at most one local minimum and no local maximum in $[\gamma_1, \gamma_2]$.

**Case $\gamma \in [\gamma_2, \gamma_3]$:** We rewrite the equation for this case in Theorem 3.2 as:

$$(1 - \gamma - s_2^\star + \gamma s_1^\star)^2 + (1 - \gamma - s_2^\star/\beta + \gamma s_1^\star/\beta)^2 = 2c\gamma(1 - \gamma).$$

Differentiating the above equation twice with respect to $\gamma$, we get:

$$2(f'(\gamma))^2 + 2f(\gamma)f''(\gamma) + 2(f_\beta'(\gamma))^2 + 2f_\beta(\gamma)f_\beta''(\gamma) = -4c,$$

where $f(\gamma)$ denotes $(1 - \gamma - s_2^\star + \gamma s_1^\star)$ and $f_\beta(\gamma)$ denotes $(1 - \gamma - s_2^\star/\beta + \gamma s_1^\star/\beta)$. Since $s_2^\star \leq s_1^\star$,

$$f(\gamma) = 1 - \gamma - s_2^\star + \gamma s_1^\star \geq 1 - \gamma - s_1^\star + \gamma s_1^\star = (1 - s_1^\star)(1 - \gamma) > 0.$$

Similarly, $f_\beta(\gamma) > 0$. Finally,

$$f''(\gamma) = -(s_2^\star)'', f_\beta''(\gamma) = -(s_2^\star)''/\beta.$$

It follows that $(s_2^\star)'' > 0$ when $\gamma \in (\gamma_2, \gamma_3)$. This shows that $s_2^\star$ is convex.

**Case $\gamma \in [\gamma_3, 1]$:** Solving for $s_2^\star$ in the last case mentioned in Theorem 3.2, we get:

$$s_2^\star = \frac{s_1^\star\gamma(1 + \beta) + \beta(1 - \gamma) - c\beta\gamma}{1 + \beta}.$$

The coefficient of $\gamma$ is equal to $\frac{1}{1+\beta}$ times

$$s_1^\star(1 + \beta) - \beta - c\beta = \beta(2 - c) - \beta - c\beta = \beta - 2c\beta.$$

If $c < 1/2$, we see that this is an increasing function of $\gamma$. Thus, we have shown the desired behavior of $s_2^\star$ in each of the given intervals. It remains to argue that $s_2^\star$ is unimodal. The only case to check is if $s_2^\star$ has a local minimum in both $[\gamma_1, \gamma_2]$ (where it has been shown to be unimodal) and in $[\gamma_2, \gamma_3]$ (where it has been shown to be convex). We can directly check by differentiating the equations for the second and the third cases in Theorem 3.2 that $(s_2^\star)'$ at $\gamma = \gamma_2$ in both the intervals is the same. Thus, if $s_2^\star$ has a local minimum in $[\gamma_1, \gamma_2]$, then it is an increasing function at $\gamma = \gamma_2$. Thus, it will remain an increasing function in $[\gamma_2, \gamma_3]$ (because it is convex in this range).

It remains to check that $(s_2^\star)'$ at $\gamma = \gamma_2$ is the same according to the second and the third equations in Theorem 3.2. We consider the second equation first. Let $g(\gamma)$ denote $\frac{(1 - \gamma - s_2^\star/\beta + \gamma s_1^\star/\beta)^2}{2\gamma(1 - \gamma)}$. Differentiating both sides and substitution $s_2^\star = 1 - \gamma_2$, we get:

$$-\frac{2s_1^\star(s_2^\star)' - (s_1^\star)^2}{2(1 - \gamma_2)} - \frac{2s_1^\star(1 - \gamma_2) - \gamma(s_1^\star)^2}{2(1 - \gamma_2)^2} + g'(\gamma_2) = 0.$$

After simplifying, the above becomes:

$$g'(\gamma_2) + \frac{-2s_1^\star(1 - \gamma_2)(s_2^\star)' + (s_1^\star)^2 - 2s_1^\star(1 - \gamma_2)}{2(1 - \gamma_2)^2} = 0. \tag{18}$$

Similarly, differentiating both sides of the third equation in Theorem 3.2 and substituting $s_2^\star = 1 - \gamma_2$, we get the same equation as above. This shows that $(s_2^\star)'$ at $\gamma = \gamma_2$ according to either of the two equations in Theorem 3.2 is the same. This completes the proof of the theorem.

$\square$

# F   Proofs of results for the representation ratio

In this section, we prove Theorem 3.3 that presents explicit formulas for the representation ratio $\mathcal{R}(\beta, \gamma)$ across the four regimes determined by the threshold values $\gamma_1, \gamma_2, \gamma_3$, under the assumption that $\beta \geq 1 - c$. These expressions are based on the governing equations for the thresholds $s_1^\star$ (Equation (4)) and $s_2^\star$ (Theorem 3.2), and are used to establish both correctness and monotonicity results. Specifically, we analyze how the formula for $\mathcal{R}(\beta, \gamma)$ evolves as $\gamma$ increases, showing that in all four regimes the representation ratio is either constant or strictly increasing in $\gamma$. These results culminate in the proofs of Theorem 3.3 and Corollary 4.1.

In Appendix F.2, we also show that for fixed $\gamma$, $\mathcal{R}(\beta, \gamma)$ is monotone in $\beta$, and prove that the normalized representation ratio $\mathcal{N}(\beta, \gamma) := \mathcal{R}(\beta, \gamma)/\mathcal{R}(\beta, 1)$ is non-decreasing in $\beta$ over the interval $[1 - c, 1]$.

Throughout, we evaluate the representation ratio via

$$\mathcal{R}(\beta, \gamma) = \frac{\Pr[\hat{u}_{i'1} \geq s_1^\star] + \Pr[\hat{u}_{i'1} < s_1^\star,\ \hat{u}_{i'2} \geq s_2^\star]}{\Pr[\hat{u}_{i1} \geq s_1^\star] + \Pr[\hat{u}_{i1} < s_1^\star,\ \hat{u}_{i2} \geq s_2^\star]}.$$

We recall Theorem 3.3 and Corollary 4.1 here.

**Theorem 3.3** (**Representation ratio**). *Fix $c < \frac{1}{2}$ and $\beta \geq 1 - c$.*

*Define* $\quad \Delta(\beta, \gamma) \quad := \quad -\beta s_1^\star \;+\; \sqrt{(s_1^\star)^2(\beta^2 + 1) - \frac{2(1-\gamma)((1-\beta)s_1^\star - c)}{\gamma}}, \quad$ *and* $\quad \theta(\beta, \gamma) \quad :=$

$\frac{-\beta(1-\beta) + \sqrt{-(1-\beta)^2 + \frac{2c\gamma(1+\beta^2)}{1-\gamma}}}{1+\beta^2}$. *Then:*

$$\mathcal{R}(\beta, \gamma) = \begin{cases} \frac{(\beta - (1-c))(\beta + 1 - c) + c^2}{1 - \beta^2(1 - 2c)} & \text{if } \gamma \in [0, \gamma_1], & \frac{1 - \frac{s_1^\star}{\beta} + \frac{\gamma \Delta^2(\beta, \gamma)}{2(1-\gamma)}}{1 - s_1^\star + c - \frac{\gamma \Delta^2(\beta, \gamma)}{2(1-\gamma)}} & \text{if } \gamma \in [\gamma_1, \gamma_2], \\[4mm] \frac{1 - \frac{s_1^\star}{\beta} + \frac{(1-\gamma)\theta^2(\beta, \gamma)}{2\gamma}}{1 - s_1^\star + c - \frac{(1-\gamma)\theta^2(\beta, \gamma)}{2\gamma}} & \text{if } \gamma \in [\gamma_2, \gamma_3], & \frac{-(1-\beta)(1+\gamma) + 4\gamma c}{(1+\gamma)(1-\beta) + 4\gamma\beta c} & \text{if } \gamma \in [\gamma_3, 1]. \end{cases}$$

**Corollary 4.1.** *Fix $c < 1/2$ and $\beta \geq 1 - c$. Then $\mathcal{R}(\beta, \gamma)$ increases monotonically in $\gamma$.*

## F.1   Proof of Theorem 3.3 and Corollary 4.1

We derive closed-form expressions for the representation ratio $\mathcal{R}(\beta, \gamma)$ using the equations governing $s_2^\star$ (Theorem 3.2), and use them to prove Theorem 3.3 and Corollary 4.1. The expressions are defined piecewise over the intervals determined by the thresholds $\gamma_1, \gamma_2, \gamma_3$, and also yield the desired monotonicity in $\gamma$.

**Representation ratio for $\gamma \leq \gamma_1$.**   We know that
$$\Pr[u_{i2} \geq s_2^\star \wedge u_{i1} < s_1^\star] + \Pr[u_{i'2} \geq s_{2,\beta}^\star \wedge u_{i'1} < s_{1,\beta}^\star] = c,$$
and for $\gamma \leq \gamma_1$, we are in Cases I and I'. Hence,
$$\Pr[u_{i2} \geq s_2^\star \wedge u_{i1} < s_1^\star] = s_1^\star - \frac{2 s_1^\star s_2^\star - \gamma(s_1^\star)^2}{2(1-\gamma)},$$
and
$$\Pr[u_{i'2} \geq s_{2,\beta}^\star \wedge u_{i'1} < s_{1,\beta}^\star] = \frac{s_1^\star}{\beta} - \frac{2 s_1^\star s_2^\star - \gamma(s_1^\star)^2}{2\beta^2(1-\gamma)}.$$

For sake of brevity, let $\psi$ denote $\frac{2 s_1^\star s_2^\star - \gamma(s_1^\star)^2}{2(1-\gamma)}$. Then we get:
$$s_1^\star(1 + 1/\beta) - \psi(1 + 1/\beta^2) = c.$$
Using $s_1^\star = \frac{2-c}{1+1/\beta}$, we see that
$$\psi = \frac{2(1-c)}{1 + 1/\beta^2}.$$
Therefore,
$$\mathcal{R}(\beta, \gamma) = \frac{1 - \psi/\beta^2}{1 - \psi} = \frac{\beta^2 - 1 + 2c}{1 - \beta^2 + 2c\beta^2}.$$

**Monotonicity.** It is clear from the expression above that $\mathcal{R}(\beta, \gamma)$ does not vary with $\gamma$.

**Representation ratio for $\gamma_1 \le \gamma \le \gamma_2$.** When $\gamma \in (\gamma_1, \gamma_2)$, Theorem 3.2 shows that:

$$s_1^\star - \frac{2s_1^\star s_2^\star - \gamma(s_1^\star)^2}{2(1-\gamma)} + \frac{(1-\gamma - s_2^\star/\beta + \gamma s_1^\star/\beta)^2}{2\gamma(1-\gamma)} = c. \tag{19}$$

Further, the representation ratio is given by

$$\mathcal{R}(\beta, \gamma) = \frac{1 - s_1^\star/\beta + \frac{(1-\gamma - s_2^\star/\beta + \gamma s_1^\star/\beta)^2}{2\gamma(1-\gamma)}}{1 - \frac{2s_1^\star s_2^\star - \gamma(s_1^\star)^2}{2(1-\gamma)}}. \tag{20}$$

Let $\Delta$ denote $\frac{s_2^\star - \gamma s_1^\star}{1-\gamma}$. Then, Equation (19) can be written as:

$$s_1^\star - \frac{\gamma(s_1^\star)^2}{2(1-\gamma)} - s_1^\star \Delta + \frac{1-\gamma}{2\gamma}\left(1 - \Delta/\beta\right)^2 = c. \tag{21}$$

Let $u$ denote $1 - \Delta/\beta$. Then the above can be written:

$$\frac{1-\gamma}{2\gamma}u^2 + s_1^\star \beta u + (1-\beta)s_1^\star - c - \frac{\gamma(s_1^\star)^2}{2(1-\gamma)} = 0. \tag{22}$$

The positive root of this equation is:

$$u = \frac{-\beta s_1^\star + \sqrt{(\beta s_1^\star)^2 - \frac{2(1-\gamma)((-\beta+1)s_1^\star - c)}{\gamma} + (s_1^\star)^2}}{(1-\gamma)/\gamma}.$$

Observe that the fraction of selected candidates from $G_2$ is

$$1 - s_1^\star/\beta + \frac{1-\gamma}{2\gamma}u^2,$$

which can be written as:

$$1 - s_1^\star/\beta + \frac{\gamma}{2(1-\gamma)}\Delta^2,$$

where

$$\Delta(\beta, \gamma) := -\beta s_1^\star + \sqrt{(s_1^\star)^2(\beta^2 + 1) - \frac{2(1-\gamma)((-\beta+1)s_1^\star - c)}{\gamma}}.$$

**Monotonicity.** In (22), we let $v$ denote $\sqrt{\frac{1-\gamma}{\gamma}}u$. Then this equation can be written as:

$$\frac{v^2}{2} + s_1^\star \beta \sqrt{\frac{\gamma}{1-\gamma}}v + (1-\beta)s_1^\star - c - \frac{\gamma(s_1^\star)^2}{2(1-\gamma)} = 0.$$

Differentiating both sides with respect to $\gamma$, we see that the sign of $v'(\gamma)$ is same as that of

$$-v\beta s_1^\star \sqrt{\frac{1-\gamma}{\gamma}} + (s_1^\star)^2 = -u s_1^\star \beta \frac{1-\gamma}{\gamma} + (s_1^\star)^2.$$

Using the fact that $u = 1 - \Delta/\beta$, the above expression is positive iff $\beta(1-\gamma)(1-\Delta/\beta) \le (s_1^\star)\gamma$, which using the definition of $\Delta$, is same as $s_2^\star \ge \beta(1-\gamma)$. This fact holds true because $\gamma \ge \gamma_1$. Thus we see that $v'(\gamma) > 0$. Since the fraction of selected candidates from $G_2$ is given by $v^2/2 + 1 - s_1^\star/\beta$, we see that as we raise $\gamma$, this fraction increases. Hence $\mathcal{R}(\beta, \gamma)$ increases with increasing $\gamma$.

**Representation ratio for $\gamma_2 \le \gamma \le \gamma_3$.** In this range of $\gamma$, we have the equation:

$$\frac{(1-\gamma - s_2^\star + \gamma s_1^\star)^2}{2\gamma(1-\gamma)} + \frac{(1-\gamma - s_2^\star/\beta + \gamma s_1^\star/\beta)^2}{2\gamma(1-\gamma)} = c.$$

The representation ratio is given by

$$\mathcal{R}(\gamma) = \frac{1 - s_1^\star/\beta + \frac{(1-\gamma - s_2^\star/\beta + \gamma s_1^\star/\beta)^2}{2\gamma(1-\gamma)}}{1 - s_1^\star + \frac{(1-\gamma - s_2^\star + \gamma s_1^\star)^2}{2\gamma(1-\gamma)}}.$$

Let $\Delta$ denote $\frac{s_2^\star - \gamma s_1^\star}{1-\gamma}$. Then the above equation can be written as:

$$(1 - \Delta)^2 + (1 - \Delta/\beta)^2 = \frac{2c\gamma}{1 - \gamma}.$$

Let $u$ denote $1 - \Delta/\beta$. Then, the above equation becomes:

$$u^2 + (1 - \beta + \beta u)^2 - \frac{2c\gamma}{1 - \gamma} = 0.$$

The non-negative root of $u$ is given by:

$$\theta(\beta, \gamma) := \frac{-\beta(1 - \beta) + \sqrt{-(1 - \beta)^2 + \frac{2c\gamma(1 + \beta^2)}{1 - \gamma}}}{1 + \beta^2}.$$

Note that the fraction of selected candidates from $G_2$ is given by

$$1 - s_1^\star/\beta + \frac{(1 - \gamma)\theta^2(\beta, \gamma)}{2\gamma}.$$

**Monotonicity.** We shall show that $\theta(\beta, \gamma) \cdot \sqrt{\frac{1-\gamma}{\gamma}}$ is an increasing function of $\gamma$. This, along with the expression for the fraction of selected candidates from $G_2$, will show that the latter fraction is an increasing function of $\gamma$. Notice that $\theta(\beta, \gamma) \cdot \sqrt{\frac{1-\gamma}{\gamma}}$ is given by

$$(1 + \beta^2)\theta(\beta, \gamma)\sqrt{\frac{1 - \gamma}{\gamma}} = -\beta(1 - \beta)\sqrt{\frac{1 - \gamma}{\gamma}} + \sqrt{-(1 - \beta)^2\sqrt{\frac{1 - \gamma}{\gamma}} + 2c(1 + \beta^2)}.$$

It is easy to check by inspection that the above increases as we increase $\gamma$.

**Representation ratio for $\gamma \geq \gamma_3$.** When $\gamma \geq \gamma_3$,

$$\Pr[u_{i2} \geq s_2^\star \wedge u_{i1} < s_1^\star] = s_1^\star - \frac{s_2^\star}{\gamma} + \frac{1 - \gamma}{2\gamma},$$

and

$$\Pr[u_{i'2} \geq s_{2,\beta}^\star \wedge u_{i'1} < s_{1,\beta}^\star] = s_{1,\beta}^\star - \frac{s_{2,\beta}^\star}{\gamma} + \frac{1 - \gamma}{2\gamma}.$$

Thus, we have the equation:

$$s_1^\star(1 + 1/\beta) - \frac{s_2^\star(1 + 1/\beta)}{\gamma} + \frac{1 - \gamma}{\gamma} = c.$$

Using $s_1^\star = \frac{2 - c}{1 + 1/\beta}$ in the above equation, we get:

$$\frac{s_2^\star}{\gamma} = \frac{\gamma(1 - 2c) + 1}{\gamma(1 + 1/\beta)}.$$

Therefore,

$$\mathcal{R}(\beta, \gamma) = \frac{\frac{1+\gamma}{2\gamma} - \frac{\gamma(1-2c)+1}{\gamma\beta(1+1/\beta)}}{\frac{1+\gamma}{2\gamma} - \frac{\gamma(1-2c)+1}{\gamma(1+1/\beta)}} = \frac{\beta - 1 + \gamma(\beta - 1 + 4c)}{1 - \beta + \gamma(1 - \beta + 4\beta c)}.$$

**Monotonicity.** Differentiating the expression for $\mathcal{R}(\beta, \gamma)$ with respect to $\gamma$, we see that its sign is the same as the sign of

$$(1 - \beta + \gamma(1 - \beta + 4\beta c))(\beta - 1 + 4c) - (\beta - 1 + \gamma(\beta - 1 + 4c))(1 - \beta + 4\beta c),$$

which is the same as $4c(1 - \beta)^2 > 0$. This shows the monotonicity of the representation ratio in this case.

This completes the proofs of Theorem 3.3 and Corollary 4.1.

### F.2 Monotonicity of representation ratios with respect to $\beta$

We now analyze how the representation ratio $\mathcal{R}(\beta, \gamma)$ varies with the bias parameter $\beta$, for fixed $\gamma$. We show that $\mathcal{R}(\beta, \gamma)$ is non-decreasing in $\beta$, and further establish that the normalized representation ratio $\mathcal{N}(\beta, \gamma)$ is also non-decreasing over the interval $\beta \in [1-c, 1]$, under mild assumptions on $c$. The first result follows from the structural monotonicity of the selection thresholds, as shown in Appendix C. The second result relies on a case analysis based on Theorem 3.3.

**Proposition F.1 (Monotonicity of representation ratio w.r.t. $\beta$).** *For any fixed $\gamma$, the representation ratio $\mathcal{R}(\beta, \gamma)$ is a non-decreasing function of the bias parameter $\beta$.*

*Proof.* Adding (11) and (12), we obtain

$$\Pr[(X \geq s_1^\star) \vee (\gamma X + (1-\gamma)Y \geq s_2^\star)] + \Pr[(\beta X \geq s_1^\star) \vee (\gamma \beta X + (1-\gamma)\beta Y \geq s_2^\star)] = 2c.$$

As shown in Proposition C.1, both $s_1^\star$ and $s_2^\star$ are non-decreasing in $\beta$. Therefore, the first term on the left-hand side is non-increasing in $\beta$, implying that the second term must be non-decreasing. Since the representation ratio is the ratio of the second term to the first, it follows that $\mathcal{R}(\beta, \gamma)$ is a non-decreasing function of $\beta$. Formally, since $A(\beta, \gamma) + B(\beta, \gamma) = 2c$ and $s_1^\star, s_2^\star$ are non-decreasing in $\beta$, $A(\beta, \gamma)$ is non-increasing and $B(\beta, \gamma)$ non-decreasing in $\beta$. Hence for $\beta_1 < \beta_2$,

$$\frac{B(\beta_2)}{A(\beta_2)} - \frac{B(\beta_1)}{A(\beta_1)} = \frac{A(\beta_1)\big(B(\beta_2) - B(\beta_1)\big) + B(\beta_1)\big(A(\beta_1) - A(\beta_2)\big)}{A(\beta_1)A(\beta_2)} \geq 0,$$

so $\mathcal{R}(\beta, \gamma)$ is non-decreasing. $\square$

**Theorem F.2 (Monotonicity of normalized representation ratio for $\beta \geq 1-c$).** *For $c \leq 1/2$, if $\beta \geq 1-c$, then for every fixed $\gamma$, the map $\beta \mapsto \mathcal{N}(\beta, \gamma)$ is non-decreasing.*

*Proof.* Our goal is to show that the normalized representation ratio $\mathcal{N}(\beta, \gamma) := \mathcal{R}(\beta, \gamma)/\mathcal{R}(\beta, 1)$ is non-decreasing in $\beta$ over the interval $[1-c, 1]$. Since $c \leq \frac{1}{2}$ and $\beta \in [1-c, 1]$, $B(\beta, \gamma) \leq A(\beta, \gamma)$ and hence $B(\beta, \gamma) \leq c$.

The strategy is as follows: We consider the logarithmic derivative $\partial_\beta \ln \mathcal{N}(\beta, \gamma)$, and aim to show that it is non-negative throughout the interval $\beta \in [1-c, 1]$.

Let $A(\beta, \gamma)$ and $B(\beta, \gamma)$ denote the fractions of candidates selected from groups $G_1$ and $G_2$, respectively. By definition, the representation ratio is given by

$$\mathcal{R}(\beta, \gamma) = \frac{B(\beta, \gamma)}{A(\beta, \gamma)}.$$

To establish the desired monotonicity, it suffices to show that $\partial_\beta B(\beta, \gamma)$ decreases with increasing $\gamma$:

**Fact F.3.** *If $\partial_\beta B(\beta, \gamma) \geq \partial_\beta B(\beta, 1)$, then $\partial_\beta \ln \mathcal{N}(\beta, \gamma) \geq 0$.*

*Proof.* By definition,

$$\mathcal{N}(\beta, \gamma) = \frac{B(\beta, \gamma)}{A(\beta, \gamma)} \cdot \frac{A(\beta, 1)}{B(\beta, 1)}.$$

Observe that for any $\gamma$, $A(\beta, \gamma) + B(\beta, \gamma) = 2c$, the total capacity in the two institutions. Therefore, the above can be expressed as:

$$\mathcal{N}(\beta, \gamma) = \frac{B(\beta, \gamma)}{2c - B(\beta, \gamma)} \cdot \frac{2c - B(\beta, 1)}{B(\beta, 1)}.$$

A direct calculation yields:

$$\partial_\beta \ln \mathcal{N}(\beta, \gamma) = \frac{\partial_\beta B(\beta, \gamma)}{B(\beta, \gamma)} + \frac{\partial_\beta B(\beta, \gamma)}{2c - B(\beta, \gamma)} - \frac{\partial_\beta B(\beta, 1)}{B(\beta, 1)} - \frac{\partial_\beta B(\beta, 1)}{2c - B(\beta, 1)}.$$

Proposition F.1 implies that $\partial_\beta B(\beta, \gamma) \geq 0$ for all $\gamma$. Since $B(\beta, \gamma) + A(\beta, \gamma) = 2c$ and $B(\beta, \gamma) \leq A(\beta, \gamma)$, we see that $B(\beta, \gamma) \leq c$. Therefore, $\frac{1}{2c - B(\beta,\gamma)} > 0$. Assuming $\partial_\beta B(\beta, \gamma) \geq \partial_\beta B(\beta, 1)$, the r.h.s. above is at least $\partial_\beta B(\beta, 1)$ times

$$\frac{1}{B(\beta, \gamma)} + \frac{1}{2c - B(\beta, \gamma)} - \left( \frac{1}{B(\beta, 1)} + \frac{1}{2c - B(\beta, 1)} \right).$$

One can easily check that $\frac{1}{x} + \frac{1}{2c-x}$ is a decreasing function of $x$ when $0 \leq x \leq c$. Therefore, the above quantity is non-negative. Thus, we have shown that $\partial_\beta \ln \mathcal{N}(\beta, \gamma) > 0$. $\qquad \square$

Thus, it is enough to check that $\partial_\beta B(\beta, \gamma) \geq \partial_\beta B(\beta, 1)$ for all $0 \leq \gamma \leq 1$. We now consider the various regimes in which $\gamma$ lies:

**Case 4 ($\gamma \geq \gamma_3 = 1/(1+c)$):** The analysis for this case in Appendix F.1 shows that

$$B(\beta, \gamma) = 1 - \frac{s_{2,\beta}^\star}{\gamma} + \frac{1 - \gamma}{2\gamma} = 1 - \frac{\gamma(1 - 2c) + 1}{\gamma(1 + \beta)} + \frac{1 - \gamma}{2\gamma}.$$

Differentiating with respect to $\beta$, we get:

$$\partial_\beta B(\beta, \gamma) = \frac{(1 - 2c) + 1/\gamma}{(1 + \beta)^2}.$$

Clearly $\partial_\beta B(\beta, \gamma)$ is a decreasing function of $\gamma$ and hence, $\partial_\beta B(\beta, \gamma) \geq \partial_\beta B(\beta, 1)$ when $\gamma \geq \gamma_3$. We also note the following for rest of the proof:

$$\partial_\beta B(\beta, 1) = \frac{2(1 - c)}{(1 + \beta)^2} \tag{23}$$

**Case 1 ($\gamma \leq \gamma_1(\beta)$):** The argument for case 1 in Appendix F.1 shows that

$$B(\beta, \gamma) = 1 - \frac{2(1 - c)}{1 + \beta^2}.$$

Therefore,

$$\partial_\beta B(\beta, \gamma) = \frac{4\beta(1 - c)}{(1 + \beta^2)^2}.$$

Since $\beta \in [1 - c, 1]$ and hence $\beta \geq 1/2$, we verify

$$\frac{4\beta(1 - c)}{(1 + \beta^2)^2} \geq \frac{2(1 - c)}{(1 + \beta)^2},$$

because multiplying by positive denominators gives $2\beta(1 + \beta)^2 \geq (1 + \beta^2)^2$, which holds for $\beta \geq 1/2$. Using (23), we see that $\partial_\beta B(\beta, \gamma) \geq \partial_\beta B(\beta, 1)$ when $\gamma \leq \gamma_1$.

**Case 2 ($\gamma \in (\gamma_1(\beta), \gamma_2(\beta)]$):** Recall the abbreviations

$$s_1^\star(\beta) = \frac{2 - c}{1 + 1/\beta}, \qquad \Delta(\beta, \gamma) = -\beta s_1^\star + \sqrt{(s_1^\star)^2 (\beta^2 + 1) - \frac{2(1 - \gamma)((1 - \beta)s_1^\star - c)}{\gamma}}.$$

The analysis in Appendix F.1 for case 2 shows that

$$B(\beta, \gamma) = 1 - \frac{s_1^\star}{\beta} + \frac{\gamma}{2(1 - \gamma)} \Delta^2.$$

Recall that $s_1^\star = \frac{2-c}{1+1/\beta}$. Using this fact in the equation above and differentiating with respect to $\beta$, we get

$$\partial_\beta B(\beta, \gamma) = \frac{2 - c}{(1 + \beta)^2} + \frac{2\gamma \Delta(\beta, \gamma)}{2(1 - \gamma)} \partial_\beta \Delta(\beta, \gamma). \tag{24}$$

The argument in Appendix F.1 shows that that $\Delta(\beta,\gamma)$ is a positive multiple of $1 - \frac{s_2^\star - \gamma s_1^\star}{\beta(1-\gamma)}$. Since $s_2^\star \leq s_1^\star \leq \beta$, we have

$$\Delta(\beta,\gamma) = 1 - \frac{s_2^\star - \gamma s_1^\star}{\beta(1-\gamma)} \geq 1 - \frac{s_1^\star - \gamma s_1^\star}{\beta(1-\gamma)} = 1 - \frac{s_1^\star}{\beta} \geq 0, \tag{25}$$

and the same argument shows $\theta(\beta,\gamma) \geq 0$, where $\theta(\beta,\gamma)$ is defined in the case $\gamma_2 \leq \gamma \leq \gamma_3$. We show in Lemma F.4 that $\partial_\beta \Delta(\beta,\gamma) \geq 0$. Thus, it follows from (24) that

$$\partial_\beta B(\beta,\gamma) \geq \frac{2-c}{(1+\beta)^2}.$$

Comparing with (23), we see that $\partial_\beta B(\beta,\gamma) \geq \partial_\beta B(\beta,1)$ when $\gamma \in (\gamma_1(\beta),\gamma_2(\beta))$.

**Case 3 ($\gamma \in (\gamma_2(\beta),\gamma_3)$):**  In this regime Theorem 3.3 shows that

$$B(\beta,\gamma) = 1 - \frac{s_1^\star}{\beta} + \frac{1-\gamma}{2\gamma}\theta^2(\beta,\gamma) = 1 - \frac{2-c}{1+\beta} + \frac{1-\gamma}{2\gamma}\theta^2(\beta,\gamma) \tag{26}$$

where

$$\theta(\beta,\gamma) := \frac{-\beta(1-\beta) + \sqrt{-(1-\beta)^2 + \dfrac{2c\gamma(1+\beta^2)}{1-\gamma}}}{1+\beta^2}.$$

Since $\theta(\beta,\gamma) = 1 - \frac{s_2^\star - \gamma s_1^\star}{1-\gamma}$, the argument as in case 2 above shows that $\theta(\beta,\gamma) \geq 0$. We show in Lemma F.4 that $\partial_\beta \theta(\beta,\gamma) \geq 0$. It follows from the expression in equation 26 that $\partial_\beta B(\beta,\gamma) \geq \frac{2-c}{(1+\beta)^2}$. Comparing with (23) that $\partial_\beta B(\beta,\gamma) \geq \partial_\beta B(\beta,1)$. Thus, for all the values of $\gamma$ lying in the range $[0,1]$, $\partial_\beta B(\beta,\gamma) \geq \partial_\beta B(\beta,1)$. Since $B(\beta,\gamma)$ is continuous and piecewise differentiable in $\gamma$, the inequality $\partial_\beta B(\beta,\gamma) \geq \partial_\beta B(\beta,1)$ extends to all boundary values of $\gamma$, completing the proof. Theorem F.2 now follows from Fact F.3. $\qquad\square$

**Lemma F.4 (Monotonicity of $\Delta$ and $\theta$ in $\beta$).** *Fix $c \in \left(0,\frac{1}{2}\right)$ and $\gamma \in (0,1)$. For $\beta \in [1-c, 1]$ define*

$$s_1^\star(\beta) := \frac{2-c}{1+1/\beta}, \quad \Delta(\beta,\gamma) := -\beta s_1^\star(\beta) + \sqrt{s_1^\star(\beta)^2(\beta^2+1) - \frac{2(1-\gamma)\big((1-\beta)s_1^\star(\beta) - c\big)}{\gamma}},$$

$$\theta(\beta,\gamma) := \frac{-\beta(1-\beta) + \sqrt{-(1-\beta)^2 + \dfrac{2c\gamma(1+\beta^2)}{1-\gamma}}}{1+\beta^2}.$$

*Then both $\Delta(\beta,\gamma)$ and $\theta(\beta,\gamma)$ are strictly increasing in $\beta$ on the interval $[1-c, 1]$.*

*Proof.* **1. $s_1^\star(\beta)$ is increasing.** Write $s(\beta) := s_1^\star(\beta) = \frac{(2-c)\beta}{1+\beta}$. A direct derivative gives

$$s'(\beta) = \frac{2-c}{(1+\beta)^2} > 0.$$

Using the above expressions for $s_1^\star(\beta)$ and its derivative, we note that

$$\frac{ds_1^\star(\beta)}{d\beta} = \frac{s_1^\star(\beta)}{\beta + \beta^2}. \tag{27}$$

**2. Monotonicity of $\Delta(\beta,\gamma)$.** Put

$$\Psi(\beta) := s(\beta)^2(\beta^2+1) - \frac{2(1-\gamma)\big((1-\beta)s(\beta) - c\big)}{\gamma}.$$

We show $\Psi'(\beta) > 0$.

*(i) The first term.* Because both $s(\beta)$ and $\beta^2 + 1$ increase, their product $s(\beta)^2(\beta^2 + 1)$ has positive derivative.

*(ii) The second term.* Define $h(\beta) := (1 - \beta)s(\beta) - c$. Since $s(\beta)$ is increasing but multiplied by $(1 - \beta)$, we have $h'(\beta) = -(1 - \beta)s'(\beta) - s(\beta) < 0$ on $[1 - c, 1]$. Hence $-(2(1 - \gamma)/\gamma)\, h(\beta)$ has *positive* derivative. Combining (i) and (ii) yields $\Psi'(\beta) > 0$.

Therefore $\sqrt{\Psi(\beta)}$ is increasing. The linear term $-\beta s(\beta)$ *decreases* (its derivative $-s - \beta s' < 0$), but the growth of $\sqrt{\Psi(\beta)}$ dominates because $\Psi'(\beta) > s^2 + 2\beta ss'$ (Appendix F.2.1). Hence

$$\frac{d}{d\beta}\Delta(\beta, \gamma) = -s(\beta) - \beta s'(\beta) + \frac{\Psi'(\beta)}{2\sqrt{\Psi(\beta)}} > 0.$$

**3. Monotonicity of $\theta(\beta, \gamma)$.** Let

$$\Phi(\beta) := -(1 - \beta)^2 + \frac{2c\gamma(1 + \beta^2)}{1 - \gamma}, \qquad T(\beta) := \sqrt{\Phi(\beta)}.$$

Since $(1 - \beta)^2$ decreases and $(1 + \beta^2)$ increases, $\Phi'(\beta) = 2(\beta + \frac{c\gamma}{1-\gamma}\beta) > 0$, so $T'(\beta) > 0$. Write $\theta(\beta, \gamma) = \frac{g(\beta)}{1 + \beta^2}, g(\beta) := -\beta(1 - \beta) + T(\beta)$. Then

$$g'(\beta) = -(1 - 2\beta) + T'(\beta) > 0 \quad \text{(because } T'(\beta) > 1 - 2\beta \text{ on } [1 - c, 1]).$$

Finally

$$\theta'(\beta) = \frac{(1 + \beta^2)\, g'(\beta) - 2\beta\, g(\beta)}{(1 + \beta^2)^2} > 0,$$

since $g(\beta)$ and $g'(\beta)$ are positive.

Thus, both $\Delta(\beta, \gamma)$ and $\theta(\beta, \gamma)$ are strictly increasing for $\beta \in [1 - c, 1]$. $\qquad\square$

### F.2.1 A lower bound on $\Psi'(\beta)$

For completeness, we record the derivative computation used in Lemma F.4. Recall

$$s(\beta) = \frac{(2 - c)\beta}{1 + \beta}, \qquad \Psi(\beta) = s(\beta)^2(\beta^2 + 1) - \frac{2(1 - \gamma)\big((1 - \beta)s(\beta) - c\big)}{\gamma},$$

with $c < \frac{1}{2}$, $\gamma \in (0, 1)$ and $\beta \in [1 - c, 1]$.

**Step 1: derivative of $s(\beta)$.**
$$s'(\beta) = \frac{2 - c}{(1 + \beta)^2} > 0.$$

**Step 2: derivative of $\Psi$.** Split $\Psi(\beta) = \Psi_1(\beta) - \Psi_2(\beta)$ with $\Psi_1 = s^2(\beta^2 + 1)$, $\Psi_2 = \frac{2(1-\gamma)}{\gamma}\big((1 - \beta)s - c\big)$.

$$\Psi_1'(\beta) = 2ss'(\beta)(\beta^2 + 1) + 2\beta s^2, \qquad \Psi_2'(\beta) = \frac{2(1 - \gamma)}{\gamma}\big(s - \beta s'(\beta)\big) < 0,$$

because the bracketed factor is positive. Hence

$$\Psi'(\beta) = \Psi_1'(\beta) - \Psi_2'(\beta) > 2ss'(\beta)(\beta^2 + 1) + 2\beta s^2.$$

**Step 3: comparison with $s^2 + 2\beta ss'$.** Since $(\beta^2 + 1) \geq 1$ and $s'(\beta) > 0$,

$$2ss'(\beta)(\beta^2 + 1) \geq 2ss'(\beta),$$

so

$$\Psi'(\beta) \geq 2ss'(\beta) + 2\beta s^2 = s^2 + 2\beta ss' + \big[s^2 + 2ss'(\beta) - s^2\big].$$

Because the bracket is non-negative, the inequality follows.

# G   Details of worked example in Section 4

This appendix elaborates on the example presented in Section 4, providing a step-by-step derivation of the representation ratio and minimal interventions using our analytical framework.

**Step 1: Set parameters.**   Fix:
$$\beta_0 = 0.85, \quad \gamma_0 = 0.40, \quad c = 0.20.$$
We begin by computing the threshold for Institution 1:
$$s_1^\star = \frac{2 - c}{1 + 1/\beta_0} = \frac{1.8}{1 + 1/0.85} \approx \frac{1.8}{2.176} \approx 0.8276.$$

**Step 2: Identify regime.**   Using Theorem 3.1, we compute the relevant thresholds:
$$\gamma_1 \approx 0.084, \quad \gamma_2 \approx 0.2904, \quad \gamma_3 = \frac{1}{1 + c} = \frac{1}{1.2} \approx 0.833.$$
Since $\gamma_2 < \gamma_0 = 0.40 < \gamma_3$, this falls into Case 3 of Theorem 3.3.

**Step 3: Compute $\mathcal{R}(\beta_0, \gamma_0)$.**   In Case 3, we define:
$$\theta(\gamma) := \frac{-\beta(1 - \beta) + \sqrt{-(1 - \beta)^2 + \frac{2c\gamma(1+\beta^2)}{1-\gamma}}}{1 + \beta^2}.$$
Substituting $\beta_0 = 0.85$, $c = 0.2$, and $\gamma_0 = 0.40$, we get:
$$\theta(\gamma_0) \approx 0.3096.$$
Then:
$$\mathcal{R}(\beta_0, \gamma_0) = \frac{1 - s_1^\star/\beta_0 + \frac{\theta^2(1-\gamma_0)}{2\gamma_0}}{1 - s_1^\star + c - \frac{\theta^2(1-\gamma_0)}{2\gamma_0}} \approx \frac{0.1044}{0.3170} \approx 0.329.$$

**Step 4: Compute $\mathcal{R}(\beta_0, 1)$ and normalize.**   When $\gamma = 1$, we use Case 4 of Theorem 3.3:
$$\mathcal{R}(\beta_0, 1) = \frac{-2(1 - \beta_0) + 4c}{2(1 - \beta_0) + 4\beta_0 c} = \frac{-2(0.15) + 0.8}{0.3 + 0.68} = \frac{0.5}{0.98} \approx 0.510.$$
Hence, the normalized representation ratio is:
$$\mathcal{N}(\beta_0, \gamma_0) = \frac{\mathcal{R}(\beta_0, \gamma_0)}{\mathcal{R}(\beta_0, 1)} \approx \frac{0.329}{0.510} \approx 0.644.$$

**Step 5: Determine minimal interventions.**   To reach a fairness target of $\tau = 0.80$, we solve for the minimal interventions $(\beta', \gamma')$ that yield $\mathcal{N}(\beta', \gamma_0) \geq 0.80$ or $\mathcal{N}(\beta_0, \gamma') \geq 0.80$.

We find:
$$\text{If } \gamma_0 = 0.40, \quad \beta' \approx 0.911, \qquad \text{If } \beta_0 = 0.85, \quad \gamma' \approx 0.640.$$
Thus, either reducing bias to $\beta' \approx 0.911$ or improving evaluator alignment to $\gamma' \approx 0.640$ suffices to meet the target $\mathcal{N} \geq 0.80$.

# H   Observed utilities

In this section, we consider the observed utility $\mathcal{U}_\ell$ received by each institution $\ell \in \{1, 2\}$, defined as the total utility from the candidates assigned to it. Let $M$ denote the deterministic assignment function that maps each candidate $i$ to an institution $\ell$, based on whether their observed scores cross the respective selection thresholds. Then the observed utility is given by:
$$\mathcal{U}_\ell := \sum_{i \in M^{-1}(\ell)} \hat{u}_{i\ell}.$$

We derive explicit expressions for $\mathcal{U}_1$ and $\mathcal{U}_2$, analyzing how they depend on the parameters $\beta$ and $\gamma$. For Institution 1, the utility can be expressed in closed form using the threshold $s_1^\star$. For Institution 2, the utility depends on the geometry of the selection region defined by the stochastic evaluation rule, and we provide piecewise expressions across different $\gamma$-regimes. Finally, we show that both $\mathcal{U}_1$ and $\mathcal{U}_2$ are non-decreasing functions of $\beta$, complementing our fairness analysis.

### H.1 Observed utility of Institution 1

Given the threshold $s_1^\star$, the (observed) utility derived by Institution 1 from a candidate $i$ is 0 if $\widehat{u}_{i1} < s_1^\star$; $\widehat{u}_{i1}$ otherwise. Therefore, the utility of Institution 1 is given by (here $i \in G_1, i' \in G_2$):

$$\mathcal{U}_1 = \nu_1 \int_0^1 s\,\mathbb{1}[s \geq s_1^\star]d\mu_s + \nu_2 \int_0^1 s\,\mathbb{1}[s \geq s_1^\star]d\mu_s', \tag{28}$$

where $\mu_s$ is the measure induced by the distribution of $\widehat{u}_{i1}$ for $i \in G_1$ (and similarly for $\mu_s'$).

**Theorem H.1** (**Observed utility of Institution 1**). *Assume $\nu_1 = \nu_2 = 1$ and $c_1 = c_2 = 1$. Then the observed utility of Institution 1 is given by*

$$\mathcal{U}_1 = \frac{1 - (s_1^\star)^2}{2} + \beta \cdot \frac{1 - (s_{1,\beta}^\star)^2}{2}.$$

*Proof.* From Equation (28), the utility is computed as the sum of two integrals over the observed scores for candidates in $G_1$ and $G_2$, respectively. The first term is:

$$\int_0^1 x \cdot \mathbb{1}[x \geq s_1^\star]\,dx = \int_{s_1^\star}^1 x\,dx = \left[\frac{x^2}{2}\right]_{s_1^\star}^1 = \frac{1 - (s_1^\star)^2}{2}.$$

For candidates in $G_2$, the observed utility is down-scaled by $\beta$, and the integral becomes:

$$\beta \cdot \int_0^1 x \cdot \mathbb{1}[x \geq s_{1,\beta}^\star]\,dx = \beta \cdot \frac{1 - (s_{1,\beta}^\star)^2}{2}.$$

Summing these two terms yields the result. $\qquad\square$

### H.2 Observed utility of Institution 2

We now turn to the observed utility $\mathcal{U}_2(\gamma)$ derived by Institution 2 from the candidates it selects. Recall that the observed utility from a candidate is zero unless they are selected, in which case it equals their score $\hat{u}_{i2}$. Thus, the total utility can be written as

$$\mathcal{U}_2 := \nu_1 \cdot \mathbb{E}[\hat{u}_{i2} \cdot \mathbb{1}\{\hat{u}_{i1} < s_1^\star, \, \hat{u}_{i2} \geq s_2^\star\}] + \nu_2 \cdot \mathbb{E}[\hat{u}_{i'2} \cdot \mathbb{1}\{\hat{u}_{i'1} < s_{1,\beta}^\star, \, \hat{u}_{i'2} \geq s_{2,\beta}^\star\}],$$

where $i \in G_1$, $i' \in G_2$, and the first term represents candidates from $G_1$ assigned to Institution 2, and the second term represents candidates from $G_2$ assigned to Institution 2.

To evaluate these expectations, we consider the geometry of the selection region under the stochastic evaluation rule. Given $\gamma$, define $A_\gamma$ as the region within the rectangle with corners $(0,0), (s_1^\star, 0), (s_1^\star, 1), (1,0)$ that lies above the line $\gamma x + (1-\gamma)y = s_2^\star(\gamma)$ (as illustrated in Figure 1). Similarly, let $B_\gamma$ denote the corresponding region for group $G_2$, bounded by $(0,0), (s_1^\star/\beta, 0), (s_1^\star/\beta, 1), (1,0)$ and the line $\gamma x + (1-\gamma)y = s_2^\star(\gamma)/\beta$.

Let $(x_\gamma^A, y_\gamma^A)$ and $(x_\gamma^B, y_\gamma^B)$ denote the centroids of $A_\gamma$ and $B_\gamma$, respectively. Then, the total observed utility $\mathcal{U}_2(\gamma)$ can be expressed as:

$$\mathcal{U}_2(\gamma) = \mathrm{Area}(A_\gamma) \cdot (\gamma x_\gamma^A + (1-\gamma)y_\gamma^A) + \beta \cdot \mathrm{Area}(B_\gamma) \cdot (\gamma x_\gamma^B + (1-\gamma)y_\gamma^B).$$

It is useful to observe the following result:

**Lemma H.2.** *Let $T$ be the triangle with vertices $(0,0)$, $\left(0, \frac{s}{1-\gamma}\right)$, and $\left(\frac{s}{\gamma}, 0\right)$, where $s > 0$ and $\gamma \in (0,1)$. Then*

$$\iint_T (\gamma x + (1-\gamma)y)\,dx\,dy = \frac{s^3}{3\,\gamma\,(1-\gamma)}.$$

*Proof.* Because $\gamma x + (1 - \gamma)y$ is an *affine* (degree-one) function, its integral over a triangle equals its value at the triangle's centroid multiplied by the triangle's area.

*1. Centroid.* The centroid $C_T = (\bar{x}, \bar{y})$ of a triangle with vertices $(0,0)$, $\left(0, \frac{s}{1-\gamma}\right)$, $\left(\frac{s}{\gamma}, 0\right)$ is the average of the vertices:

$$\bar{x} = \frac{1}{3}\frac{s}{\gamma}, \qquad \bar{y} = \frac{1}{3}\frac{s}{1-\gamma}.$$

Evaluating the integrand at the centroid gives

$$f(C_T) = \gamma\bar{x} + (1-\gamma)\bar{y} = \gamma\left(\frac{s}{3\gamma}\right) + (1-\gamma)\left(\frac{s}{3(1-\gamma)}\right) = \frac{s}{3} + \frac{s}{3} = \frac{2s}{3}.$$

*2. Area.* The base of the triangle is $\frac{s}{\gamma}$ and its height is $\frac{s}{1-\gamma}$, so

$$\text{Area}(T) = \frac{1}{2} \cdot \frac{s}{\gamma} \cdot \frac{s}{1-\gamma} = \frac{s^2}{2\gamma(1-\gamma)}.$$

*3. Integral.* Multiplying the centroid value by the area,

$$\iint_T \left(\gamma x + (1-\gamma)y\right) dx\, dy = f(C_T)\,\text{Area}(T) = \frac{2s}{3} \cdot \frac{s^2}{2\gamma(1-\gamma)} = \frac{s^3}{3\gamma(1-\gamma)}. \qquad \square$$

We now use Lemma H.2 to evaluate $\mathcal{U}_2$ for various cases.

**Case I ($s_2^\star \leq \min(s_1^\star\gamma, 1-\gamma)$):** In this case, the line intersects $y$-axis at $\frac{s_2^\star}{1-\gamma} \leq 1$ and the $x$-axis at $\frac{s_2^\star}{\gamma} \leq s_1^\star$. The integral of the utility over the entire rectangle $[0, s_1^\star] \times [0, 1]$ can be given as follows: area of the rectangle is $s_1^\star$ and its centroid is at $\left(\frac{s_1^\star}{2}, \frac{1}{2}\right)$. Therefore, the integral of $\gamma x + (1-\gamma)y$ over the entire rectangle is

$$s_1^\star \cdot \left(\frac{\gamma s_1^\star}{2} + \frac{1-\gamma}{2}\right).$$

Subtracting out the integral of the utility over the triangle formed by $(0,0)$, $\left(0, \frac{s_2^\star}{1-\gamma}\right)$, $\left(\frac{s_2^\star}{\gamma}, 0\right)$, we see that the observed utility (for Institution 2) of the selected candidates from $G_2$ is

$$\frac{s_1^\star(\gamma s_1^\star + 1 - \gamma)}{2} - \frac{(s_2^\star)^3}{3\gamma(1-\gamma)}.$$

**Case II ($s_2^\star \geq \max(s_1^\star\gamma, 1-\gamma)$):** In this case, the line intersects the $y$-axis at $\frac{s_2^\star}{1-\gamma} > 1$ and the $x$-axis at $\frac{s_2^\star}{\gamma} > s_1^\star$. Thus, its intersection point with the line $y = 1$ is at $\frac{s_2^\star - (1-\gamma)}{\gamma}$ and its intersection point with the line $x = s_1^\star$ is at $\frac{s_2^\star - \gamma s_1^\star}{1-\gamma}$. The centroid of the triangle is

$$\frac{1}{3}\left(\frac{s_2^\star - (1-\gamma) + 2\gamma s_1^\star}{\gamma}, \frac{2 - 2\gamma + s_2^\star - \gamma s_1^\star}{1-\gamma}\right).$$

Evaluated on the line $\gamma x + (1-\gamma)y$, we get the value $\frac{1}{3}(1 - \gamma + 2s_2^\star + \gamma s_1^\star)$. Thus, the utility of institute 2 from $G_1$ is:

$$\frac{1}{3}(1 - \gamma + 2s_2^\star + \gamma s_1^\star) \cdot \frac{(1 - \gamma - s_2^\star + \gamma s_1^\star)^2}{2\gamma(1-\gamma)}.$$

**Case III ($s_1^\star\gamma < s_2^\star < 1-\gamma$):** In this case, the line intersects $y$-axis at $\frac{s_2^\star}{1-\gamma} \leq 1$ and the line $x = s_1^\star$ at $\frac{s_2^\star - \gamma s_1^\star}{1-\gamma}$. We break the area into two parts, the upper rectangle whose centroid is $\frac{1}{2}\left(s_1^\star, \frac{1-\gamma+s_2^\star}{1-\gamma}\right)$. The area of this rectangle is $s_1^\star \cdot \left(1 - \frac{s_2^\star}{1-\gamma}\right)$. The centroid of the lower triangle is $\frac{1}{3}\left(2s_1^\star, \frac{3s_2^\star - \gamma s_1^\star}{1-\gamma}\right)$. The area of this triangle is $\frac{s_1^\star}{2} \cdot \frac{\gamma s_1^\star}{1-\gamma}$. Thus, the total utility is

$$s_1^\star \cdot \left(1 - \frac{s_2^\star}{1-\gamma}\right)\left(\frac{\gamma s_1^\star}{2} + \frac{1-\gamma+s_2^\star}{2}\right) + \frac{s_1^\star}{2} \cdot \frac{\gamma s_1^\star}{1-\gamma} \cdot \left(\frac{2\gamma s_1^\star}{3} + \frac{3s_2^\star - \gamma s_1^\star}{3}\right).$$

**Case IV ($1 - \gamma < s_2^\star < s_1^\star \gamma$):** In this case, the line intersects the $y$-axis at $\frac{s_2^\star}{1-\gamma} > 1$ and the $x$-axis at $\frac{s_2^\star}{\gamma} \leq s_1^\star$.

We break the area into two parts, the right rectangle whose centroid is $\frac{1}{2}\left(s_1^\star + \frac{s_2^\star}{\gamma}, 1\right)$. The area of this rectangle is $s_1^\star - \frac{s_2^\star}{\gamma}$. The centroid of the left triangle is $\frac{1}{3}\left(\frac{3s_2^\star - (1-\gamma)}{\gamma}, 2\right)$. The area of this triangle is $\frac{1}{2} \cdot \frac{1-\gamma}{\gamma}$. Thus, the total utility is

$$\left(s_1^\star - \frac{s_2^\star}{\gamma}\right) \cdot \left(\frac{\gamma s_1^\star + s_2^\star}{2} + \frac{1-\gamma}{2}\right) + \frac{(1-\gamma)}{2\gamma} \cdot \left(\frac{3s_2^\star - (1-\gamma)}{3} + \frac{2(1-\gamma)}{3}\right).$$

The above quantities can be evaluated for $G_2$ in an analogous manner (by replacing $s_1^\star$ and $s_2^\star$ by $s_{1,\beta}^\star$ and $s_{2,\beta}^\star$ respectively, and multiplying the resulting expression by $\beta$). We can now write down the total (observed) utility of Institution 2.

**Theorem H.3** (**Characterizing $\mathcal{U}_2(\gamma)$**). *Assume that $\beta \geq 1 - c$ and $c < 1/2$. Then, $\mathcal{U}_2(\gamma)$ is:*

- *For $\gamma \in [0, \gamma_1]$:* $s_1^\star \cdot \left(1 - \frac{s_2^\star}{1-\gamma}\right)\left(\frac{\gamma s_1^\star}{2} + \frac{1-\gamma+s_2^\star}{2}\right) + \frac{s_1^\star}{2} \cdot \frac{\gamma s_1^\star}{1-\gamma} \cdot \left(\frac{2\gamma s_1^\star}{3} + \frac{3s_2^\star - \gamma s_1^\star}{3}\right)$
  $+ s_1^\star \cdot \left(1 - \frac{s_2^\star}{\beta(1-\gamma)}\right)\left(\frac{\gamma s_1^\star}{2\beta} + \frac{1-\gamma+s_2^\star/\beta}{2}\right) + \frac{s_1^\star}{2\beta} \cdot \frac{\gamma s_1^\star}{1-\gamma} \cdot \left(\frac{2\gamma s_1^\star}{3\beta} + \frac{3s_2^\star - \gamma s_1^\star}{3\beta}\right).$

- *For $\gamma \in [\gamma_1, \gamma_2]$:* $s_1^\star \cdot \left(1 - \frac{s_2^\star}{1-\gamma}\right)\left(\frac{\gamma s_1^\star}{2} + \frac{1-\gamma+s_2^\star}{2}\right) + \frac{s_1^\star}{2} \cdot \frac{\gamma s_1^\star}{1-\gamma} \cdot \left(\frac{2\gamma s_1^\star}{3} + \frac{3s_2^\star - \gamma s_1^\star}{3}\right) + \frac{\beta}{3}(1 - \gamma + 2s_2^\star/\beta + \gamma s_1^\star/\beta) \cdot \frac{(1-\gamma-s_2^\star/\beta+\gamma s_1^\star/\beta)^2}{2\gamma(1-\gamma)}.$

- *For $\gamma \in [\gamma_2, \gamma_3]$:* $\frac{1}{3}(1 - \gamma + 2s_2^\star + \gamma s_1^\star) \cdot \frac{(1-\gamma-s_2^\star+\gamma s_1^\star)^2}{2\gamma(1-\gamma)} + \frac{\beta}{3}(1 - \gamma + 2s_2^\star/\beta + \gamma s_1^\star/\beta) \cdot \frac{(1-\gamma-s_2^\star/\beta+\gamma s_1^\star/\beta)^2}{2\gamma(1-\gamma)}.$

- *For $\gamma \in [\gamma_3, 1]$:* $\left(s_1^\star - \frac{s_2^\star}{\gamma}\right) \cdot \left(\frac{\gamma s_1^\star + s_2^\star}{2} + \frac{1-\gamma}{2}\right) + \frac{(1-\gamma)}{2\gamma} \cdot \left(\frac{3s_2^\star - (1-\gamma)}{3} + \frac{2(1-\gamma)}{3}\right)$
  $+ \left(s_1^\star - \frac{s_2^\star}{\gamma}\right) \cdot \left(\frac{\gamma s_1^\star/\beta + s_2^\star/\beta}{2} + \frac{1-\gamma}{2}\right) + \frac{\beta(1-\gamma)}{2\gamma} \cdot \left(\frac{3s_2^\star/\beta - (1-\gamma)}{3} + \frac{2(1-\gamma)}{3}\right).$

*Proof.* When $\gamma \in (0, \gamma_1)$, we are in cases III and III'. When $\gamma \in [\gamma_1, \gamma_2)$, we are in cases III and II'. When $\gamma \in [\gamma_2, \gamma_3)$, we are in cases II and II'. Finally, for $\gamma \in [\gamma_3, 1)$, we are in cases IV and IV'. $\square$

Plugging in the values of $s_1^\star$ and $s_2^\star$ into the utility expressions, one can carry out an analysis analogous to that for $s_2^\star$ and the representation ratio. We omit the details and instead present a plot in Figure 6, which shows how $\mathcal{U}_2$ varies with $\gamma$. We observe that the behavior of $\mathcal{U}_2$ mirrors that of $s_2^\star$, which is expected since increasing the selection threshold typically results in a higher utility.

Specifically, $\mathcal{U}_2$ is larger for smaller values of $\gamma$. An intuitive explanation is that the area Area($A_\gamma$)—which corresponds to the fraction of $G_2$ candidates selected by Institution 2—is a decreasing function of $\gamma$. Likewise, the threshold $s_2^\star$ increases as $\gamma$ decreases. This stands in contrast to the behavior of the representation ratio $\mathcal{R}$, which increases with $\gamma$. Thus, increasing $\gamma$ induces two opposing effects: it improves representational fairness (as measured by $\mathcal{R}$) but decreases the total observed utility of Institution 2 (as measured by $\mathcal{U}_2$).

### H.3 Monotonicity of utilities with respect to $\beta$

We now show that the observed utility of each institution is a non-decreasing function of the bias parameter $\beta$, for fixed $\gamma$; see also Figure 5.

**Proposition H.4** (**Monotonicity of $\mathcal{U}_1$**). *For any fixed $\gamma$, the observed utility of Institution 1 is a non-decreasing function of the bias parameter $\beta$.*

*Proof.* For simplicity, we assume $c_1 = c_2 = c$; the argument generalizes to arbitrary capacities. From Theorem H.1, the utility of Institution 1 is given by

$$\mathcal{U}_1 = \frac{1 - (s_1^\star)^2}{2} + \beta \cdot \frac{1 - (s_{1,\beta}^\star)^2}{2}.$$

Differentiating with respect to $\beta$, we obtain:

$$\frac{d\mathcal{U}_1}{d\beta} = \frac{1 - (s_{1,\beta}^\star)^2}{2} - s_1^\star \cdot \frac{ds_1^\star}{d\beta} - \beta s_{1,\beta}^\star \cdot \frac{ds_{1,\beta}^\star}{d\beta}.$$

It suffices to show that this derivative is non-negative. From the threshold constraint (see Equation (11)),

$$(1 - s_1^\star) + (1 - s_{1,\beta}^\star) = c,$$

and hence,

$$\frac{ds_1^\star}{d\beta} + \frac{ds_{1,\beta}^\star}{d\beta} = 0.$$

Using this, the sum of the last two terms becomes

$$-s_1^\star \cdot \frac{ds_1^\star}{d\beta} - s_{1,\beta}^\star \cdot \frac{ds_{1,\beta}^\star}{d\beta} = (s_{1,\beta}^\star - s_1^\star) \cdot \frac{ds_1^\star}{d\beta}.$$

This expression is non-negative because $s_{1,\beta}^\star \geq s_1^\star$ and $\frac{ds_1^\star}{d\beta} \geq 0$ by Proposition C.1. Therefore, $\mathcal{U}_1$ is non-decreasing in $\beta$. $\qquad\square$

**Proposition H.5 (Monotonicity of $\mathcal{U}_2$).** *For any fixed $\gamma$, the observed utility of Institution 2 is a non-decreasing function of the bias parameter $\beta$.*

*Proof.* Let $U_j$ denote the utility contribution to Institution 2 from a candidate in group $G_j$. Since the expectation of a non-negative random variable $Z$ can be written as $\mathbb{E}[Z] = \int_0^1 \Pr[Z \geq u]\,du$, we have:

$$\mathbb{E}[U_j] = \int_0^{s_2^\star} \Pr[U_j \geq s_2^\star]\,du + \int_{s_2^\star}^1 \Pr[U_j \geq u]\,du$$

$$= s_2^\star \cdot \Pr[U_j \geq s_2^\star] + \int_{s_2^\star}^1 \Pr[U_j \geq u]\,du,$$

since $\Pr[U_j \geq u] = \Pr[U_j \geq s_2^\star]$ for all $0 < u \leq s_2^\star$.

Differentiating with respect to $\beta$, and applying the Leibniz rule, we obtain:

$$\frac{d}{d\beta}\mathbb{E}[U_j] = \frac{ds_2^\star}{d\beta} \cdot \Pr[U_j \geq s_2^\star] + s_2^\star \cdot \frac{d}{d\beta}\Pr[U_j \geq s_2^\star] + \int_{s_2^\star}^1 \frac{d}{d\beta}\Pr[U_j \geq u]\,du - \frac{ds_2^\star}{d\beta} \cdot \Pr[U_j \geq s_2^\star].$$

The first and last terms cancel, leaving:

$$\frac{d}{d\beta}\mathbb{E}[U_j] = s_2^\star \cdot \frac{d}{d\beta}\Pr[U_j \geq s_2^\star] + \int_{s_2^\star}^1 \frac{d}{d\beta}\Pr[U_j \geq u]\,du.$$

Since an increase in $\beta$ increases the effective score of group $G_2$ candidates and thus the probability that they are selected, both terms on the right-hand side are non-negative. Hence, the expected utility $\mathbb{E}[U_j]$ — and therefore $\mathcal{U}_2$ — is a non-decreasing function of $\beta$. $\qquad\square$

## H.4 Variation of $\mathcal{U}_2$ with respect to $\gamma$

We investigate how the observed utilities of institutions vary with respect to the correlation parameter $\gamma$. As shown in Figure 6, the utility of Institution 2 exhibits a non-monotonic relationship with $\gamma$, displaying a U-shaped pattern across all tested values of $\beta$. This arises due to competing effects: higher $\gamma$ increases alignment between institutional rankings, reducing the chance of cross-institution gains, while also increasing selectivity.

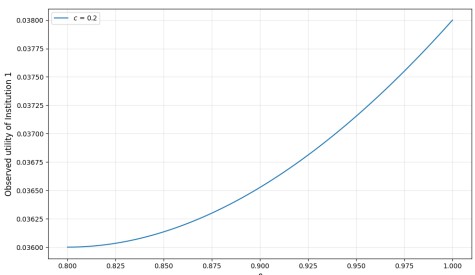 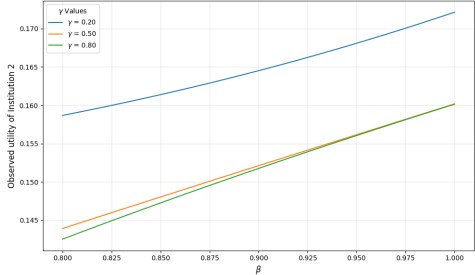

Figure 5: (Left) Utility of Institution 1 as a function of $\beta$ for fixed $\gamma = 0.5$ and $c = 0.2$. (Right) Utility of Institution 2 as a function of $\beta$ for fixed $c = 0.2$ and varying $\gamma \in \{0.2, 0.5, 0.8\}$.

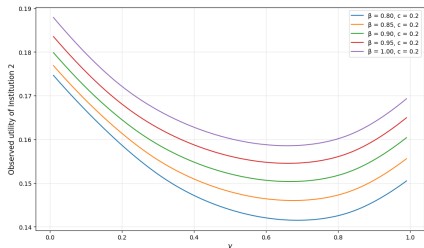

Figure 6: Utility of Institution 2 as a function of $\gamma$ for various $\beta$ at fixed $c = 0.2$, showing non-monotonic behavior.

# I  Extension to distinct bias parameters for two institutions

Our main analysis assumes a common bias parameter $\beta$ applied uniformly across both institutions. In practice, however, evaluation standards may differ—one institution may adopt more rigorous fairness safeguards or use a less biased scoring process than the other. To capture this asymmetry, we extend the model to allow distinct bias parameters $(\beta_{12}, \beta_{22})$ with respect to Institutions 1 and 2 for the candidates in group $G_2$ respectively. Analogous to the $\beta \geq 1 - c$ assumption, we assume that both the parameters $\beta_{12}, \beta_{22}$ are at least $1 - c$. For the sake of concreteness, we also assume $\beta_{22} \geq \beta_{12}$. In Section I.1, we show that the thresholds and representation ratio can be computed numerically. In Section I.2, we demonstrate numerically that the qualitative properties of these metrics remain unchanged, and we further verify this analytically in an extreme parameter regime.

## I.1  Computing thresholds and representation ratio

The expression for $s_1^\star$ remains same as in (4) with $\beta$ replaced by $\beta_{12}$:

$$s_1^\star = \frac{2-c}{1+1/\beta}. \tag{29}$$

We now outline the steps for computing $s_2^\star$ and $\mathcal{R}(\gamma)$. Given values $s_1, s_2, \gamma$, let $P(s_1, s_2, \gamma)$ denote $\Pr[\gamma v_{i1} + (1 - \gamma)v_{i2} \geq s_2 \wedge u_{i1} < s_1]$. This can be evaluated using the four cases mentioned in Section 3.

Since $\beta_{12} \geq 1 - c$, it follows that $s_1^\star \leq \beta_{12}$. However, unlike Proposition C.2, it may not happen that $s_2^\star \leq \beta_{22}$. We first check if $s_2^\star \leq \beta_{22}$. For this, we evaluate $P(s_1^\star, \beta_{22}, \gamma)$. If $P(s_1^\star, \beta_{22}, \gamma) \geq c$, then we know that $s_2^\star$ has to be at least $\beta_{22}$ – otherwise the fraction of candidates assigned to institution 2 would exceed its capacity. Similarly, if $P(s_1^\star, \beta_{22}, \gamma) < c$, then $s_2^\star \leq \beta_{22}$.

**Case $s_2^\star > \beta_{22}$:**  In this case, all the candidates selected by institution 2 are from $G_1$. Therefore, $s_2^\star$ is obtained by solving the equation:

$$P(s_1^\star, s_2^\star, \gamma) = c.$$

Further, $\mathcal{R}(\gamma) = \frac{1 - s_1^\star / \beta_{12}}{c + (1 - s_1^\star)}$.

**Case $s_2^\star \leq \beta_{22}$:** To evaluate $s_2^\star$, we need to solve for the following equation:

$$P(s_1^\star, s_2^\star, \gamma) + P(s_1^\star/\beta_{12}, s_2^\star/\beta_{22}, \gamma) = c.$$

Note that there will be four cases for each of the terms on the l.h.s. above. More precisely, while evaluating $P(s_1^\star, s_2^\star, \gamma)$, we will have to *guess* one of the following cases:

   (i) $s_2 \leq \min(s_1\gamma, 1 - \gamma)$
   (ii) $s_2 \geq \max(s_1\gamma, 1 - \gamma)$
   (iii) $s_1\gamma < s_2 < 1 - \gamma$
   (iv) $1 - \gamma < s_2 < s_1\gamma$.

Similarly, while evaluating $P(s_1^\star/\beta_{12}, s_2^\star/\beta_{22}, \gamma)$, we have to guess one of the following four cases. Let $s_{1,\beta}^\star$ denote $s_1^\star/\beta_{12}$ and $s_{2,\beta}^\star$ denote $s_2^\star/\beta_{22}$

   (i)' $s_{2,\beta} \leq \min(s_{1,\beta}\gamma, 1 - \gamma)$
   (ii)' $s_{2,\beta} \geq \max(s_{1,\beta}\gamma, 1 - \gamma)$
   (iii)' $s_{1,\beta}\gamma < s_{2,\beta} < 1 - \gamma$
   (iv)' $1 - \gamma < s_{2,\beta} < s_{1,\beta}\gamma$.

For each combination of 16 possibilities, we solve for $s_2^\star$ and then take the solution that is consistent with the corresponding guess. After solving for $s_2^\star$, the representation ratio is given by:

$$\frac{1 - s_{1,\beta}^\star + P(s_{1,\beta}^\star, s_{2,\beta}^\star, \gamma)}{1 - s_1^\star + P(s_1^\star, s_2^\star, \gamma)}.$$

## I.2 Numerical evaluation

Numerical experiments in this setting indicate that, across a wide range of asymmetric configurations, $s_2^\star$ is unimodal in $\gamma$, while $\mathcal{R}(\gamma)$ increases consistently. For example, the plots in Figure 7 show this behavior for different values of $\beta_{12}$ and $\beta_{22}$.

To illustrate this phenomenon analytically, we show that even in an extreme asymmetric setting, the representation ratio $\mathcal{R}(\gamma)$ is strictly higher when the institutions are fully aligned ($\gamma = 1$) than when they evaluate candidates independently ($\gamma = 0$).

**Proposition I.1.** *Consider a setting where $1 > \beta_{12} > 1 - c$ and $\beta_{22} = 1$. Then, $\mathcal{R}(\gamma) = 1$ when $\gamma = 1$, and $\mathcal{R}(\gamma) = \frac{2c + \beta_{12} - 1}{2c\beta_{12} - \beta_{12} + 1} < 1$ when $\gamma = 0$.*

*Proof.* When $\gamma = 1$, the two institutions are fully aligned in their evaluations. A candidate $i$ is selected if either $\hat{u}_{i1} \geq s_1^\star$ or $\hat{u}_{i2} \geq s_2^\star$. We first claim that $s_2^\star \leq s_1^\star$. Suppose not. The fraction of candidates from $G_1$ admitted to Institution 1 is $1 - s_1^\star$, which must be at most $c$ by the capacity constraint. If $s_2^\star > s_1^\star$, then no candidate from $G_1$ would be assigned to Institution 1. Moreover, the fraction of candidates from $G_2$ admitted to Institution 2 is less than $1 - s_1^\star \leq c$, violating the capacity condition at Institution 2. Hence, $s_2^\star \leq s_1^\star$. Consequently, any candidate with $\hat{u}_{i2} \geq s_2^\star$ is selected. Since $\beta_{22} = 1$, an equal fraction of candidates from both groups satisfy $\hat{u}_{i2} \geq s_2^\star$, implying that $\mathcal{R}(\gamma) = 1$.

We now consider the case $\gamma = 0$. In this regime, the institutions evaluate candidates independently. Institution 1 admits candidates based on the threshold $s_1^\star = \frac{2-c}{1+1/\beta_{12}}$. The fraction of candidates from $G_1$ not admitted to Institution 1 is $s_1^\star$, while that for $G_2$ is $s_1^\star/\beta_{12}$. Since $\beta_{22} = 1$ and the attribute $v_{i2}$ is i.i.d. across groups, the fraction of candidates from $G_1$ admitted to Institution 2 is $\frac{cs_1^\star}{s_1^\star + s_1^\star/\beta_{12}}$, and the corresponding fraction for $G_2$ is $\frac{cs_1^\star/\beta_{12}}{s_1^\star + s_1^\star/\beta_{12}}$. Hence, the overall representation ratio is given by

$$\frac{\frac{cs_1^\star/\beta_{12}}{s_1^\star + s_1^\star/\beta_{12}} + (1 - s_1^\star/\beta_{12})}{\frac{cs_1^\star}{s_1^\star + s_1^\star/\beta_{12}} + (1 - s_1^\star)}.$$

Substituting the expression for $s_1^\star$ from above and simplifying yields the desired result. $\square$

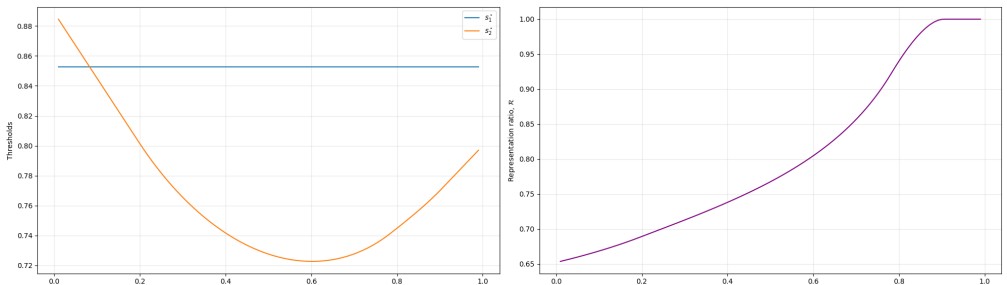

Figure 7: Variation of thresholds $s_1^\star$ and $s_2^\star$ and representation ratio $\mathcal{R}$ with $\gamma$ when $c = 0.2, \beta_{12} = 0.9, \beta_{22} = 1$.

## J  Thresholds and representation ratio computation for the setting of additive bias

We now consider the additive-bias model in which the observed utility of a candidate $i \in G_2$ for institution $j$ is $\hat{u}_{ij} = u_{ij} - \beta$, where $u_{ij}$ is the true utility and $\beta \in [0, 1]$ is an additive bias parameter (observe that, unlike the multiplicative bias parameter setting, smaller values of $\beta$ imply less bias). Under the natural assumption $\beta \leq c$ — the analogue of the multiplicative regime $\beta \geq 1 - c$ that ensures Institution 1 continues to admit some candidates from $G_2$ — the admission threshold $s_1^\star$ at Institution 1 satisfies the equation:

$$1 - s_1^\star + (1 - s_1^\star - \beta) = c.$$

Thus, we get

$$s_1^\star = 1 - \frac{c + \beta}{2}. \tag{30}$$

The geometric structure of the selection regions remains unchanged up to a translation by $\beta$, and the $\gamma$-threshold framework developed in Section 3 continues to apply. Thus, the expression for $\Pr[\hat{u}_{i1} < s_1, \hat{u}_{i2} \geq s_2]$ in the four cases $I, \ldots, IV$ corresponding to group $G_1$ remain unchanged. Similarly, we have the corresponding cases $I', \ldots. IV'$ for $G_2$, where we redefine $s_{1,\beta}^\star = \min(1, s_1^\star + \beta)$ and $s_{2,\beta}^\star = \min(1, s_2^\star + \beta)$.

Similarly, the proof of Proposition C.2 showing $s_2^\star \leq s_1^\star$ extends directly. We now establish the existence of the $\gamma$ thresholds. As in Theorem 3.1, we show that there is a constant $c_0 < 1$ such that if the available capacity $c \leq c_0$, the relative ordering of the $\gamma$ thresholds is fixed.

**Theorem J.1.** *Assume $c < c_0 = 0.36$, and $\beta \leq c$. There exist thresholds $0 \leq \gamma_1 \leq \gamma_2 \leq \gamma_3 \leq \gamma_4 \leq 1$ such that:*

*(1)* $s_2^\star + \beta \leq 1 - \gamma \iff \gamma \leq \gamma_1$,

*(2)* $s_2^\star \leq 1 - \gamma \iff \gamma \leq \gamma_2$,

*(3)* $s_2^\star \geq \gamma s_1^\star \iff \gamma \leq \gamma_3$,

*(4)* $s_2^\star + \beta \geq \gamma (s_1^\star + \beta) \iff \gamma \leq \gamma_4$.

*Proof.* We first show the existence of the thresholds $\gamma_1, \ldots, \gamma_4$. Arguing as in the proof of Theorem 3.1, we can show that $\frac{s_2^\star(\gamma)}{\gamma}$ is a decreasing function of $\gamma$ and $\frac{s_2^\star(\gamma)}{1-\gamma}$ is an increasing function of $\gamma$: the argument follows verbatim with $s_{1,\beta}^\star$ and $s_{2,\beta}^\star$ denoting $\min(1, s_1^\star + \beta)$ and $\min(1, s_2^\star + \beta)$ respectively. Since $\beta \leq c$, $s_1^\star \leq 1 - \beta$. Therefore, $s_{1,\beta}^\star = s_1^\star + \beta$. Since $s_2^\star \leq s_1^\star$, it follows that $s_{2,\beta}^\star = s_2^\star + \beta$ as well.

Since $\frac{s_2^\star(\gamma)}{1-\gamma}$ is an increasing function of $\gamma$, $\frac{s_2^\star(\gamma)+\beta}{1-\gamma}$ is also an increasing function of $\gamma$. When $\gamma = 0$, this ratio is $s_2^\star(\gamma) + \beta \leq s_1^\star + \beta \leq 1$; and when $\gamma$ approaches 1, this ratio approaches infinity. Thus,

there is a unique value $\gamma_1 \in [0,1]$ such that $\frac{s_2^\star(\gamma_1)+\beta}{1-\gamma_1} = 1$. Further, $\frac{s_2^\star(\gamma)+\beta}{1-\gamma} < 1$ iff $\gamma < \gamma_1$. The existence of the thresholds $\gamma_2, \ldots, \gamma_4$ can be shown similarly.

At $\gamma = \gamma_1$, $\frac{s_2^\star(\gamma_1)}{1-\gamma_1} = 1 - \frac{\beta}{1-\gamma_1} < 1$. Therefore, the definition of $\gamma_2$ implies that $\gamma_1 \leq \gamma_2$. Similarly, $\gamma_3 \leq \gamma_4$. It remains to show that $\gamma_2 \leq \gamma_3$. The proof proceeds in two steps: (i) we first show that $\gamma_2 \notin [\gamma_4, 1]$, and (ii) then we show that $\gamma_2 \notin (\gamma_3, \gamma_4)$. This will prove that $\gamma_2 \leq \gamma_3$, and hence, $\gamma_1 \leq \gamma_2 \leq \gamma_3 \leq \gamma_4$.

**Proving $\gamma_2 \notin [\gamma_4, 1]$.** Assume for the sake of contradiction that $\gamma_2 \in [\gamma_4, 1]$. Then, at $\gamma = \gamma_2$, $s_2^\star(\gamma) = 1 - \gamma$ and $s_2^\star(\gamma) \leq \gamma s_1^\star(\gamma)$ (because $\gamma_2 \geq \gamma_4 \geq \gamma_3$). Thus we are in case IV for $G_1$. Similarly, we are in case IV' for $G_2$. Therefore, constraint (2) implies that

$$s_1^\star - \frac{s_2^\star}{\gamma} + \frac{1-\gamma}{2\gamma} + s_1^\star + \beta - \frac{s_2^\star + \beta}{\gamma} + \frac{1-\gamma}{2\gamma} = c.$$

Substituting $s_2^\star = 1 - \gamma$ above, we get

$$2s_1^\star - (1+\beta)\left(\frac{1-\gamma}{\gamma}\right) = c.$$

When $\gamma = \gamma_2$, $1 - \gamma = s_2^\star(\gamma) \leq \gamma s_1^\star$. Thus, $\frac{1-\gamma}{\gamma} \leq s_1^\star$. Using this fact and (30), we get:

$$\left(1 - \frac{c+\beta}{2}\right)(1-\beta) \leq c.$$

The l.h.s. above is a decreasing function of $\beta$. Since $\beta \leq c$, we get

$$(1-c)^2 \leq c.$$

But this contradicts the fact that $c < c_0 = 0.36$. Therefore, $\gamma_2 \notin [\gamma_4, 1]$.

**Proving $\gamma_2 \notin (\gamma_3, \gamma_4)$.** Assume for the sake of contradiction that $\gamma_2 \in (\gamma_3, \gamma_4)$. Then we are in case IV and II' when $\gamma = \gamma_2$. Thus, constraint (2) implies

$$s_1^\star - \frac{s_2^\star}{\gamma} + \frac{1-\gamma}{2\gamma} + \frac{(1 - \gamma - s_2^\star - \beta + (s_1^\star + \beta)\gamma))^2}{2\gamma(1-\gamma)} = c.$$

Substituting $s_2^\star = 1 - \gamma$ above, we get:

$$s_1^\star - \frac{1-\gamma}{2\gamma} + \frac{((s_1^\star + \beta)\gamma - \beta)^2}{2\gamma(1-\gamma)} = c.$$

At $\gamma = \gamma_2$, $\frac{1-\gamma}{\gamma} \leq s_1^\star$. Therefore,

$$\frac{s_1^\star}{2} + \frac{((s_1^\star + \beta)\gamma) - \beta)^2}{2\gamma(1-\gamma)} \leq c.$$

Now observe that $(s_1^\star + \beta)\gamma - \beta \geq (s_2^\star(\gamma) + \beta)\gamma - \beta = (1 - \gamma + \beta)\gamma - \beta = (1-\beta)(1-\gamma) > 0$. Keeping $c$ fixed, as we raise $\beta$, $s_1^\star$ decreases. Thus, the l.h.s. above is a decreasing function of $\beta$. Since $\beta \leq c$, and $s_1^\star = 1 - c$ when $\beta = c$, we get:

$$\frac{1-c}{2} + \frac{(\gamma - c)^2}{2\gamma(1-\gamma)} \leq c.$$

Simplifying the above, we get:

$$3c\gamma^2 - \gamma(5c + 1) + c^2 \leq 0.$$

When $\gamma = \gamma_2$, $1 - \gamma \leq s_1^\star\gamma \leq \gamma$. This implies that $\gamma \geq 1/2$. It is easy to verify that the l.h.s. above is an increasing function of $\gamma$ when $\gamma \in [1/2, 1]$. Indeed, the derivative w.r.t. $\gamma$ of the l.h.s. above is $6c\gamma - 5c + 1 \geq 3c - 5c + 1 = 1 - 2c > 0$ because $c < 1/2$. Therefore, substituting $\gamma = 1/2$ above, we get

$$c^2 - \frac{7c}{4} + \frac{1}{2} \leq 0,$$

which is not possible if $c < 0.36$. Thus, we see that $\gamma_2 \notin (\gamma_3, \gamma_4)$. This shows that $\gamma_2 \leq \gamma_3$, and completes the proof of the theorem. $\qquad\square$

Using these regime-characterizations together with the probability computations from Section 3 for $\Pr[\gamma v_{i1} + (1-\gamma)v_{i2} \geq s_2 \,\wedge\, v_{i1} < s_1]$, we obtain a piecewise set of equations that determine $s_2^\star$ under additive bias:

**Theorem J.2** (**Equations for $s_2^\star(\gamma)$**). *Assume $\beta \leq c$ and $c < c_0 = 0.36$. Then $s_2^\star(\gamma)$ satisfies:*

- $[0, \gamma_1]$: $s_1^\star - \frac{2s_1^\star s_2^\star - \gamma(s_1^\star)^2}{2(1-\gamma)} + s_1^\star + \beta - \frac{2(s_1^\star + \beta)(s_2^\star + \beta) - \gamma(s_1^\star + \beta)^2}{2(1-\gamma)} = c$

- $[\gamma_1, \gamma_2]$: $s_1^\star - \frac{2s_1^\star s_2^\star - \gamma(s_1^\star)^2}{2(1-\gamma)} + \frac{(1 - \gamma - (s_2^\star + \beta) + \gamma(s_1^\star + \beta))^2}{2\gamma(1-\gamma)} = c$

- $[\gamma_2, \gamma_3]$: $\frac{(1 - \gamma - s_2^\star + \gamma s_1^\star)^2}{2\gamma(1-\gamma)} + \frac{(1 - \gamma - (s_2^\star + \beta) + \gamma(s_1^\star + \beta))^2}{2\gamma(1-\gamma)} = c$

- $[\gamma_3, \gamma_4]$: $s_1^\star - \frac{s_2^\star}{\gamma} + \frac{1-\gamma}{2\gamma} + \frac{(1 - \gamma - (s_2^\star + \beta) + \gamma(s_1^\star + \beta))^2}{2\gamma(1-\gamma)} = c$

- $[\gamma_4, 1]$: $2s_1^\star + \beta - \frac{2s_2^\star + \beta}{\gamma} + \frac{1-\gamma}{\gamma} = c$

*Proof.* The proof follows along the same lines as that of Theorem 3.2. For example, when $\gamma \leq \gamma_1$, Theorem J.1 shows that we are in cases III and III'. Therefore, for a candidate $i \in G_1$,

$$\Pr[\widehat{u}_{i1} < s_1^\star, \widehat{u}_{i2} \geq s_2^\star] = \frac{s_1^\star}{2}\left(2 - \frac{2s_2^\star - \gamma s_1^\star}{1 - \gamma}\right).$$

Similarly, for a candidate $i \in G_2$,

$$\Pr[\widehat{u}_{i1} < s_1^\star, \widehat{u}_{i2} \geq s_2^\star] = \frac{s_1^\star + \beta}{2}\left(2 - \frac{2(s_2^\star + \beta) - \gamma(s_1^\star + \beta)}{1 - \gamma}\right).$$

The equation for $s_2^\star(\gamma)$ now follows from (2). Other cases can be handled similarly. $\qquad\square$

We can use the equations in Theorem J.2 to solve for the threshold $s_2^\star$. Once $s_1^\star$ and $s_2^\star$ are determined, expression (3) yields the representation ratio $\mathcal{R}$. We omit the details.

These results show that the geometric and analytical structure underlying the multiplicative-bias model extends naturally to the additive setting, providing a unified framework for quantifying how evaluator bias influences equilibrium thresholds and representation outcomes.

# K  The setting where $\beta < 1 - c$

In Section 3, we analyzed the setting where the bias parameter satisfies $\beta \geq 1 - c$. This condition ensures that Institution 1 admits at least some candidates from the disadvantaged group $G_2$, and the analysis led to a clean three-phase structure characterized by the thresholds $\gamma_1, \gamma_2, \gamma_3$ with a fixed ordering.

In this section, we depart from that setting and study the regime where $\beta < 1 - c$. This regime is qualitatively distinct and exhibits complex structural behavior in the equilibrium thresholds and selection dynamics. To analyze this, we again use a finite-to-infinite reduction framework analogous to Appendix B, deriving mean-field equations for thresholds and examining their qualitative behavior as a function of the correlation parameter $\gamma$.

Our main findings in the $\beta < 1 - c$ regime are summarized above and contrasted with the $\beta \geq 1 - c$ case in Table 3.

- The number of $\gamma$-thresholds increases from three to four: $\gamma_1, \gamma_2, \gamma_3, \gamma_4$. Moreover, unlike the $\beta \geq 1 - c$ case, the ordering of these thresholds is no longer fixed, but instead depends sensitively on the value of $\beta$.

- We establish the existence of a critical value $\beta_c$ such that for $\beta < \beta_c$, the thresholds follow the order $\gamma_1 < \gamma_3 < \gamma_2 < \gamma_4$, whereas for $\beta > \beta_c$, the order is $\gamma_1 < \gamma_2 < \gamma_3 < \gamma_4$.

- The threshold $s_2^\star(\gamma)$ may no longer be unimodal. Unlike the earlier regime where $s_2^\star(\gamma)$ had a single minimum, we show that it may have multiple local minima and maxima. This non-monotonicity also manifests in the representation ratio $\mathcal{R}(\beta, \gamma)$, which need not be monotonic in $\gamma$.

| Structural property | Result for $\beta < 1 - c$ | Result for $\beta \geq 1 - c$ |
|---|---|---|
| Closed-form expression for $s_1^\star$ | $s_1^\star = 1 - c$ (Eq. (31)) | $s_1^\star = \frac{2-c}{1+1/\beta}$ (Eq. (4)) |
| Monotonicity of $s_1^\star$ w.r.t. $\beta$ | Independent of $\beta$ (Eq. (31)) | Increasing in $\beta$ (follows from Eq. (4)) |
| Regimes of $\gamma$ | $\gamma_1 \leq \gamma_2 \leq \gamma_3 \leq \gamma_4$ or $\gamma_1 \leq \gamma_3 \leq \gamma_2 \leq \gamma_4$ (Theorem K.1, Proposition K.3) | $\gamma_1 \leq \gamma_2 \leq \gamma_3$ (Theorem 3.1) |
| Piecewise expression for $s_2^\star$ | Characterized via $\gamma_1, \ldots, \gamma_4$ (Theorem K.5) | Characterized via $\gamma_1, \gamma_2, \gamma_3$ (Theorem 3.2) |
| Variation of $s_2^\star$ w.r.t. $\gamma$ | Continuous, possibly with multiple local extrema (Theorem K.6) | Continuous, with at most one minimum (Theorem E.1) |
| Monotonicity of $s_2^\star$ w.r.t. $\beta$ | Increasing in $\beta$ (Proposition C.1) | Increasing in $\beta$ (Proposition C.1) |
| Monotonicity of $\mathcal{R}$ w.r.t. $\beta$ | Increasing in $\beta$ for fixed $\gamma$ (Proposition F.1) | Increasing in $\beta$ for fixed $\gamma$ (Proposition F.1) |
| Monotonicity of $\mathcal{R}$ w.r.t. $\gamma$ | Non-monotonic in $\gamma$ for fixed $\beta$ (Proposition K.8) | Non-decreasing in $\gamma$ for fixed $\beta$ (Corollary 4.1) |

Table 3: Comparison of structural properties of thresholds and representation metrics between the regimes $\beta < 1 - c$ and $\beta \geq 1 - c$.

This section is organized as follows:

- In Appendix K.2, we generalize the notion of $\gamma$-thresholds and formally state the structural result as Theorem K.1 and Proposition K.2. We also define and characterize the critical value $\beta_c$ in Proposition K.3 and provide lower bounds on its value in Proposition K.4.

- In Appendix K.3, we characterize the piecewise structure of $s_2^\star(\gamma)$ using Theorem K.5, and show how it varies across the threshold intervals. We also prove in Theorem K.6 that $s_2^\star(\gamma)$ exhibits multiple local extrema. Finally, we demonstrate in Proposition K.8 that the representation ratio $\mathcal{R}(\gamma)$ can decrease with increasing $\gamma$, in contrast to the monotonic increase observed in the $\beta \geq 1 - c$ regime.

## K.1   Expression for $s_1^\star$

When $\beta < 1 - c$, then $s_1^\star \geq \beta$. Indeed, otherwise the fraction of candidates from $G_1$ selected by institution 1 would be at least $1 - s_1^\star \geq 1 - \beta \geq c$, which is a contradiction. Thus, $s_1^\star \geq \beta$, and hence, $s_{1,\beta}^\star = 1$. This implies that no candidate from $G_2$ gets selected. Thus, we have the following equation for institution 1: $1 - s_1^\star = c$, which implies

$$s_1^\star = 1 - c. \tag{31}$$

## K.2   Existence of $\gamma$-thresholds

We first show the existence of $\gamma$ thresholds.

**Theorem K.1 (Existence of four $\gamma$-thresholds).** *Assume $c < 1/2$ and $\beta > \frac{1-2c}{1-c}$. Then, there exist unique values $\gamma_1, \gamma_2, \gamma_3, \gamma_4 \in [0, 1]$ such that for any $\gamma \in [0, 1]$, the following hold:*

*(i) $s_{2,\beta}^\star(\gamma) \leq (1 - \gamma)$ if and only if $\gamma \leq \gamma_1$.*

*(ii) $s_2^\star(\gamma) \leq (1 - \gamma)$ if and only if $\gamma \leq \gamma_2$.*

*(iii) $s_2^\star(\gamma) \geq \gamma s_1^\star$ if and only if $\gamma \leq \gamma_3$.*

*(iv) $s_{2,\beta}^\star(\gamma) \geq \gamma s_{1,\beta}^\star$ if and only if $\gamma \leq \gamma_4$.*

*In case $1 - 2c \leq \beta < \frac{1-2c}{1-c}$, the thresholds $\gamma_2, \ldots, \gamma_4$ exist with the above mentioned properties. When $\beta < 1 - 2c$, the thresholds $\gamma_2, \gamma_3$ with the above-mentioned properties exist.*

*Proof.* Monotonicity of $\frac{s_{2,\beta}^\star}{1-\gamma}, \frac{s_2^\star}{\gamma}, \frac{s_{2,\beta}^\star}{\gamma}$ and $\frac{s_{2,\beta}^\star}{\gamma}$ follows from the same arguments as in the proof of Theorem 3.1. However, we need to now show that these ratios achieve the desired values.

When $\gamma = 0$, $\frac{s_{2,\beta}^\star(0)}{1-\gamma} = s_{2,\beta}^\star(0) = \frac{s_2^\star(0)}{\beta} \leq 1$ if $\gamma \geq \frac{1-2c}{1-c}$. As $\gamma$ approaches 1, $\frac{s_{2,\beta}^\star(0)}{1-\gamma}$ approaches infinity. Therefore, intermediate value theorem and monotonicity of $\frac{s_{2,\beta}^\star}{1-\gamma}$ shows that this ratio equals

1 for a unique $\gamma$, which we denote $\gamma_1$. In case $\beta < \frac{1-2c}{1-c}$, $s_{2,\beta}^\star(0) = 1$, and hence the ratio $\frac{s_{2,\beta}^\star}{1-\gamma}$ stays above 1 for all $\gamma \in [0, 1]$.

Since $s_2^\star(0) \leq 1$, the ratio $\frac{s_2^\star}{1-\gamma}$ is at most 1 at $\gamma = 0$. Therefore, $\gamma_2$ exists for the entire range of $\beta$. Similarly, the ratio $\frac{s_2^\star}{\gamma}$ approaches infinity as $\gamma$ approaches 0, and is equal to $s_2^\star(1) \leq s_1^\star$ (using Proposition C.2) when $\gamma = 1$. Thus, the threshold $\gamma_3$ also exists for the entire range of $\beta$.

When $\beta \geq 1 - c$, the condition for $\gamma_3$ is the same as that for $\gamma_4$. Hence, assume $\beta < 1 - c$. In this case, $s_{1,\beta}^\star = 1$. If $\beta \geq 1 - 2c$, $s_2^\star(1) \leq \beta$. Thus, the condition for $\gamma_4$ becomes: $s_2^\star(\gamma) \geq \beta\gamma$. Since $\frac{s_2^\star}{\gamma} \leq \beta$ at $\gamma = 1$ and approaches infinity at $\gamma = 0$, existence of $\gamma_4$ follows. Observe that when $\beta < 1 - 2c$, the only value of $\gamma$ satisfying (iv) is $\gamma = 1$, and hence, $\gamma_4$ does not reveal any useful information. $\qquad\square$

For sake of completeness, we shall set $\gamma_1$ to 0 when $\beta < \frac{1-2c}{1-c}$ and $\gamma_4$ to 1 when $\beta < 1 - 2c$. We now show that the only possible orderings of these thresholds are (i) $\gamma_1, \gamma_2, \gamma_3, \gamma_4$, or (ii) $\gamma_1, \gamma_3, \gamma_2, \gamma_4$. Monotonicity properties of $\frac{s_2^\star(\gamma)}{\gamma}$ and $\frac{s_2^\star(\gamma)}{1-\gamma}$ show that $\gamma_1 \leq \gamma_2$ and $\gamma_3 \leq \gamma_4$. We now show that $\gamma_1$ is the smallest and $\gamma_4$ is the largest among $\gamma_1, \dots, \gamma_4$.

**Proposition K.2 (Extremal ordering of thresholds).** *Assume $c < 1/2$ and $\beta < 1 - c$. Then $\gamma_1 \leq \gamma_j$ for all $j \in \{1, \dots, 4\}$ and $\gamma_4 \geq \gamma_j$ for all $j \in \{1, \dots, 4\}$.*

*Proof.* Assume for the sake of contradiction that $\gamma_1$ is not the smallest among these thresholds. Then, the only other choice is $\gamma_3$ is the smallest, i.e., $\gamma_3 < \gamma_1$. It follows that $\beta > 1 - 2c$, otherwise $\gamma_1 = 0$. For any $\gamma \in [0, \gamma_3]$,
$$\gamma s_1^\star \leq s_2^\star(\gamma) \leq \beta(1 - \gamma),$$
because $\gamma_3 < \gamma_1$ (and hence, $s_{2,\beta}^\star = \frac{s_2^\star}{\beta}$). Substitution $\gamma = \gamma_3$ above, we get
$$\gamma_3 \leq \frac{\beta}{\beta + s_1^\star}. \tag{32}$$

Now for $\gamma \in [0, \gamma_3]$, we are in case $III$ and $III'$. Therefore, $s_2^\star$ satisfies:
$$s_1^\star - \frac{s_1^\star(2s_2^\star - \gamma s_1^\star)}{2(1-\gamma)} + 1 - \frac{2s_2^\star/\beta - \gamma}{2(1-\gamma)} = c. \tag{33}$$

Substituting $\gamma = \gamma_3$ and $s_2^\star = s_1^\star\gamma_3$ in the equation above, we get:
$$s_1^\star - \frac{(s_1^\star)^2\gamma_3}{2(1-\gamma_3)} + 1 - \frac{2\gamma_3 s_1^\star/\beta - \gamma_3}{2(1-\gamma_3)} = c.$$

Solving for $\gamma_3$ and using the fact that $c = 1 - s_1^\star$, we get
$$\gamma_3 = \frac{4s_1^\star}{4s_1^\star + (s_1^\star)^2 + 2s_1^\star/\beta - 1} \leq \frac{\beta}{s_1^\star + \beta},$$
where the second inequality follows from (32). Simplifying the above, we get:
$$4(s_1^\star)^2 - 2s_1^\star \leq \beta((s_1^\star)^2 - 1).$$

Now, the r.h.s. above is negative, but the l.h.s. is positive because $s_1^\star = 1 - c \geq 1/2$. This leads to a contradiction, and hence, it must be the case that $\gamma_1 \leq \gamma_3$.

Now we proceed to show that $\gamma_4$ is the largest among $\gamma_1, \dots, \gamma_4$. Suppose not. Then, $\gamma_2$ must be the largest value (since $\gamma_3 \leq \gamma_4$). Thus, the ordering of these thresholds would be $\gamma_1, \gamma_3, \gamma_4, \gamma_2$. In the interval $\gamma \in [\gamma_4, \gamma_2]$, we are in the case $I, IV'$ (and $s_2^\star \leq \gamma\beta \leq \beta$). Thus, $s_2^\star$ satisfies:
$$s_1^\star - \frac{(s_2^\star)^2}{2\gamma(1-\gamma)} + 1 - \frac{s_2^\star}{\beta\gamma} + \frac{1-\gamma}{2\gamma} = c.$$

When $\gamma = \gamma_2$, $s_2^\star = 1 - \gamma_2$. Substituting this fact above and solving for $\gamma_2$, we get
$$\gamma_2 = \frac{1}{1 + 2s_1^\star\beta}.$$

Since $\gamma_4 < \gamma_2$, we know that $1 - \gamma_2 < \beta\gamma_2$, i.e., $\gamma_2 > \frac{1}{1+\beta}$. Using this fact above, we see that $s_1^\star < 1/2$, which is a contradiction because $s_1^\star = 1 - c \geq 1/2$. Thus, we see that $\gamma_2 \leq \gamma_4$. This completes the proof of the desired result $\qquad\square$

It remains to decide between the orderings $\gamma_1, \gamma_2, \gamma_3, \gamma_4$ and $\gamma_1, \gamma_3, \gamma_2, \gamma_4$. It turns out that for a fixed $c$, both of these orderings can emerge as we vary $\beta$ from $1 - 2c$ to $1 - c$, but there is a critical value of $\beta$, denoted $\beta_c$, such that for all $\beta < \beta_c$, the latter ordering occurs; and we get the former ordering when $\beta > \beta_c$.

**Proposition K.3** (**Existence of a critical $\beta_c$ where the threshold order changes**). *Assume $c < 1/2$. Then there exists a value $\beta_c \in [0, 1 - c]$ such that for $1 - 2c \leq \beta < \beta_c$, $\gamma_1(\beta) \leq \gamma_3(\beta) \leq \gamma_2(\beta) \leq \gamma_4(\beta)$; and for all $\beta_c \leq \beta \leq 1 - c$, $\gamma_1(\beta) \leq \gamma_2(\beta) \leq \gamma_3(\beta) \leq \gamma_4(\beta)$.*

Observe that if $\beta_c = 0$ above, the ordering remains $\gamma_1, \gamma_2, \gamma_3, \gamma_4$ for the entire interval $[0, 1 - c]$ (and hence for the interval $[0, 1]$).

*Proof.* The proposition follows from the monotonicity of $\gamma_2(\beta)$ and $\gamma_3(\beta)$ as functions of $\beta$. We shall show that $\gamma_2(\beta)$ is a non-increasing function and $\gamma_3(\beta)$ is a non-decreasing function of $\beta$. The result now follows from the observation and the fact that when $\beta = 1 - c$, $\gamma_1 \leq \gamma_2 \leq \gamma_3 \leq \gamma_4$ (Theorem 3.1).

We shall treat $s_2^\star$ as a function of both $\beta$ and $\gamma$. Now observe that the ratio $\frac{s_2^\star(\beta,\gamma)}{1-\gamma}$ is monotonically increasing if we fix $\beta$ and increase $\gamma$ and monotonically non-decreasing if we fix $\gamma$ and increase $\beta$. Now, suppose there are values $\beta < \beta'$ such that $\gamma_2(\beta) < \gamma_2(\beta')$. By definition of $\gamma_2$,

$$\frac{s_2^\star(\beta, \gamma_2(\beta))}{1 - \gamma_2(\beta)} = \frac{s_2^\star(\beta', \gamma_2(\beta'))}{1 - \gamma_2(\beta')}.$$

Since $\gamma_2(\beta) < \gamma_2(\beta')$, monotonicity of $\frac{s_2^\star(\beta,\gamma)}{1-\gamma}$ in $\gamma$ implies that

$$\frac{s_2^\star(\beta, \gamma_2(\beta))}{1 - \gamma_2(\beta)} < \frac{s_2^\star(\beta, \gamma_2(\beta'))}{1 - \gamma_2(\beta')}.$$

Since $\beta < \beta'$, the monotonicity of $\frac{s_2^\star(\beta,\gamma)}{1-\gamma}$ in $\beta$ yields:

$$\frac{s_2^\star(\beta, \gamma_2(\beta))}{1 - \gamma_2(\beta)} < \frac{s_2^\star(\beta', \gamma_2(\beta'))}{1 - \gamma_2(\beta')},$$

which is a contradiction.

We can prove monotonicity of $\gamma_3(\beta)$ in a similar manner (note that $s_1^\star = 1 - c$, and hence, is independent of $\beta$ when $\beta \leq 1 - c$). $\qquad\square$

We now show that for $\beta_c$ to be strictly larger than $0$, $c$ must be at least $1/3$. In other words, if $c < 1/3$, then the ordering will be $\gamma_1, \gamma_2, \gamma_3, \gamma_4$ for all $\beta \in [0, 1 - c]$.

**Proposition K.4** (**Lower bound on $\beta_c$**). *Assume $c < 1/2$ and $\beta_c > 0$. Then $c \geq 1/3$.*

*Proof.* Suppose $c < 1/3$ and $\beta_c > 0$. Then consider the setting where $\beta = \beta_c$. Here, $\gamma_2(\beta_c) = \gamma_3(\beta_c)$. We consider the equation satisfied by $s_2^\star$ in the interval $[\gamma_1, \gamma_3]$. If $s_2^\star(\gamma) \leq \beta$, it satisfies:

$$s_1^\star - \frac{2s_1^\star s_2^\star - \gamma(s_1^\star)^2}{2(1 - \gamma)} + \frac{(1 - s_2^\star/\beta)^2}{2\gamma(1 - \gamma)} = c.$$

Otherwise, it satisfies:

$$s_1^\star - \frac{2s_1^\star s_2^\star - \gamma(s_1^\star)^2}{2(1 - \gamma)} = c.$$

In any case, we have

$$c \geq s_1^\star - \frac{2s_1^\star s_2^\star - \gamma(s_1^\star)^2}{2(1 - \gamma)}.$$

Now, $\gamma = \gamma_2$, which is also equal to $\gamma_3$, we have $s_2^\star = 1 - \gamma$ and $s_2^\star = \gamma s_1^\star$. Using both these conditions in the inequality above, we get:

$$c \geq s_1^\star - s_1^\star + \frac{(1-\gamma)s_1^\star}{2(1-\gamma)} = \frac{s_1^\star}{2} = \frac{1-c}{2}.$$

This shows that $c \geq 1/3$. $\hspace{1cm}$ $\square$

We make a few observations:

- Unlike the $\beta > 1 - c$ case, we can have four distinct thresholds. As we decrease $\beta$, some of these thresholds may not exist. In fact when $\beta$ goes below $\frac{1-2c}{1-c}$, threshold $\gamma_1$ does not appear. Intuitively, this means that for such small values of $\beta$, a candidate from $G_2$ cannot get admission in institute 2 based on $v_{i2}$ score alone, irrespective of the correlation parameter. Similarly $\gamma_4$ does not appear when $\beta$ goes below $1 - 2c$.

- The relative order of the thresholds may depend on the value of $\beta$. This is again different from the $\beta \geq 1 - c$ setting, where there was no such dependence on $\beta$.

### K.3 Variation of $s_2^\star$ with $\gamma$

In this section, we consider the variation of $s_2^\star$ with $\gamma$. In Theorem E.1, we had shown that $s_2^\star$ can vary in a non-monotone manner. We consider whether such a non-monotone behavior exists in the $\beta < 1 - c$ regime.

Since the relative order of the $\gamma$-thresholds depend on whether $\beta < \beta_c$, our results (and their analysis) split into two corresponding cases. The following proposition follows analogously as Theorem 3.2.

**Theorem K.5 (Characterization of $s_2^\star$ for low $\beta$).** *Assume that $\beta \leq 1 - c$, $\beta < \beta_c$ and $c < 1/2$. Then, assuming $s_2^\star(\gamma) \leq \beta$, it satisfies the following equations as $\gamma$ varies from $0$ to $1$:*

*(i) $[0, \gamma_1]$:*
$$s_1^\star - \frac{2s_1^\star s_2^\star - \gamma(s_1^\star)^2}{2(1-\gamma)} + 1 - \frac{2s_2^\star/\beta - \gamma}{2(1-\gamma)} = c. \tag{34}$$

*(ii) $[\gamma_1, \gamma_3]$:*
$$s_1^\star - \frac{2s_1^\star s_2^\star - \gamma(s_1^\star)^2}{2(1-\gamma)} + \frac{(1-s_2^\star/\beta)^2}{2\gamma(1-\gamma)} = c. \tag{35}$$

*(iii) $[\gamma_3, \gamma_2]$:*
$$s_1^\star - \frac{(s_2^\star)^2}{2\gamma(1-\gamma)} + \frac{(1-s_2^\star/\beta)^2}{2\gamma(1-\gamma)} = c. \tag{36}$$

*(iv) $[\gamma_2, \gamma_4]$:*
$$s_1^\star - \frac{s_2^\star}{\gamma} + \frac{1-\gamma}{2\gamma} + \frac{(1-s_2^\star/\beta)^2}{2\gamma(1-\gamma)} = c. \tag{37}$$

*(v) $[\gamma_4, 1]$:*
$$s_1^\star - \frac{s_2^\star}{\gamma} + \frac{1-\gamma}{\gamma} + 1 - \frac{s_2^\star}{\beta\gamma} = c \tag{38}$$

*When $\beta \leq 1 - c$ and $\beta \geq \beta_c, c < 1/2$, the equations satisfied in the intervals $[0, \gamma_1]$ and $[\gamma_4, 1]$ remain as above. For the remaining cases, we have:*

*(i) $[\gamma_1, \gamma_2]$:*
$$s_1^\star - \frac{2s_1^\star s_2^\star - \gamma(s_1^\star)^2}{2(1-\gamma)} + \frac{(1-s_2^\star/\beta)^2}{2\gamma(1-\gamma)} = c. \tag{39}$$

*(ii) $[\gamma_2, \gamma_3]$:*
$$\frac{(1-\gamma-s_2^\star+\gamma s_1^\star)^2}{2\gamma(1-\gamma)} + \frac{(1-s_2^\star/\beta)^2}{2\gamma(1-\gamma)} = c. \tag{40}$$

*(iii) $[\gamma_3, \gamma_4]$:*
$$s_1^\star - \frac{s_2^\star}{\gamma} + \frac{1-\gamma}{2\gamma} + \frac{(1-s_2^\star/\beta)^2}{2\gamma(1-\gamma)} = c. \tag{41}$$

The above equations only consider the case when $s_2^\star(\gamma) \leq \beta$. In case $s_2^\star(\gamma) > \beta$, we need to remove the terms corresponding to $G_2$ in the corresponding equation above. The following result shows that as we vary $\gamma$, $s_2^\star(\gamma)$ can have several local minima or maxima. This is in contrast to the $\beta > 1 - c$ setting, where there was only one local minimum.

**Theorem K.6 (Threshold in low $\beta$ regime).** *Let $c \in [0, 1/2)$. Assume that the bias parameter $\beta$ lies in the range $[0, 1 - c]$. When $\beta \leq \beta_c$, $s_2^\star$ varies as follows as we vary $\gamma$ from $0$ to $1$:*

*(i) $[0, \gamma_1]$: $s_2^\star$ is a linearly decreasing function of $\gamma$.*

*(ii) $[\gamma_1, \gamma_3]$: $s_2^\star$ is a unimodal function of $\gamma$: it has no local maxima and has at most one local minimum.*

*(iii) $[\gamma_3, \gamma_2]$ : $s_2^\star$ is a unimodal function of $\gamma$: it has no local minima and at most one local maximum.*

*(iv) $[\gamma_2, \gamma_4]$: $s_2^\star$ is a unimodal function; it has no local maximum and at most one local minimum.*

*(v) $[\gamma_4, 1]$: $s_2^\star$ is a linearly increasing function of $\gamma$.*

*When $\beta > \beta_c$, $s_2^\star$ behaves as in the above case for the intervals $[0, \gamma_1]$, $[\gamma_4, 1]$. For the remaining three intervals, $s_2^\star$ varies as follows:*

*(i) $[\gamma_1, \gamma_2]$: $s_2^\star$ is a unimodal function of $\gamma$: it has no local maxima and has at most one local minimum.*

*(ii) $[\gamma_2, \gamma_3]$: $s_2^\star$ is a convex function, and hence, has at most one local minimum and no local maximum.*

*(iii) $[\gamma_3, \gamma_4]$: $s_2^\star$ is a unimodal function. If $c > 1/4$, it has no local maximum and at most one local minimum. If $c \leq 1/4$, it has no local minimum and at most one local maximum.*

*Proof.* First, consider the case when $\beta < \beta_c$. We consider various cases:

**Case $\gamma \in [0, \gamma_1]$:** Since $\gamma \leq \gamma_1$, $s_2^\star \leq \beta$. Further, (34) shows that $s_2^\star$ is a linear function of $\gamma$ with the slope given by $\frac{1}{2s_1^\star + 2/\beta}$ times $-4s_1^\star + (s_1^\star)^2 + 1$. The latter quantity is negative if $s_1^\star \geq 1/2$ (which is the case when $c < 1/2$).

**Case $\gamma \in [\gamma_1, \gamma_3]$:** We know that $s_2^\star(\gamma_1) \leq \beta$. If $s_2^\star(\gamma) \geq \beta$, then $s_2^\star(\gamma)$ satisfies the following truncated version of (35):

$$s_1^\star - \frac{2s_1^\star s_2^\star - \gamma(s_1^\star)^2}{2(1 - \gamma)} = c.$$

This becomes a linear function in $s_2^\star$ with slope proportional to $(s_1^\star)^2 - 4(s_1^\star) + 2$, which is positive iff $c > \sqrt{2} - 1$. In case $c > \sqrt{2} - 1$, and if $s_2^\star$ exceeds $\beta$, it will remain above $\beta$ (and increase linearly) for the rest of the interval. However if $c < \sqrt{2} - 1$, $s_2^\star$ remains below $\beta$ during this interval. Thus, there is a value $\gamma_3' \leq \gamma_3$, such that $s_2^\star$ remains at most $\beta$ during $[\gamma_1, \gamma_3']$, and remains above $\beta$ during $(\gamma_3', \gamma_3]$. Note that $\gamma_3'$ may equal $\gamma_3$, in which case the latter interval is empty. We now study the behavior of $s_2^\star$ in $[\gamma_1, \gamma_3']$. We know that it satisfies (35).

Using $c = 1 - s_1^\star$, we rewrite (35) as

$$2(2s_1^\star - 1)\gamma(1 - \gamma) - 2s_1^\star s_2^\star \gamma + \gamma^2(s_1^\star)^2 + (1 - s_2^\star/\beta)^2 = 0.$$

Differentiating with respect to $\gamma$, we get

$$2(1 - 2\gamma)(2s_1^\star - 1) - 2s_1^\star s_2^\star + 2\gamma(s_1^\star)^2 - 2(s_1^\star \gamma + 1/\beta(1 - s_2^\star/\beta))(s_2^\star)' = 0.$$

Differentiating again, we get

$$2(s_2^\star)''(s_1^\star \gamma + 1/\beta(1 - s_2^\star/\beta) - \frac{2}{\beta^2}((s_2^\star)')^2 + 4s_1^\star(s_2^\star)' + 8s_1^\star - 4(s_1^\star)^2 - 4 = 0.$$

Note that $8s_1^\star - 4(s_1^\star)^2 - 4 < 0$. It follows that when $|(s_2^\star)'|$ is close to $0$, $(s_2^\star)''$ becomes positive. Thus, once the slope $(s_2^\star)'$ becomes positive, it does not become $0$ again. Therefore, there can only be a local minimum in $[\gamma_1, \gamma_3]$.

**Case $\gamma \in [\gamma_3, \gamma_2]$:** We know that if $s_2^\star \leq \beta$, it satisfies (36). In case, $s_2^\star > \beta$, it would satisfy:

$$s_1^\star - \frac{(s_2^\star)^2}{2\gamma(1 - \gamma)} = c.$$

If $\gamma < 1/2$, $s_2^\star$ will be an increasing function. At $\gamma = \gamma_3$, we know that $s_1^\star \gamma_3 = 1 - \gamma_3$, which implies that $\gamma_3 < 1/2$. Thus, if $s_2^\star(\gamma_3) \geq \beta$, there would be an initial sub-interval $[\gamma_3, \gamma_2']$ of $[\gamma_3, \gamma_2]$ where $s_2^\star$ will be above $\beta$ and would subsequently stay below $\beta$.

Now, we consider those values of $\gamma$ where $s_2^\star \leq \beta$. Multiplying both sides of (36) by $2\gamma(1 - \gamma)$ and differentiating, we see that $(s_2^\star)'$ is equal to a positive quantity times $(s_1^\star - c)(1 - 2\gamma)$. Since $c < 1/2$, $s_1^\star = 1 - c > c$. Thus, $(s_2)^\star$ is an increasing function till $\gamma = 1/2$ and then it becomes a decreasing function. Combining the above observations, we see that $s_2^\star$ will be initially increasing, and then will become decreasing – note that $\gamma_2 > 1/2$ because $1 - \gamma_2 = s_2^\star(\gamma_2) \leq s_1^\star \gamma_2 < \gamma_2$.

**Case $\gamma \in [\gamma_2, \gamma_4]$:** If $s_2^\star \leq \beta$, then $s_2^\star$ satisfies (37). Otherwise, it would satisfy:

$$s_1^\star - \frac{s_2^\star}{\gamma} + \frac{1 - \gamma}{2\gamma} = c, \tag{42}$$

In this case, we multiply both sides of (42) by $2\gamma$ and differentiate w.r.t. $\beta$ to get:

$$2(s_2^\star)' = 2(s_1^\star - c) - 1 = 1 - 4c < 0,$$

because $c > 1/4$ (Proposition K.4). Thus, $s_2^\star$ would be a decreasing function till it becomes at most $\beta$. Hence we get: there is a value $\gamma_2' \in [\gamma_2, \gamma_4]$ such that $s_2^\star(\gamma) \geq \beta$ for all $\gamma \in (\gamma_2, \gamma_2']$ and is a decreasing function of $\gamma$ during this interval. Note that $\gamma_2'$ could be same as $\gamma_2$ in which case $s_2^\star(\gamma_2) \leq \beta$.

Now we consider the remaining interval $[\gamma_2', \gamma_4]$. When $s_2^\star \leq \beta$, it satisfies (37). Multiplying both sides by $2\gamma(1 - \gamma)$ and differentiating (and using $c = 1 - s_1^\star$), we get:

$$2(s_1^\star - c)(1 - 2\gamma) + 2s_2^\star - 2(1 - \gamma) - \frac{2}{\beta}(1 - s_2^\star/\beta)(s_2^\star)' - (1 - \gamma)(s_2^\star)' = 0.$$

Differentiating the above equation again, we get:

$$A(\gamma)(s_2^\star)'' - \frac{2}{\beta}((s_2^\star)')^2 - \left(3 + \frac{2}{\beta^2}\right)(s_2^\star)' + 4(s_1^\star - c) - 2 = 0,$$

where $A(\gamma) = (1 - \gamma) + \frac{2}{\beta}(1 - s_2^\star/\beta) > 1 - \gamma_4$. Further, $4(s_1^\star - c) - 2 = 4(1 - 2c) - 2 = 2 - 8c < 0$ because $c \geq 1/3$ (Proposition K.4). Thus, when $(s_2^\star)'$ becomes small enough, $(s_2^\star)''$ becomes positive. Therefore, $s_2^\star$ does not have a local maximum in $[\gamma_2', \gamma_4]$. Further, $s_2^\star(\gamma_4) = \beta\gamma_4 < \beta$. Therefore, $s_2^\star \leq \beta$ throughout the interval $[\gamma_2', \gamma_4]$. Combining the observations about $[\gamma_2, \gamma_2')$ and $[\gamma_2', \gamma_4]$, we see that $s_2^\star$ initially decreases and does not have a local maximum in $[\gamma_2, \gamma_4]$.

**Case $\gamma \in [\gamma_4, \gamma_1]$:** Since $s_2^\star \leq \beta\gamma < \beta$ throughout this interval, it satisfies (38) for all $\gamma \in [\gamma_4, 1]$. It follows from (38) that

$$s_2^\star = \frac{2s_1^\star \beta\gamma + (1 - \gamma)\beta}{\beta + 1}.$$

Since $s_1^\star > 1/2$, $s_2^\star$ is an increasing function of $\gamma$.

This completes the discussion on various cases when $\beta < \beta_c$. It is also worth noting from the discussion for intervals $[\gamma_1, \gamma_3]$ and $[\gamma_3, \gamma_2]$ the set of $\gamma$ values where $s_2^\star(\gamma) \geq \beta$ forms an interval.

Now we consider the case when $\beta > \beta_c$. Clearly the cases $[0, \gamma_1]$ and $[\gamma_4, 1]$ follow as above. We now consider the remaining settings:

**Case $\gamma \in [\gamma_1, \gamma_2]$:** In this interval, $s_2^\star$ satisfies (39) which is identical to (35). Thus, the arguments used in the case $\gamma \in [\gamma_1, \gamma_3]$ when $\beta < \beta_c$ apply in the same manner here.

**Case $\gamma \in [\gamma_2, \gamma_3]$:** In this case, $s_2^\star$ satisfies (40) if assuming $s_2^\star \leq \beta$. In case, $s_2^\star > \beta$, it satisfies:

$$\frac{(1 - \gamma - s_2^\star + \gamma s_1^\star)^2}{2\gamma(1 - \gamma)} = c. \tag{43}$$

Multiplying both sides by $2\gamma(1 - \gamma)$ and differentiating twice, we see that $(s_2^\star)'' > 0$. When $s_2^\star(\gamma) \leq \beta$, it satisfies (40). Again, multiplying both sides of this equation by $2\gamma(1 - \gamma)$ and differentiating twice, we see that $(s_2^\star)'' > 0$. Thus $(s_2^\star)'' > 0$ at all points in $[\gamma_2, \gamma_3]$ (one can check

that if $s_2^\star(\gamma) = \beta$, then the slope $(s_2^\star(\gamma))'$ given by the two equation (40) and (42) are identical). Hence, $s_2^\star$ is convex in this interval.

**Case $\gamma \in [\gamma_3, \gamma_4]$:** In this case, $s_2^\star$ satisfies (41) which is identical to (37). Thus, the arguments used in the case for the case $\beta \in [\gamma_2, \gamma_4]$ when $\beta < \beta_c$ apply in a similar manner as long as $c > 1/4$.

We complete the argument when $c < 1/4$. In case $s_2^\star(\gamma_3) > \beta$, it would satisfy (42). Multiplying both sides by $2\gamma$ and differentiating, we see that $s_2^\star$ will be a non-decreasing linear function (and hence, remain above $\beta$ throughout this interval).

Now consider the case when $s_2^\star(\gamma_3) \leq \beta$. As in the case $\beta \in [\gamma_2, \gamma_4]$ for $\beta < \beta_c$, we get the equation:

$$A(\gamma)(s_2^\star)'' - \frac{2}{\beta}((s_2^\star)')^2 - \left(3 + \frac{2}{\beta^2}\right)(s_2^\star)' + 4(s_1^\star - c) - 2 = 0,$$

where $A(\gamma) = (1 - \gamma) + \frac{2}{\beta}(1 - s_2^\star/\beta) > 1 - \gamma_4$. Now, if $c > 1/4$, $4(s_1^\star - c) - 2 > 0$, and rest of the arguments in the $\beta < \beta_c$ case hold.

We have shown that $s_2^\star(\gamma_2) \leq \beta$. If $c < 1/4$, we see that $4(s_1^\star - c) - 2 < 0$. This shows that if $(s^\star)'$ is close enough to 0, then $(s_2^\star)'' < 0$. Thus, it does not achieve a local minimum, though it may be a local maximum. This completes the discussion on all the cases. □

The result above shows that when $\beta < 1 - c$, $s_2^\star(\gamma)$ can behave in a highly non-monotone manner as we vary $\gamma$. Similar observations hold for the representation ratio:

**Variation of representation ratio with $\gamma$.** In Theorem 3.3, we showed that when $\beta \geq 1 - c$, $\mathcal{R}(\gamma)$ is a monotonically non-decreasing function of $\gamma$. However, when $\beta < 1 - c$, it is possible that $\mathcal{R}(\gamma)$ is a decreasing function of $\gamma$ in a sub-interval of $[0, 1]$ and is an increasing function of $\gamma$ in another sub-interval. Thus, $\mathcal{R}(\gamma)$ may not be a monotone function of $\gamma$. We now show that unlike the $\beta > 1 - c$ case, $\mathcal{R}(\gamma)$ can decrease with $\gamma$. We shall first need a useful result:

**Fact K.7 (Monotonicity under an implicit constraint).** *Let $f, g : \mathbb{R}^2 \to \mathbb{R}$ be functions of two variables, and $z : [0, 1] \to \mathbb{R}$ be a real valued function defined on the interval $[0, 1]$. Suppose the following relation holds for all $\gamma \in [a, b]$ for some $0 \leq a \leq b \leq 1$:*

$$f(\gamma, z(\gamma)) + g(\gamma, z(\gamma)) = c,$$

*where $c$ is a constant. Let $f_\gamma, f_z$ denote the partial derivatives of $f$ with respect to the two coordinates (and define $g_\gamma, g_z$ similarly). Suppose $f_z, g_z > 0$ and*

$$\frac{f_\gamma}{f_z} \leq \frac{g_\gamma}{g_z} \tag{44}$$

*for all $\gamma \in [0, 1]$. Then $f(\gamma, z(\gamma))$ is a non-increasing function of $\gamma \in [a, b]$. Similarly, if $f_z, g_z < 0$ and*

$$\frac{f_\gamma}{f_z} \geq \frac{g_\gamma}{g_z} \tag{45}$$

*for all $\gamma \in [a, b]$. Then $f(\gamma, z(\gamma))$ is a non-increasing function of $\gamma \in [a, b]$.*

*Proof.* First assume $f_z, g_z > 0$. Differentiating

$$f(\gamma, z(\gamma)) + g(\gamma, z(\gamma)) = c$$

with respect to $\gamma$ (and using $z'$ to denote $\frac{dz}{d\gamma}$), we get

$$f_\gamma + f_z z' + g_\gamma + g_z z' = 0.$$

Therefore,

$$z' = -\frac{f_\gamma + g_\gamma}{f_z + g_z}.$$

Since $f_z, g_z > 0$, (44) implies that

$$\frac{f_\gamma + g_\gamma}{f_z + g_z} \geq \frac{f_\gamma}{f_z}.$$

Therefore, $z' \leq -\frac{f_\gamma}{f_z}$. Now,

$$\frac{df(\gamma, z(\gamma))}{d\gamma} = f_z z' + f_\gamma \leq -f_\gamma + f_\gamma \leq 0.$$

This proves the desired result. The argument when $f_z, g_z < 0$ is similar. $\qquad\square$

We now show that the representation ratio may decrease with increasing $\gamma$:

**Proposition K.8 (Representation ratio decreases in mid-bias range).** *Assume $c < 1/2$ and $\frac{1-2c}{1-c} < \beta < 1 - c$. Then representation ratio decreases monotonically with $\gamma$ for $\gamma \in [0, \gamma_1]$.*

Note that the condition $\frac{1-2c}{1-c} < \beta$ is needed to ensure that $\gamma_1 > 0$ (Proposition K.2).

*Proof.* For $\gamma \in [0, \gamma_1]$, $s_2^\star \leq \beta$ and hence, $s_2^\star$ satisfies Equation (34). We show that the representation ratio is decreasing using the notation in (44). Here, we define $z = -\frac{s_2^\star}{(1-\gamma)}$ (the negative sign is needed because we want $f_z, g_z$ to be positive). Then, the terms in the l.h.s. of (34) corresponding to $G_1$ and $G_2$ can be expressed as:

$$f(\gamma, z) = s_1^\star(1 + z) + \frac{\gamma(s_1^\star)^2}{2(1-\gamma)}, \quad g(\gamma, z) = 1 + z/\beta + \frac{\gamma}{2(1-\gamma)}.$$

A routine calculation shows that

$$\frac{f_\gamma}{f_z} = \frac{s_1^\star}{2(1-\gamma)^2}, \quad \frac{g_\gamma}{g_z} = \frac{\beta}{2(1-\gamma)^2}.$$

Since $s_1^\star \geq \beta$, the reverse of (44) holds. This shows that $f(\gamma, z)$, which represents the fraction of selected candidates from $G_1$, is an increasing function of $\gamma$. Thus $\mathcal{R}(\gamma)$ is a decreasing function of $\gamma$. $\qquad\square$

# L   The general preferences case

In this section, we extend our analysis to the general preference setting, where candidates prefer Institution 1 with probability $p \in (0, 1)$. Unlike the case when Institution 1 is always preferred, the equilibrium thresholds $s_1^\star$ and $s_2^\star$ now satisfy nonlinear equations that do not admit closed-form solutions. Nevertheless, this generality reveals rich structural behavior.

- In Appendix L.1, we derive the equilibrium conditions that implicitly define $s_1^\star$ and $s_2^\star$ for arbitrary $p$. We further prove the existence and uniqueness of these thresholds.

- In Appendix L.2, we show how to compute these thresholds using closed-form probability expressions developed earlier, enabling efficient numerical evaluation.

- In Appendix L.3, we provide formulas for computing key outcome metrics in terms of the thresholds: the representation ratio and each institution's observed utility.

- In Appendix L.4, we prove that the thresholds vary monotonically with $p$, holding other parameters fixed. However, this variation is not necessarily strictly monotonic, as illustrated in Figure 8. In particular, the threshold $s_1^\star$ remains constant for values of $p$ up to a critical point $p^\star$, beyond which it increases strictly with $p$.
  This behavior arises because, for small values of $p$, the threshold $s_1^\star(p)$ is sufficiently low while $s_2^\star(p)$ is relatively high, resulting in the probability $\Pr[u_{i1} < s_1^\star(p) \wedge u_{i2} \geq s_2^\star(p)]$ being zero. Consequently, all candidates with $u_{i1} \geq s_1^\star(p)$ are selected by one of the two institutions, regardless of the value of $p$. Since this selection condition does not depend on $p$, the threshold $s_1^\star(p)$ remains unchanged for $p \leq p^\star$. We formalize this argument—along with a similar statement for $s_2^\star(p)$—in Appendix L.4.1.

- In Appendix L.5, we numerically examine how thresholds, fairness metrics, and utilities evolve as $p$ and the correlation parameter $\gamma$ vary. A key contribution of this section lies in the comprehensive set of plots (Figures 8–14) that reveal monotonicity, phase transitions, and other emergent behaviors, offering insights that go beyond what closed-form analysis can capture.

## L.1 Equations for the thresholds and existence/uniqueness of solution

### L.1.1 Equations for the thresholds

We now derive the threshold equations for the two-group model, where each candidate independently prefers Institution 1 over Institution 2 with probability $p \in [0, 1]$, and belongs to either group $G_1$ or $G_2$. Candidates in group $G_2$ face a multiplicative bias $\beta \in (0, 1]$ in their evaluations. Let $i \in G_1$, $i' \in G_2$, and define the bias-adjusted thresholds:

$$s_{1,\beta} = \min(1, s_1/\beta), \quad s_{2,\beta} = \min(1, s_2/\beta).$$

We assume uniform group sizes ($\nu_1 = \nu_2 = 1$) and symmetric capacities ($c_1 = c_2 = c$). Our approach follows a finite-to-infinite reduction analogous to that in Section B, transitioning from random threshold behavior in the finite setting to deterministic thresholds in the continuum limit. The capacity constraints reduce to the following pair of equations:

$$p\left((1 - s_1) + (1 - s_{1,\beta})\right) + (1 - p)\left(\Pr[u_{i1} \geq s_1 \wedge u_{i2} < s_2] + \Pr[u_{i'1} \geq s_{1,\beta} \wedge u_{i'2} < s_{2,\beta}]\right) = c,$$
$$(46)$$

$$(1 - p)\left(\Pr[u_{i2} \geq s_2] + \Pr[u_{i'2} \geq s_{2,\beta}]\right) + p\left(\Pr[u_{i1} < s_1 \wedge u_{i2} \geq s_2] + \Pr[u_{i'1} < s_{1,\beta} \wedge u_{i'2} \geq s_{2,\beta}]\right) = c.$$
$$(47)$$

### L.1.2 Existence and uniqueness of $(s_1^\star, s_2^\star)$ for general $p$

The proof of existence and uniqueness also follows the $p = 1$ setting closely (see the proof of Proposition B.2). Define the residuals

$$F_1(s_1, s_2) = p\left[(1 - s_1) + (1 - s_{1,\beta})\right] + (1 - p)\left[\Phi_{10}(s_1, s_2) + \Phi_{10,\beta}(s_1, s_2)\right] - c,$$

$$F_2(s_1, s_2) = (1 - p)\left[\Psi_0(s_2) + \Psi_{0,\beta}(s_2)\right] + p\left[\Phi_{01}(s_1, s_2) + \Phi_{01,\beta}(s_1, s_2)\right] - c,$$

where, for example, $\Phi_{10}(s_1, s_2) = \Pr[u_{i1} \geq s_1, \ u_{i2} < s_2]$, $\Psi_0(s_2) = \Pr[u_{i2} \geq s_2]$, and the "$\beta$"–variants are defined analogously.

**Proposition L.1** (**Existence and uniqueness**). *The system*

$$F_1(s_1, s_2) = 0, \quad F_2(s_1, s_2) = 0, \qquad (s_1, s_2) \in (0, 1)^2$$

*admits a unique solution $(s_1^\star, s_2^\star)$. Moreover, the map $p \mapsto (s_1^\star, s_2^\star)$ is continuous on $(0, 1)$.*

*Proof.* **1. Monotonicity of each equation.** By the uniform noise formulas, one checks that

$$\frac{\partial F_1}{\partial s_1} < 0, \quad \frac{\partial F_1}{\partial s_2} > 0, \quad \frac{\partial F_2}{\partial s_1} > 0, \quad \frac{\partial F_2}{\partial s_2} < 0,$$

throughout $(0, 1)^2$. In particular, each residual is strictly monotone in each variable.

**2. Existence via one-dimensional roots.** - For each fixed $s_2 \in [0, 1]$, the function $s_1 \mapsto F_1(s_1, s_2)$ is continuous, strictly decreasing, and one verifies

$$F_1(0, s_2) > 0, \quad F_1(1, s_2) < 0.$$

Hence there is a unique root $s_1 = \Lambda(s_2) \in (0, 1)$ with $F_1\left(\Lambda(s_2), s_2\right) = 0$. By strict monotonicity in $s_1$, $\Lambda$ is continuous and strictly *increasing* in $s_2$.

Likewise, for each fixed $s_1 \in [0, 1]$, the function $s_2 \mapsto F_2(s_1, s_2)$ is continuous, strictly *decreasing*, and satisfies

$$F_2\left(s_1, 0\right) > 0, \quad F_2\left(s_1, 1\right) < 0,$$

so there is a unique root $s_2 = \Gamma(s_1) \in (0, 1)$. Again by strict monotonicity, $\Gamma$ is continuous and strictly *decreasing* in $s_1$.

**3. Uniqueness by graph intersection.** The solution $(s_1^\star, s_2^\star)$ must satisfy both

$$s_1 = \Lambda(s_2), \qquad s_2 = \Gamma(s_1).$$

Equivalently, the continuous curve $s_2 \mapsto \Lambda(s_2)$ (in the $s_1$-direction) and the graph of the inverse $\Gamma^{-1}$ meet. Since $\Lambda$ is strictly increasing and $\Gamma$ strictly decreasing, there is exactly one intersection in $(0,1)^2$. This proves uniqueness.

**4. Continuity in $p$.** One checks that the Jacobian $\det D_{(s_1,s_2)}(F_1, F_2)$ is strictly negative on $(0,1)^2$ (it is $\partial_{s_1} F_1 \cdot \partial_{s_2} F_2 - \partial_{s_2} F_1 \cdot \partial_{s_1} F_2 < 0$), so by the Implicit Function Theorem, the unique solution $(s_1^\star, s_2^\star)$ depends $C^1$-smoothly on $p$. In particular, it is continuous. $\qquad\square$

### L.2 Computing the thresholds

In this section, we describe our approach for computing the thresholds $s_1^\star$ and $s_2^\star$. Specifically, we show how the probability expressions in equations (46) and (47) can be evaluated using the analytical results developed in Section 3. Unlike the special case when $p = 1$, it is now possible that $s_2^\star > s_1^\star$, which necessitates careful treatment of events such as $\{u_{i2} \geq s_2 \wedge u_{i1} < s_1\}$. In particular, when

$$s_2 \geq s_1 \gamma + (1 - \gamma),$$

this event has zero probability due to the structure of the support, i.e., $\Pr[u_{i2} \geq s_2 \wedge u_{i1} < s_1] = 0$. A similar observation applies when evaluating the corresponding bias-adjusted probabilities for candidates in group $G_2$.

Our procedure consists of the following steps:

- **Rewriting probability expressions.** We express the probability terms in (46) and (47) using quantities of the form

  $$\Pr[u_{i1} < s_1 \wedge u_{i2} \geq s_2] - \Pr[u_{i2} \geq s_2],$$

  and $\Pr[u_{i2} \geq s_2]$ for candidates $i \in G_1$, and analogously for $i' \in G_2$. These forms simplify the evaluation using the results from Section 3.

- **Case identification.** The expression $\Pr[u_{i1} < s_{1,\beta} \wedge u_{i2} \geq s_{2,\beta}]$ depends on the relative positioning of $s_2$ with respect to $s_1 \gamma$ and $1 - \gamma$. Section 3 identifies four distinct cases under the assumption that $s_2^\star \leq s_1^\star$, but this assumption may not hold in the general setting. For instance, if $s_2^\star \geq s_1 \gamma + (1 - \gamma)$, then the probability is zero. Geometrically, this corresponds to the line $L(\gamma, s_2)$ not intersecting the rectangle $[0, s_1] \times [0, 1]$.

  Similar logic applies to $\Pr[u_{i2} \geq s_2]$. Overall, we identify seven mutually exclusive cases for computing $\Pr[u_{i1} < s_1 \wedge u_{i2} \geq s_2] - \Pr[u_{i2} \geq s_2]$ and $\Pr[u_{i2} \geq s_2]$. For candidates in $G_1$, only seven cases are feasible. For candidates in $G_2$, we must consider twenty-eight total configurations, accounting for the seven cases above along with independent binary assumptions on whether $s_1 \leq \beta$ and whether $s_2 \leq \beta$.

- **Solving and verifying.** For each guess of the applicable case(s), we substitute the corresponding closed-form expressions into equations (46) and (47), resulting in a system of two equations in the unknowns $s_1$ and $s_2$. We solve this system and verify whether the solution satisfies all assumptions made in the guess. Since the solution to the threshold equations is unique, exactly one of the constant number of guesses will yield a consistent and valid threshold pair $(s_1^\star, s_2^\star)$.

We now elaborate on each of the steps above:

**Rewriting probability expressions.** We first consider (46). We apply the identity:

$$\Pr[u_{i1} \geq s_1 \wedge u_{i2} < s_2] = (1 - s_1) - \Pr[u_{i2} \geq s_2] + \Pr[u_{i1} < s_1 \wedge u_{i2} \geq s_2],$$

and similarly for $i'$. This yields:

$$(1 - s_1) + (1 - s_{1,\beta}) + (1 - p)\big(\Pr[u_{i1} < s_1 \wedge u_{i2} \geq s_2] - \Pr[u_{i2} \geq s_2] \tag{48}$$
$$+ \Pr[u_{i'1} < s_{1,\beta} \wedge u_{i'2} \geq s_{2,\beta}] - \Pr[u_{i'2} \geq s_{2,\beta}]\big) = c.$$

Note that $\Pr[u_{i1} \geq s_2]$ can be expressed as $\Pr[u_{i1} \geq s_2 \wedge u_{i1} \leq 1]$.

The second equation (47) remains unchanged – the probability terms are already expressed in terms of the desired events.

**Case identification.** Both the equations (48) and (47) involve $\Pr[u_{i1} < s_1 \wedge u_{i2} \geq s_2] - \Pr[u_{i2} \geq s_2]$ and $\Pr[u_{i1} \geq s_2]$ for a candidate $i \in G_1$ and similarly for a candidate $i' \in G_2$. We now show the possibilities for a candidate $i \in G_1$:

(i) $s_2 \leq \max(1 - \gamma, \gamma s_1)$: we are in Case I while computing both $\Pr[u_{i1} < s_1 \wedge u_{i2} \geq s_2]$ and $\Pr[u_{i2} \geq s_2]$. Thus,

$$\Pr[u_{i1} < s_1 \wedge u_{i2} \geq s_2] - \Pr[u_{i2} \geq s_2] = s_1 - 1,$$

and

$$\Pr[u_{i2} \geq s_2] = 1 - \frac{s_2^2}{2\gamma(1 - \gamma)}.$$

(ii) $1 - \gamma < s_2 \leq \max(1 - \gamma, \gamma s_1)$: we are in Case IV for the both the probability events. Hence,

$$\Pr[u_{i1} < s_1 \wedge u_{i2} \geq s_2] - \Pr[u_{i2} \geq s_2] = s_1 - 1,$$

and

$$\Pr[u_{i2} \geq s_2] = 1 - \frac{s_2}{\gamma} + \frac{1 - \gamma}{2\gamma}.$$

(iii) $\frac{s_2}{\gamma} \in [s_1, 1], \frac{s_2 - \gamma s_1}{1 - \gamma} \in [0, 1]$ : Observe that $\gamma s_1 \leq s_2 \leq 1 - \gamma$ and $s_2 \leq \min(\gamma, 1 - \gamma)$. Thus, we are in Case III for $\Pr[u_{i1} < s_1 \wedge u_{i2} \geq s_2]$, but in Case I for $\Pr[u_{i2} \geq s_2]$. Therefore,

$$\Pr[u_{i1} < s_1 \wedge u_{i2} \geq s_2] - \Pr[u_{i2} \geq s_2] = \frac{s_2^2}{2\gamma(1 - \gamma)} - \frac{s_1(2s_2 - \gamma s_1)}{2(1 - \gamma)},$$

and

$$\Pr[u_{i2} \geq s_2] = 1 - \frac{s_2^2}{2\gamma(1 - \gamma)}.$$

(iv) $\frac{s_2}{\gamma} \in [s_1, 1], \frac{s_2 - \gamma s_1}{1 - \gamma} > 1$ : Observe that $s_2 \geq \gamma s_1 + (1 - \gamma)$ and $1 - \gamma \leq s_2 \leq \gamma$. Thus, $\Pr[u_{i1} < s_1 \wedge u_{i2} \geq s_2] = 0$, but we are in Case IV for $\Pr[u_{i2} \geq s_2]$. Therefore,

$$\Pr[u_{i1} < s_1 \wedge u_{i2} \geq s_2] - \Pr[u_{i2} \geq s_2] = -1 + \frac{s_2}{\gamma} - \frac{1 - \gamma}{2\gamma},$$

and

$$\Pr[u_{i2} \geq s_2] = 1 - \frac{s_2}{\gamma} + \frac{1 - \gamma}{2\gamma}.$$

(v) $\frac{s_2}{\gamma} > 1, \frac{s_2 - \gamma s_1}{1 - \gamma} > 1$ : Observe that $s_2 \geq \gamma s_1 + (1 - \gamma)$ and $s_2 \geq \max(\gamma, 1 - \gamma)$. Thus, $\Pr[u_{i1} < s_1 \wedge u_{i2} \geq s_2] = 0$, but we are in Case II for $\Pr[u_{i2} \geq s_2]$. Therefore,

$$\Pr[u_{i1} < s_1 \wedge u_{i2} \geq s_2] - \Pr[u_{i2} \geq s_2] = -\frac{(1 - s_2)^2}{2\gamma(1 - \gamma)},$$

and

$$\Pr[u_{i2} \geq s_2] = \frac{(1 - s_2)^2}{2\gamma(1 - \gamma)}.$$

(vi) $\frac{s_2}{\gamma} > 1, \frac{s_2 - \gamma s_1}{1 - \gamma} \in [0, 1], s_2 \leq 1 - \gamma$ : In this Case, $s_1 \gamma \leq s_2 \leq 1 - \gamma$ and $\gamma \leq s_2 \leq 1 - \gamma$. Thus, we are in Case III for both probability events. Hence,

$$\Pr[u_{i1} < s_1 \wedge u_{i2} \geq s_2] - \Pr[u_{i2} \geq s_2] = s_1 - \frac{s_1(2s_2 - \gamma s_1)}{2(1 - \gamma)} - 1 + \frac{2s_2 - \gamma}{1 - \gamma},$$

and

$$\Pr[u_{i2} \geq s_2] = 1 - \frac{2s_2 - \gamma}{2(1 - \gamma)}$$

(vii) $\frac{s_2}{\gamma} > 1, \frac{s_2 - \gamma s_1}{1 - \gamma} \in [0, 1], s_2 \geq 1 - \gamma$ : Here $s_2 \geq \max(\gamma, 1 - \gamma)$ and $s_2 \geq \max(\gamma, s_1 \gamma)$. Thus, we are in Case II for both the probability expressions. Thus,

$$\Pr[u_{i1} < s_1 \wedge u_{i2} \geq s_2] - \Pr[u_{i2} \geq s_2] = \frac{1 - \gamma - s_2 + \gamma s_1)^2}{2\gamma(1 - \gamma)} - \frac{(1 - s_2)^2}{2\gamma(1 - \gamma)},$$

and

$$\Pr[u_{i2} \geq s_2] = \frac{(1 - s_2)^2}{2\gamma(1 - \gamma)}.$$

Similarly, we can express $\Pr[u_{i'1} < s_{1,\beta} \wedge u_{i'2} \geq s_2] - \Pr[u_{i'2} \geq s_{2,\beta}]$.

**Formulation of system of equations for $s_1^\star$ and $s_2^\star$.** We now illustrate an example of the system of equations that arises from specific guesses made in the previous step. Suppose, we consider Case (v) for $G_1$ and Case (vi) for $G_2$. Further, we also assume that $s_1^\star \leq \beta$ but $s_2^\star \geq \beta$. Under these assumptions, we have

$$\Pr[u_{i1} < s_1 \wedge u_{i2} \geq s_2] - \Pr[u_{i2} \geq s_2] = -\frac{(1-s_2)^2}{2\gamma(1-\gamma)}$$

for a candidate $i \in G_1$ and

$$\Pr[u_{i'1} < s_{1,\beta} \wedge u_{i'2} \geq s_{2,\beta}] - \Pr[u_{i2} \geq s_{2,\beta}] = \frac{s_1}{\beta} - \frac{s_1(2 - \gamma s_1/\beta)}{2\beta(1-\gamma)} - 1 + \frac{2-\gamma}{1-\gamma}.$$

Thus, Equation (48) can be written as:

$$(1 - s_1) - p(s_1/\beta - 1) - (1-p)\frac{(1-s_2)^2}{2\gamma(1-\gamma)} - (1-p)\frac{s_1(2 - \gamma s_1/\beta)}{2\beta(1-\gamma)} + (1-p)\frac{2-\gamma}{1-\gamma} = c.$$

The second equation obtained from (47) is:

$$\frac{(1-s_2)^2}{2\gamma(1-\gamma)} + \frac{2-\gamma}{2(1-\gamma)} + p\left(-\frac{(1-s_2)^2}{2\gamma(1-\gamma)} + \frac{s_1}{\beta} - \frac{s_1(2 - \gamma s_1/\beta)}{2\beta(1-\gamma)} - 1 + \frac{2-\gamma}{1-\gamma}\right) = c.$$

Together, these yield a system of two equations in the unknowns $s_1$ and $s_2$, whose solution gives a candidate threshold pair $(s_1^\star, s_2^\star)$. We then verify whether this solution satisfies the following consistency conditions:

1. $s_1^\star \leq \beta$ and $s_2^\star > \beta$,

2. Case (v) applies to $(s_1^\star, s_2^\star)$, i.e., $s_2^\star > \gamma$ and $\frac{s_2^\star - \gamma s_1^\star}{1-\gamma} > 1$,

3. Case (vi) applies to $(s_1^\star/\beta, 1)$, i.e., $\frac{1 - \gamma s_1}{1-\gamma} \in [0, 1]$.

If all conditions are met, the thresholds $(s_1^\star, s_2^\star)$ are valid. Otherwise, we proceed to test one of the remaining possibilities.

Unlike the $p = 1$ case, where the equations admit a closed-form solution for $s_1^\star$ and a piecewise formula for $s_2^\star$, the general $p \in (0, 1)$ case yields a coupled nonlinear system that typically does not admit a closed-form solution. Even in the symmetric case $p = 1/2$, the equations remain analytically intractable due to the interaction between the thresholds and regime-specific probability expressions. Numerical solutions for various parameter settings are provided in Appendix L.5.

### L.3 Expressions for metrics

In this section, we give expressions for computing the representation ratio and the observed utility of each of the two institutions.

#### L.3.1 Representation ratio

The representation ratio is defined as the ratio $A/B$, where the numerator $A$ corresponds to the fraction of candidates from the disadvantaged group $G_2$ who are selected, and the denominator $B$ is the corresponding quantity for the advantaged group $G_1$.

The numerator can be written as:

$$\begin{aligned}
A &= (1 - s_{1,\beta}) + (1-p)\left(\Pr[u_{i'1} < s_{1,\beta} \wedge u_{i'2} \geq s_{2,\beta}] - \Pr[u_{i'2} \geq s_{2,\beta}]\right) \\
&\quad + (1-p)\Pr[u_{i'2} \geq s_{2,\beta}] + p\,\Pr[u_{i'1} < s_{1,\beta} \wedge u_{i'2} \geq s_{2,\beta}] \\
&= (1 - s_{1,\beta}) + \Pr[u_{i'1} < s_{1,\beta} \wedge u_{i'2} \geq s_{2,\beta}],
\end{aligned}$$

where the final simplification follows by cancellation of terms.

Similarly, the denominator is:

$$B = (1 - s_1) + \Pr[u_{i1} < s_1 \wedge u_{i2} \geq s_2].$$

Hence, the representation ratio is given by:

$$\mathcal{R}(p, \beta, \gamma) = \frac{(1 - s_{1,\beta}) + \Pr[u_{i'1} < s_{1,\beta} \wedge u_{i'2} \geq s_{2,\beta}]}{(1 - s_1) + \Pr[u_{i1} < s_1 \wedge u_{i2} \geq s_2]}.$$

**Special case: $\gamma = 1$.** We now analyze a simplified setting where $\gamma = 1$ and $p \geq 1/2$ (the case $p < 1/2$ is symmetric). Since more candidates prefer Institution 1, it follows that $s_1^\star(p) \geq s_2^\star(p)$. Under full correlation ($\gamma = 1$), the observed utilities are deterministic functions of the true values, so:

- A candidate is selected if their utility to either institution exceeds the corresponding threshold.

- In this case, both groups face the same selection criterion: being above $s_2^\star(p)$.

Thus, the capacity constraint becomes:

$$(1 - s_2^\star(p)) + \left(1 - \frac{s_2^\star(p)}{\beta}\right) = 2c,$$

which yields:

$$s_2^\star(p) = \frac{2(1-c)}{1 + 1/\beta}.$$

Note that $s_2^\star(p)$ is independent of $p$.

Now consider the assignment to Institution 1. A candidate from $G_1$ is assigned to Institution 1 if:

1. They prefer Institution 1 (which happens with probability $p$), and

2. Their utility exceeds $s_1^\star(p)$. Otherwise, they go to Institution 2 (if their utility exceeds $s_2^\star(p)$).

The capacity constraint for Institution 1 becomes:

$$p(1 - s_1^\star(p)) + p\left(1 - s_1^\star(p)\beta\right) = c,$$

which simplifies to:

$$s_1^\star(p) = \frac{2 - c/p}{1 + 1/\beta}.$$

Since selection depends on whether the candidate's value exceeds $s_2^\star(p)$, the representation ratio becomes:

$$\mathcal{R} = \frac{1 - s_2^\star(p)/\beta}{1 - s_2^\star(p)}.$$

This reflects the differential in admission likelihood due to the biased utility scaling between the two groups.

### L.3.2   Observed utility of institutions

In the general preference setting, the observed utility of Institution 1 (assuming $\nu_1 = \nu_2 = 1$) is given by:

$$\mathcal{U}_1 = p\,\mathbb{E}_{G_{11}}[\widehat{u}_{i1} \cdot \mathbb{1}[\widehat{u}_{i1} \geq s_1^\star]] + (1-p)\,\mathbb{E}_{G_{12}}[\widehat{u}_{i1} \cdot \mathbb{1}[\widehat{u}_{i1} \geq s_1^\star \wedge \widehat{u}_{i2} < s_2^\star]]$$
$$+ p\,\mathbb{E}_{G_{21}}[\widehat{u}_{i'1} \cdot \mathbb{1}[\widehat{u}_{i'1} \geq s_{1,\beta}^\star]] + (1-p)\,\mathbb{E}_{G_{22}}[\widehat{u}_{i'1} \cdot \mathbb{1}[\widehat{u}_{i'1} \geq s_{1,\beta}^\star \wedge \widehat{u}_{i'2} < s_{2,\beta}^\star]], \quad (49)$$

where $G_{i\ell}$ denotes the subset of candidates in group $G_i$ who prefer Institution $\ell$.

To simplify the utility calculation, we define the following integral:

$$I_1(\gamma, s_1, s_2) := \int_0^1 \int_0^1 x \cdot \mathbb{1}[\gamma x + (1 - \gamma)y < s_2 \wedge x \geq s_1]\, dx\, dy.$$

**Proposition L.2** (**Observed utility of Institution 1 in the general preference setting**). *The utility of institution 1, $\mathcal{U}_1$ is given by:*

$$p\left(\frac{1 - s_1^2}{2} + \frac{\beta(1 - s_{1,\beta}^2)}{2}\right) + (1-p)\left(I_1(\gamma, s_1, s_2) + \beta I_1(\gamma, s_{1,\beta}, s_{2,\beta})\right).$$

*Proof.* For a candidate $i \in G_{11}$,

$$\mathbb{E}[\widehat{u}_{i1} \mathbb{1}[\widehat{u}_{i1} \geq s_1]] = \int_{s_1}^1 s\,ds = \frac{(1 - s_1^2)}{2}.$$

For a candidate $i \in G_{12}$, For a candidate $i \in G_{12}$, the definition of the integral $I_1$ shows that $\mathbb{E}[\widehat{u}_{i1} \mathbb{1}[\widehat{u}_{i1} \geq s_1^\star \wedge \widehat{u}_{i2} < s_2^\star]]$ is equal to $I_1(\gamma, s_1, s_2)$. Arguing similarly for the two sub-groups of $G_2$ and using (49) implies the desired result. $\qquad\square$

It remains to evaluate $I_1(\gamma, s_1, s_2)$. This integral can be evaluated based on the following cases:

- **Case I ($s_2 < \gamma s_1$):** In this case, $I_1(\gamma, s_1, s_2) = 0$. For a given $y$, $x$ varies from $s_1$ to $\frac{s_2 - (1-\gamma)y}{\gamma} \leq \frac{s_2}{\gamma} < s_1$. Thus, the range of $x$ is empty for any $y \in [0, 1]$.

- **Case II ($\frac{s_2}{\gamma} \in [s_1, 1]$, $\frac{s_2 - \gamma s_1}{1 - \gamma} \in [0, 1]$):** Note that $x$ varies from $s_1$ to 1. For a given $x$, $y$ varies from 0 to $\min(1, \frac{s_2 - \gamma x}{1 - \gamma})$. Since $x \geq s_1$, $\frac{s_2 - \gamma x}{1 - \gamma} \leq \frac{s_2 - \gamma s_1}{1 - \gamma} \leq 1$. However, $\frac{s_2 - \gamma x}{1 - \gamma} \geq 0$ only if $x \leq s_2/\gamma \in [s_1, 1]$. Thus, the integral is

$$\int_{s_1}^{s_2/\gamma} \frac{s_2 - \gamma x}{1 - \gamma} x\,dx = \frac{1}{1 - \gamma}\left(\frac{s_2^3}{6\gamma^2} - \frac{s_2 s_1^2}{2} + \frac{\gamma s_1^3}{3}\right).$$

- **Case III ($\frac{s_2}{\gamma} \in [s_1, 1]$, $\frac{s_2 - \gamma s_1}{1 - \gamma} > 1$):** The argument here is similar to case II above, $\frac{s_2 - \gamma x}{1 - \gamma}$ may not remain at most 1 for all $x \in [s_1, 1]$. In fact, when $x \leq \frac{s_2 - (1-\gamma)}{\gamma}$, $\min(1, \frac{s_2 - \gamma x}{1 - \gamma}) = 1$, and then $y$ varies from 0 to 1. Thus, the integral is

$$\int_{s_1}^{\frac{s_2 - (1-\gamma)}{\gamma}} x\,dx + \int_{\frac{s_2 - (1-\gamma)}{\gamma}}^{s_2/\gamma} \frac{s_2 - \gamma x}{1 - \gamma} x\,dx = \frac{1}{2}\left(\left(\frac{s_2 - (1-\gamma)}{\gamma}\right)^2 - s_1^2\right)$$

$$+ \frac{1}{1 - \gamma}\left[\frac{s_2^3}{6\gamma^2} - \left(\frac{s_2 - (1-\gamma)}{\gamma}\right)^2\left(\frac{s_2}{6} + \frac{1 - \gamma}{3}\right)\right].$$

- **Case IV ($\frac{s_2}{\gamma} > 1$, $\frac{s_2 - \gamma s_1}{1 - \gamma} \in [0, 1]$):** The argument is again similar to case II, except for the fact that $s_2 - \gamma x$ remains non-negative for all $x \in [s_1, 1]$. Therefore, the desired integral is

$$\int_{s_1}^1 \frac{s_2 - \gamma x}{1 - \gamma} x\,dx = \frac{1}{1 - \gamma}\left[\frac{s_2}{2}(1 - s_1^2) - \frac{\gamma}{3}(1 - s_1^3)\right].$$

- **Case V ($\frac{s_2}{\gamma} > 1$, $\frac{s_2 - \gamma s_1}{1 - \gamma} > 1$):** As $x$ varies from $s_1$ to 1, $\frac{s_2 - \gamma s_1}{1 - \gamma}$ remains non-negative because $s_2 \geq \gamma$, but this expression exceeds 1 when $x$ exceeds $\frac{s_2 - (1-\gamma)}{\gamma}$. Thus, the desired integral is

$$\int_{s_1}^{\frac{s_2 - (1-\gamma)}{\gamma}} x\,dx + \int_{\frac{s_2 - (1-\gamma)}{\gamma}}^1 \frac{s_2 - \gamma x}{1 - \gamma} x\,dx = \frac{1}{2}\left(\left(\frac{s_2 - (1-\gamma)}{\gamma}\right)^2 - s_1^2\right)$$

$$+ \frac{1}{1 - \gamma}\left[\left(\frac{s_2}{2} - \frac{\gamma}{3}\right) - \left(\frac{s_2 - (1-\gamma)}{\gamma}\right)^2\left(\frac{s_2}{6} + \frac{1 - \gamma}{3}\right)\right]$$

We now give the expression for the expected utility of Institution 2. Define the integral:

$$I_2(\gamma, s_1, s_2) := \int_0^1 \int_0^1 (\gamma x + (1 - \gamma)y) \mathbb{1}[\gamma x + (1 - \gamma)y \geq s_2 \wedge x < s_1]\,dx\,dy.$$

Arguing as in the proof of Proposition L.2, we get

**Proposition L.3 (Observed utility of Institution 2 in the general preference setting).** *The utility of Institution 2, $\mathcal{U}_2$ is given by:*

$$(1 - p)\left(I_2(\gamma, 1, s_2) + \beta I_2(\gamma, 1, s_2)\right) + p\left(I_2(\gamma, s_1, s_2) + \beta I_2(\gamma, s_{1,\beta}, s_{2,\beta})\right).$$

Finally, observe that the integral $I_2(\gamma, s_1, s_2)$ is exactly the integral computed in the four cases stated in Appendix H.2. We also need to take care of a fifth possibility when $s_2^\star$ exceeds $s_1^\star \gamma + (1 - \gamma)$ – in this case, $I_2(\gamma, s_1, s_2) = 0$. Thus, we have:

- **Case I ($s_2^\star \leq \min(s_1^\star \gamma, 1 - \gamma)$ ):** In this case, the integral is

$$\frac{s_1^\star(\gamma s_1^\star + 1 - \gamma)}{2} - \frac{(s_2^\star)^3}{3\gamma(1 - \gamma)}.$$

- **Case II ($1 - \gamma + s_1^\star \gamma \geq s_2^\star \geq \max(s_1^\star \gamma, 1 - \gamma)$):** The integral is

$$\frac{1}{3}(1 - \gamma + 2s_2^\star + \gamma s_1^\star) \cdot \frac{(1 - \gamma - s_2^\star + \gamma s_1^\star)^2}{2\gamma(1 - \gamma)}.$$

- **Case III ($s_1^\star \gamma < s_2^\star < 1 - \gamma$):** The integral is

$$s_1^\star \cdot \left(1 - \frac{s_2^\star}{1 - \gamma}\right)\left(\frac{\gamma s_1^\star}{2} + \frac{1 - \gamma + s_2^\star}{2}\right) + \frac{s_1^\star}{2} \cdot \frac{\gamma s_1^\star}{1 - \gamma} \cdot \left(\frac{2\gamma s_1^\star}{3} + \frac{3s_2^\star - \gamma s_1^\star}{3}\right).$$

- **Case IV ($1 - \gamma < s_2^\star < s_1^\star \gamma$):** The integral is

$$\left(s_1^\star - \frac{s_2^\star}{\gamma}\right) \cdot \left(\frac{\gamma s_1^\star + s_2^\star}{2} + \frac{1 - \gamma}{2}\right) + \frac{(1 - \gamma)}{2\gamma} \cdot \left(\frac{3s_2^\star - (1 - \gamma)}{3} + \frac{2(1 - \gamma)}{3}\right).$$

- **Case V ($s_2^\star > s_1^\star \gamma + (1 - \gamma)$):** In this case, the integral is $0$.

## L.4  Monotonicity properties of $s_1^\star$ and $s_2^\star$ with respect to $p$

In this section, we show that for any fixed values of the parameters $c$, $\beta$, and $\gamma$, the thresholds $s_1^\star$ and $s_2^\star$ are monotone functions of the parameter $p$. More formally, we show:

**Theorem L.4** (**Monotonicity of thresholds w.r.t. $p$**). *For any fixed values of $c, \beta, \gamma$, the thresholds $s_1^\star(p)$ and $s_2^\star(p)$ are non-decreasing and non-increasing functions of $p$ respectively.*

*Proof.* The equations for solving $s_1^\star(p)$ and $s_2^\star(p)$ can be written as follows:

$$p(\Pr[u_{i1} \geq s_1] + \Pr[u_{i'1} \geq s_{1,\beta}]) + (1 - p)(\Pr[u_{i1} \geq s_1 \wedge u_{i2} < s_2]$$
$$+ \Pr[u_{i'1} \geq s_{1,\beta} \wedge u_{i'2} < s_{2,\beta}]) = c_1, \tag{50}$$
$$p(\Pr[u_{i1} < s_1 \wedge u_{i2} \geq s_2] + \Pr[u_{i'1} < s_{1,\beta} \wedge u_{i'2} \geq s_{2,\beta}])$$
$$+ (1 - p)(\Pr[u_{i2} \geq s_2] + \Pr[u_{i'2} \geq s_{2,\beta}]) = c_2. \tag{51}$$

Since the set of selected candidates from $G_1$ is exactly those for which $u_{i1} \geq s_1$ or $u_{i2} \geq s_2$ and similarly for $G_2$, we get

$$\Pr[u_{i1} \geq s_1 \vee u_{i2} \geq s_2] + \Pr[u_{i'1} \geq s_{1,\beta} \vee u_{i'2} \geq s_{2,\beta}] = c_1 + c_2 \tag{52}$$

Note that the l.h.s. of the above equation does not depend on $p$ explicitly. A formal proof of the above statement can be given by adding (50) and (51). Indeed, the sum of the terms involving candidate $i$ in the two equations is:

$$p\Pr[u_{i1} \geq s_1] + (1 - p)\Pr[u_{i1} \geq s_1 \wedge u_{i2} < s_2] + p\Pr[u_{i1} < s_1 \wedge u_{i2} \geq s_2] + (1 - p)\Pr[u_{i2} \geq s_2]$$
$$= \Pr[u_{i1} \geq s_1] - (1 - p)\Pr[u_{i1} \geq s_1 \wedge u_{i2} \geq s_2] + \Pr[u_{i2} \geq s_2] - p\Pr[u_{i1} \geq s_1 \wedge u_{i2} \geq s_2]$$
$$= Pr[u_{i1} \geq s_1] + \Pr[u_{i2} \geq s_2] - \Pr[u_{i1} \geq s_1 \wedge u_{i2} \geq s_2] = \Pr[u_{i1} \geq s_1 \vee u_{i2} \geq s_2].$$

Arguing similarly for $i'$, we get (52). Given values $p, s_1, s_2$, let $f(p, s_1, s_2)$ and $g(s_1, s_2)$ denote the l.h.s. of (50) and (52) respectively, i.e., $f(p, s_1, s_2) :=$

$$p(\Pr[u_{i1} \geq s_1] + \Pr[u_{i'1} \geq s_{1,\beta}]) + (1 - p)(\Pr[u_{i1} \geq s_1 \wedge u_{i2} < s_2] + \Pr[u_{i'1} \geq s_{1,\beta} \wedge u_{i'2} < s_{2,\beta}]),$$

and

$$g(s_1, s_2) := \Pr[u_{i1} \geq s_1 \vee u_{i2} \geq s_2] + \Pr[u_{i'1} \geq s_{1,\beta} \vee u_{i'2} \geq s_{2,\beta}].$$

We now show monotonicity properties of these functions.

**Fact L.5.** *Let $s_1, s_1', s_2, s_2' \in [0,1]$ such that $s_1 \leq s_1'$ and $s_2 \leq s_2'$. Then, $g(s_1, s_2) \leq g(s_1', s_2')$. Further, $g(s_1, s_2) < g(s_1', s_2')$ if $s_1 < s_1'$ and $s_2 < s_2'$.*

*Proof.* The first statement follows from the fact that $\Pr[u_{i1} \geq s_1 \vee u_{i2} \geq s_2]$ and $\Pr[u_{i'1} \geq s_{1,\beta} \vee u_{i'2} \geq s_{2,\beta}]$ are non-increasing functions of $s_1$ and $s_2$.

For the second statement, assume $s_1 < s_1'$ and $s_2 < s_2'$. Observe that

$$\Pr[u_{i1} \geq s_1 \vee u_{i2} \geq s_2] - \Pr[u_{i1} \geq s_1' \vee u_{i2} \geq s_2']$$
$$= \Pr[s_1 \leq u_{i1} \leq s_1' \wedge u_{i2} \leq s_2'] + \Pr[u_{i1} \leq s_1' \wedge s_2 \leq u_{i2} \leq s_2'].$$

Now, if $\Pr[u_{i1} \leq s_1' \wedge s_2 \leq u_{i2} \leq s_2'] = 0$, then it must be the case that $u_{i2} \leq s_2$ whenever $u_{i1} \leq s_1'$, i.e., $s_2 \geq s_1' \gamma + (1 - \gamma)$. But then, $u_{i2} \leq s_2'$ whenever $u_{i1} \leq s_1'$. Thus,

$$\Pr[s_1 \leq u_{i1} \leq s_1' \wedge u_{i2} \leq s_2'] = \Pr[s_1 \leq u_{i1} \leq s_1'] > 0.$$

This shows that $g(s_1, s_2) < g(s_1', s_2')$. $\qquad\square$

Now we show the monotonicity properties of the function $f$.

**Fact L.6.** *Let $s_1, s_1', s_2, s_2', p, p' \in [0,1)$ such that $s_1 \geq s_1', s_2 \leq s_2', p \leq p'$. Then, $f(p, s_1, s_2) \leq f(p', s_1', s_2')$. Further,*

(i) $f(p, s_1, s_2) < f(p', s_1', s_2')$ *if $p < p'$.*

(ii) $f(p, s_1, s_2) < f(p', s_1', s_2')$ *if $s_1 > s_1'$.*

*Proof.* We first show monotonicity with respect to $p$. Note that

$$p \Pr[u_{i1} \geq s_1] + (1 - p)\left(\Pr[u_{i1} \geq s_1] - \Pr[u_{i1} \geq s_1, u_{i2} \geq s_2]\right)$$
$$= \Pr[u_{i1} \geq s_1] - (1 - p) \Pr[u_{i1} \geq s_1, u_{i2} \geq s_2]$$

Similarly,

$$p \Pr[u_{i'1} \geq s_{1,\beta}] + (1 - p)\left(\Pr[u_{i'1} \geq s_{1,\beta}] - \Pr[u_{i'1} \geq s_{1,\beta}, u_{i'2} \geq s_{2,\beta}]\right)$$
$$= \Pr[u_{i'1} \geq s_{1,\beta}] - (1 - p) \Pr[u_{i'1} \geq s_{1,\beta}, u_{i'2} \geq s_{2,\beta}]$$

Thus,

$$f(p', s_1, s_2) - f(p, s_1, s_2) \geq (p' - p) \Pr[u_{i1} \geq s_1 \wedge u_{i2} \geq s_2] > 0. \tag{53}$$

Now we show monotonicity with respect to $s_1$: since $\Pr[u_{i1} \geq s_1]$, $\Pr[u_{i1} \geq s_1, u_{i2} < s_2]$, $\Pr[u_{i'1} \geq s_{1,\beta}]$, $\Pr[u_{i'1} \geq s_{1,\beta}, u_{i'2} < s_{2,\beta}]$, are non-increasing functions of $s_1$ (and the first one is strictly increasing), we see (using the definition of $f$) that

$$f(p', s_1', s_2) \geq f(p', s_1, s_2), \tag{54}$$

with the inequality being strict if $s_1' < s_1$. The monotonicity with respect to $s_2$ follows similarly: $\Pr[u_{i1} \geq s_1, u_{i2} < s_2], \Pr[u_{i'1} \geq s_{1,\beta}, u_{i'2} < s_{2,\beta}]$ are non-decreasing functions of $s_2$, and hence,

$$f(p', s_1', s_2') \geq f(p', s_1', s_2). \tag{55}$$

The desired statements now follow by combining (53), (54) and (55). $\qquad\square$

We are now ready to show the monotonicity property of $s_1^\star$. Let $0 \leq p < p' \leq 1$, and assume for the sake of contradiction that $s_1^\star(p) < s_1^\star(p')$. Two cases arise:

(i) $s_2^\star(p) < s_2^\star(p')$: It follows by Fact L.5 that $g(s_1^\star(p), s_2^\star(p)) < g(s_1^\star(p'), s_2^\star(p'))$, which is a contradiction because both of these quantities must be equal to $c_1 + c_2$ (using (52)).

(ii) $s_2^\star(p) \geq s_2^\star(p')$: Using Fact L.6 (and part (ii) of result),

$$f(p, s_1^\star(p), s_2^\star(p)) < f(p', s_1^\star(p'), s_2^\star(p')),$$

which is again a contradiction because both of the above expressions should be equal to $c_1$ (using (50)).

Thus, we see that neither of the two cases above can happen and hence, $s_1^\star(p) \geq s_1^\star(p')$. The monotonicity property of $s_2^\star(p)$ can be shown similarly by analyzing the properties of the function $h(p, s_1, s_2)$ that denotes the l.h.s. of (51). $\square$

We note a useful corollary of this result:

**Corollary L.7.** *Given parameters $c, \beta, \gamma$ and $p, p' \in [0, 1]$, it cannot happen that $s_1^\star(p) = s_1^\star(p')$ and $s_2^\star(p) = s_2^\star(p')$.*

*Proof.* Assume w.l.o.g. that $p < p'$. Assume for the sake of contradiction that $s_1^\star(p) = s_1^\star(p')$ and $s_2^\star(p) = s_2^\star(p')$. Then, Fact L.6 shows that

$$f(p', s_1^\star(p'), s_2^\star(p')) > f(p, s_1^\star(p'), s_2^\star(p')) = f(p, s_1^\star(p), s_2^\star(p)),$$

which is a contradiction because both $f(p, s_1^\star(p), s_2^\star(p))$ and $f(p', s_1^\star(p'), s_2^\star(p'))$ are equal to $c_1$. $\square$

### L.4.1 Strict monotonicity properties of the thresholds with respect to $p$

Theorem L.4 showed that $s_1^\star(p)$ and $s_2^\star(p)$ are non-decreasing and non-increasing functions of $p$ respectively. In this section, we show strict monotonicity properties of these thresholds.

**Proposition L.8 (Strict monotonicity of $s_1^\star$ w.r.t. $p$).** *For any fixed values of the parameters $c, \beta, \gamma$, there exists a value $p^\star \in [0, 1]$ such that $s_1^\star(p)$ is strictly increasing for $p \in [p^\star, 1]$ and does not vary with $p$ for $p \in [0, p^\star)$. Assuming $\beta \geq 1 - (c_1 + c_2)$, $s_1^\star(p) = \frac{2 - c_1 - c_2}{1 + 1/\beta}$ for all $p \in [0, p^\star)$ and $s_2^\star(p)$ is strictly decreasing in $[0, p^\star)$. Further, $p^\star > 0$ iff $s_2^\star(0) > \gamma s_1^\star(0) + (1 - \gamma)$.*

We first prove an auxiliary result.

**Fact L.9.** *For any fixed values of the parameters $\gamma, s_1, s_2 \in (0, 1)$, $\Pr[u_{i1} \leq s_1 \wedge u_{i2} \geq s_2] = 0$ iff $s_2 \geq \gamma s_1 + (1 - \gamma)$.*

*Proof.* Suppose $\Pr[u_{i1} \leq s_1 \wedge u_{i2} \geq s_2] = 0$. Assume for the sake of contradiction that $s_2 < \gamma s_1 + (1 - \gamma)$. For the sake of concreteness, let $\varepsilon := s_1 + (1 - \gamma) - s_2$. Recall that $u_{i1} = v_{i1}$ and $u_{i2} = \gamma v_{i1} + (1 - \gamma) v_{i2}$ where $v_{i1}$ and $v_{i2}$ are independent and distributed uniformly in $[0, 1]$. Now if $v_{i1} \in [\max(0, s_1 - \varepsilon), s_1]$ and $v_{i2} \in [1 - \varepsilon, 1]$, then

$$u_{i2} \geq \gamma(s_1 - \varepsilon) + (1 - \gamma)(1 - \varepsilon) = \gamma s_1 + (1 - \gamma) - \varepsilon = s_2.$$

Since $s_1 > 0$, we see that $\Pr[u_{i1} \leq s_1 \wedge u_{i2} \geq s_2] = 0$, a contradiction.

For the converse, suppose $s_2 \geq s_1 \gamma + (1 - \gamma)$. If $u_{i1} \leq s_1$, then $u_{i2} = \gamma v_{i1} + (1 - \gamma) v_{i2}$ can exceed $s_2$ only when $v_{i2} \geq 1$, which is a zero probability event. $\square$

We now complete the proof of Proposition L.8. Define

$$I := \{p \in [0, 1] : s_2^\star(p) \geq \gamma s_1^\star(p) + 1 - \gamma\}.$$

Theorem L.4 shows that if $I$ is non-empty, then it is a closed interval of $[0, 1]$. Assuming $I$ is non-empty, let it be of the form $[0, p^\star]$ (in case $I$ is empty, let $p^\star = 0$). We now show that $s_1^\star(p)$ does not vary with $p$ when $p < p^\star$.

**Fact L.10.** *Assuming $p^\star > 0$, $s_1^\star(p)$ does not vary with $p$ for $p \in [0, p^\star]$. If $\beta \geq 1 - (c_1 + c_2)$, $s_1^\star(p) = \frac{2(1-c)}{1 + 1/\beta}$ for all $p \in [0, p^\star]$. Further, $s_2^\star(p)$ is a strictly decreasing function of $p \in [0, p^\star]$.*

*Proof.* Suppose $p < p^\star$. Then Fact L.9 shows that $\Pr[u_{i1} \le s_1 \wedge u_{i2} \ge s_2] = 0$. We show that

$$s_{2,\beta}^\star(p) \ge \gamma s_{1,\beta}^\star(p) + 1 - \gamma$$

as well. Indeed, if $s_{2,\beta}^\star(p) = 1$, this follows trivially. Hence, assume $s_{2,\beta}^\star(p) = \frac{s_2^\star(p)}{\beta}$. Then

$$s_{2,\beta}^\star(p) \ge \frac{\gamma s_1^\star(p) + 1 - \gamma}{\beta} \ge s_{1,\beta}^\star(p) + 1 - \gamma.$$

It follows by Fact L.9 that $\Pr[u_{i'1} \le s_{1,\beta} \wedge u_{i'2} \ge s_{2,\beta}] = 0$ also. Thus, (52) becomes:

$$\Pr[u_{i1} \ge s_1^\star(p)] + \Pr[u_{i'1} \ge s_{1,\beta}^\star(p)] = c_1 + c_2.$$

In other words,

$$(1 - s_1^\star(p)) + (1 - s_{1,\beta}^\star(p)) = c_1 + c_2.$$

First assume that $s_1^\star(p) \ge \beta$. Then the above equation has the solution:

$$s_1^\star(p) = 1 - (c_1 + c_2).$$

Thus $s_1^\star(p)$ does not vary with $p$. If $\beta \ge 1 - (c_1 + c_2)$, this case cannot occur. Thus, the above equation becomes:

$$(1 - s_1^\star(p)) + (1 - s_1^\star(p)/\beta) = c_1 + c_2,$$

which has the solution:

$$s_1^\star(p) = \frac{2 - (c_1 + c_2)}{1 + 1/\beta}.$$

Thus, $s_1^\star(p)$ does not vary with $p$ in $[0, p^\star]$. The fact that $s_2^\star(p)$ strictly decreases in this interval follows from Corollary L.7.

$\square$

Thus we have shown the statements in Proposition L.8 when $p \in [0, p^\star]$. We now show that $s_1^\star(p)$ is a strictly increasing function of $p$ for $p > p^\star$. We first need the following extension of Fact L.9:

**Fact L.11.** *Consider $s_1 \in [0,1]$ and $s_2, s_2' \in [0,1]$ such that $s_2 < s_2' \le s_1 \gamma + (1 - \gamma)$. Then, $\Pr[u_{i1} < s_1 \wedge s_2 \le u_{i2} < s_2'] > 0$.*

*Proof.* First, consider the case when $s_2' \le 1 - \gamma$. Let $\varepsilon > 0$ be a small enough positive constant (whose value will be fixed later). Suppose $v_{i1} \in [0, \frac{\varepsilon}{\gamma}]$ and $v_{i2} \in [\frac{s_2' - 2\varepsilon}{1-\gamma}, \frac{s_2' - \varepsilon}{1-\gamma})$. Here $\varepsilon$ is such that $\frac{\varepsilon}{\gamma} < s_1$, $\frac{s_2' - 2\varepsilon}{1-\gamma} \ge s_2$. It is easy to verify that $u_{i2} = \gamma v_{i1} + (1 - \gamma)v_{i2} \in [s_2, s_2')$. Thus, $\Pr[u_{i1} < s_1 \wedge s_2 \le u_{i2} < s_2'] > 0$.

Now consider the case when $s_2' > 1 - \gamma$. Express $s_2'$ as $\alpha\gamma + (1 - \gamma)$ for some $\alpha \le s_1$. A similar argument as above applies with $v_{i1} \in [\alpha - \varepsilon, \alpha]$ and $v_{i2} \in [1 - \varepsilon, 1)$. $\square$

**Fact L.12.** *The threshold $s_1^\star(p)$ is a strictly increasing function of $p$ for $p \in [p^\star, 1]$.*

*Proof.* Consider values $p, p' \in [p^\star, 1]$ and assume $p < p'$. Assume for the sake of contradiction that $s_1^\star(p) = s_1^\star(p')$. Let $s_1^\star$ denote this common value. Constraint (52) can be expressed as:

$$\Pr[u_{i1} \ge s_1] + \Pr[u_{i1} < s_1 \wedge u_{i2} \ge s_2] + \Pr[u_{i'1} \ge s_{1,\beta}] + \Pr[u_{i'1} < s_{1,\beta} \wedge u_{i'2} \ge s_{2,\beta}] = c_1 + c_2.$$

Substituting $(s_1^\star, s_2^\star(p))$ and $(s_1^\star, s_2^\star(p'))$, and equating the two equations, we get (let $s_{1,\beta}^\star$ denote $\min(1, s_1^\star/\beta)$):

$$\Pr[u_{i1} < s_1^\star \wedge u_{i2} \ge s_2^\star(p)] + \Pr[u_{i'1} < s_{1,\beta}^\star \wedge u_{i'2} \ge s_{2,\beta}^\star(p)]$$
$$= \Pr[u_{i1} < s_1^\star \wedge u_{i2} \ge s_2^\star(p')] + \Pr[u_{i'1} < s_{1,\beta}^\star \wedge u_{i'2} \ge s_{2,\beta}^\star(p')].$$

We know from Theorem L.4 and Corollary L.7 that $s_2^\star(p) > s_2^\star(p')$. Therefore, the above can be written as:

$$\Pr[u_{i1} < s_1^\star \wedge s_2^\star(p') \le u_{i2} < s_2^\star(p)] + \Pr[u_{i'1} < s_{1,\beta}^\star \wedge s_{2,\beta}^\star(p') \le u_{i'2} < s_{2,\beta}^\star(p)] = 0.$$

This implies that $\Pr[u_{i1} < s_1^\star \wedge s_2^\star(p') \le u_{i2} < s_2^\star(p)] = 0$, but this is a contradiction because of Fact L.11. Thus, $s_1^\star(p) < s_1^\star(p')$. $\square$

We now show the last statement in Proposition L.8:

**Fact L.13.** *The quantity $p^\star$ is strictly positive iff $s_2^\star(0) > \gamma s_1^\star(0) + (1 - \gamma)$.*

*Proof.* Suppose $p^\star > 0$. Then $0 \in I$ and hence, $s_2^\star(0) \geq \gamma s_1^\star(0) + (1 - \gamma)$. Assume for the sake of contradiction that $s_2^\star(0) = \gamma s_1^\star(0) + (1 - \gamma)$. Let $p \in (0, p^\star)$. We know from Theorem L.4 that $s_2^\star(p) \leq s_2^\star(0)$ and $s_1^\star(p) \geq s_1^\star(0)$, and at least one of these inequalities must be strict (using Corollary L.7). But then $s_2^\star(p) < \gamma s_1^\star(p) + (1 - \gamma)$, a contradiction. Therefore, $s_2^\star(0) > \gamma s_1^\star(0) + (1 - \gamma)$.

For the converse, assume $s_2^\star(0) > \gamma s_1^\star(0) + (1 - \gamma)$. Then continuity of $s_1^\star(p)$ and $s_1^\star(p)$ shows that $p^\star > 0$. $\qquad\square$

This completes the proof of Proposition L.8.

**Calculating $p^\star$.** Assume that $s_2^\star(0) > \gamma s_1^\star(0) + (1 - \gamma)$. Then monotonicity of $s_1^\star(p)$ and $s_2^\star(p)$ (using Theorem L.4) shows that $p^\star$ satisfies:

$$s_2^\star(p^\star) = \gamma s_1^\star(p^\star) + (1 - \gamma).$$

Now, Proposition L.8 shows that

$$s_2^\star(p^\star) = \frac{\gamma(2 - c_1 - c_2)}{1 + 1/\beta} + (1 - \gamma). \tag{56}$$

Since $\Pr[u_{i2} \geq s_2^\star(p^\star) \wedge u_{i1} < s_1^\star(p^\star)] = 0$ and similarly for $i'$, we can express Equation (51) as:

$$(1 - p^\star)(\Pr[u_{i2} \geq s_2^\star(p^\star)] + \Pr[u_{i'2} \geq s_{2,\beta}^\star(p^\star)]) = c_2.$$

Equation (56) shows that $s_2^\star(p^\star) > 1 - \gamma$ and hence, $s_{2,\beta}^\star(p^\star) > 1 - \gamma$. Now, three cases arise:

- Here, $s_2^\star(p^\star) \geq \gamma$. Then $s_{2,\beta}^\star(p^\star) \geq \gamma$ as well. Then, $p^\star$ is given by

$$1 - p^\star = \frac{2c_2\gamma(1 - \gamma)}{(1 - s_2^\star(p^\star))^2 + (1 - s_{2,\beta}^\star(p^\star))^2}.$$

- Here $s_2^\star(p^\star) < \gamma$ but $s_{2,\beta}^\star(p^\star) \geq \gamma$. Then, $p^\star$ is given by

$$1 - p^\star = \frac{c_2}{\frac{(1 - s_2^\star(p^\star))^2}{2\gamma(1 - \gamma)} + 1 - \frac{s_{2,\beta}^\star(p^\star)}{\gamma} + \frac{1 - \gamma}{2\gamma}}.$$

- Here $s_{2,\beta}^\star < \gamma$ and hence $s_2^\star(p^\star) < \gamma$ as well. Then, $p^\star$ is given by

$$1 - p^\star = \frac{c_2}{2 - \frac{s_{2,\beta}^\star(p^\star) + s_2^\star(p^\star)}{\gamma} + \frac{1 - \gamma}{\gamma}}.$$

So far, we have discussed the strict monotonicity properties of $s_1^\star(p)$. One can analogously show the following lemma regarding the strict monotonicity of $s_2^\star(p)$. We omit the proof.

**Proposition L.14 (Strict monotonicity of $s_2^\star$ w.r.t. $p$).** *For any fixed values of the parameters $c, \beta, \gamma$, there exists a value $q^\star \in [0, 1]$ such that $s_2^\star(p)$ is strictly decreasing for $p \in [0, q^\star]$ and does not vary with $p$ for $p \in (q^\star, 1]$. Assuming $q^\star < 1$, $s_2^\star(p)$, $p \in (q^\star, 1]$, is given by the following equation: $\Pr[u_{i2} \geq s_2] + \Pr[u_{i'2} \geq s_{2,\beta}] = c_1 + c_2$. Further, $q^\star < 1$ iff $\gamma s_1^\star(1) > s_2^\star(1)$.*

## L.5 Numerical results on behavior across preference and correlation parameters

This section numerically examines how equilibrium quantities respond to changes in the candidate preference parameter $p$ and the evaluator correlation parameter $\gamma$. Figures 8–12 present four key quantities across varying parameter settings: (i) the equilibrium thresholds $s_1^\star$ and $s_2^\star$, (ii) the representation ratio $\mathcal{R}$, (iii) the normalized representation ratio $\mathcal{N}$, and (iv) the utilities $\mathcal{U}_1$ and $\mathcal{U}_2$ achieved by the two institutions. In addition, Figure 14 explores how fairness metrics vary with evaluator correlation $\gamma$, for a fixed preference weight $p$ and varying levels of group bias $\beta$.

**Effect of preference alignment ($p$).**   Figures 8–11 fix $c = 0.2$ and vary $p$ for different values of $\beta$ and $\gamma$.

- In Figure 8 ($\beta = 0.8, \gamma = 0.9$), increased preference alignment with Institution 1 leads to widening disparity: $s_1^\star$ rises, $s_2^\star$ falls (consistent with Theorem L.4, Proposition L.8,Proposition L.14), and both $\mathcal{R}$ and $\mathcal{N}$ drop significantly monotonically. Institution 1's utility rises with $p$, while Institution 2's utility declines (both monotonically).

- In Figure 9 ($\beta = 1.0, \gamma = 0.9$), evaluations are unbiased. Representation ratios remain flat at 1, and utilities vary symmetrically with $p$, highlighting fairness preservation under bias-free conditions.

- Figure 10 ($\beta = 1.0, \gamma = 0.0$) shows a similar fairness pattern: no correlation and no bias yield uniform thresholds and balanced utilities, with representation unaffected by preferences.

- Figure 11 ($\beta = 0.8, \gamma = 0.8$) exhibits gradual threshold divergence and monotonic decline in $\mathcal{R}$ and $\mathcal{N}$, indicating compounding effects of preference, bias, and correlation.

**Effect of selectivity increase.**   Figure 12 repeats the previous setting with $c = 0.5$, revealing sharper phase transitions: $\mathcal{R}, \mathcal{N}$ fall rapidly beyond a critical value of $p$, and the utility gap between institutions grows more pronounced. This demonstrates how institutional capacity amplifies the effect of bias under correlated evaluations.

**Effect of correlation ($\gamma$).**   Figure 13 (for $\beta = 0.8, p = 0.6$) highlights the non-monotonic impact of increasing $\gamma$. As the correlation grows:

- The selection threshold $s_2^\star(\gamma)$ initially decreases and then increases, resulting in a convex-shaped utility curve for Institution 2, consistent with the behavior described in Theorem E.1.

- Both the representation ratio $\mathcal{R}(\gamma)$ and the normalized selection share $\mathcal{N}(\gamma)$ increase monotonically, as established in Corollary 4.1. In addition, Figure 14 shows that for any fixed $\gamma$ and $p$, the representation ratio $\mathcal{R}(\gamma)$ and the normalized selection share $\mathcal{N}(\gamma)$ increase monotonically with $\beta$ – this pattern closely mirrors the trend observed in Figure 2.

**First-choice ratio parameter.**   When $p < 1$, we can further define a fairness metric called the *first-choice ratio*, denoted by $\mathcal{F}$, which measures the relative likelihood that candidates from the two groups receive their first-choice institution. Formally, for given parameters $p, c, \beta, \gamma$, let $f_j$ denote the fraction of candidates from group $G_j$ who are assigned their first-choice institution under the stable assignment, i.e., given the stable matching thresholds $(s_1^\star, s_2^\star)$,

$$f_j := p \Pr[\hat{u}_{i_j 1} \geq s_1^\star] + (1 - p) \Pr[\hat{u}_{i_j 2} \geq s_2^\star],$$

where $i_j$ denotes a candidate from group $G_j$. Then, $\mathcal{F} := f_2/f_1$. This metric was also studied in the centralized selection setting by [15].

Figure 15 illustrates how $\mathcal{F}$ varies with $p$ for different values of $\beta$ and $\gamma$ when $c = 0.3$. We find that $\mathcal{F}$ is typically unimodal, attaining its maximum when $p$ is near $1/2$. At this point, the top-ranking candidates from $G_1$ are evenly distributed across both institutions, allowing a larger fraction of $G_2$ candidates to secure their most preferred institution. As expected, increasing $\beta$ reduces the bias against $G_2$ and consequently increases $\mathcal{F}$.

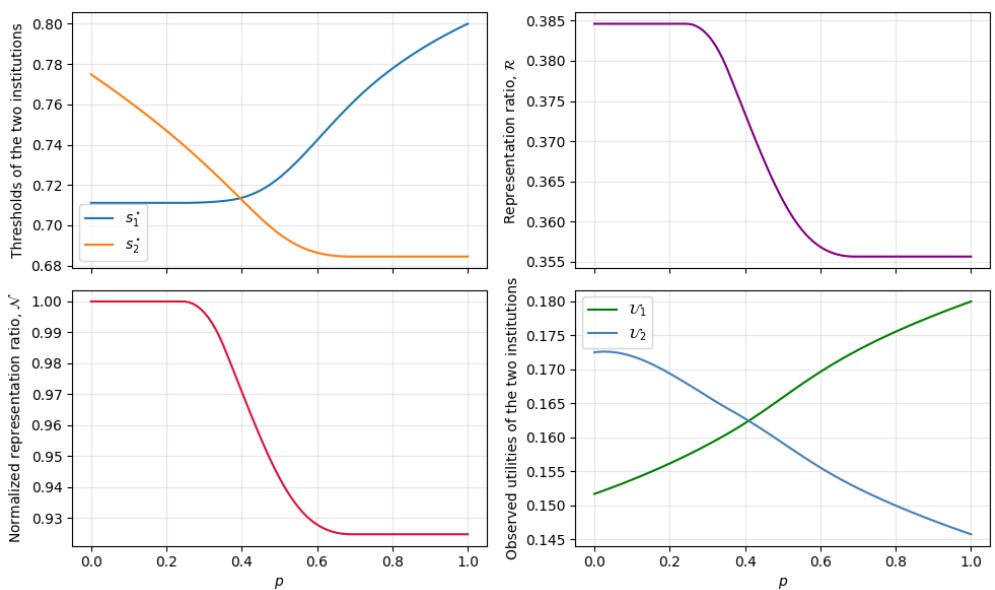

Figure 8: Variation of thresholds, representation ratio, normalized representation ratio and utilities with $p$ for a fixed value of $c = 0.2$, $\beta = 0.8$, and $\gamma = 0.9$.

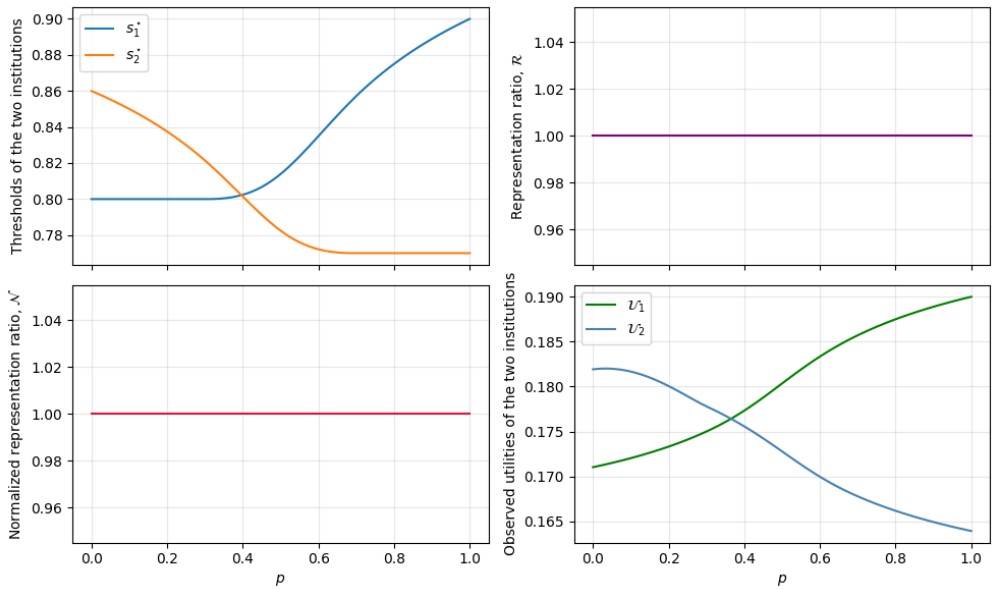

Figure 9: Variation of thresholds, representation ratio, normalized reference ratio, and utilities with $p$ for a fixed value of $c = 0.2$, $\beta = 1$, and $\gamma = 0.9$.

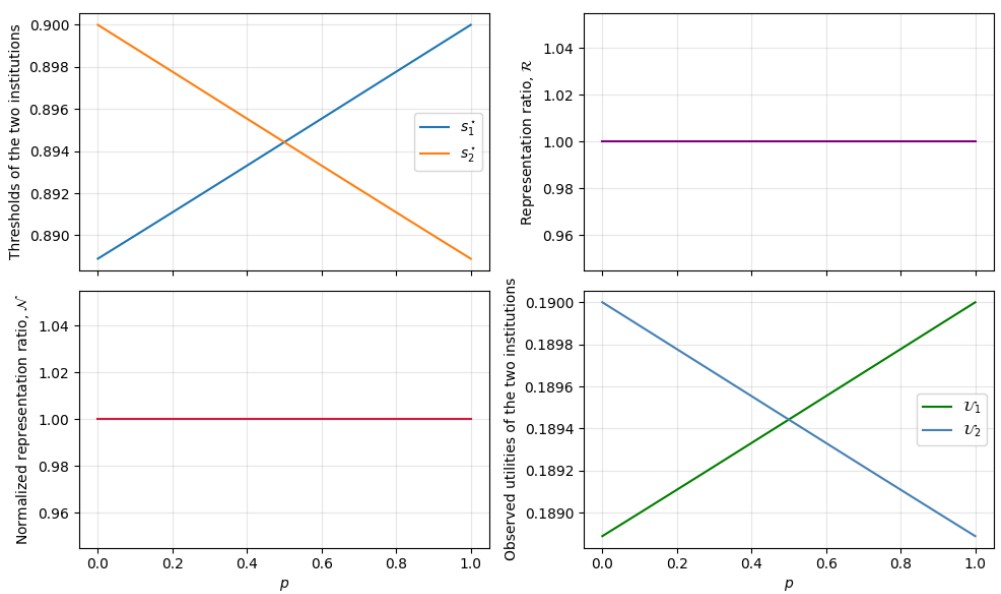

Figure 10: Variation of thresholds, representation ratio, normalized representation ratio, and utilities with $p$ for a fixed value of $c = 0.2$, $\beta = 1$, and $\gamma = 0$.

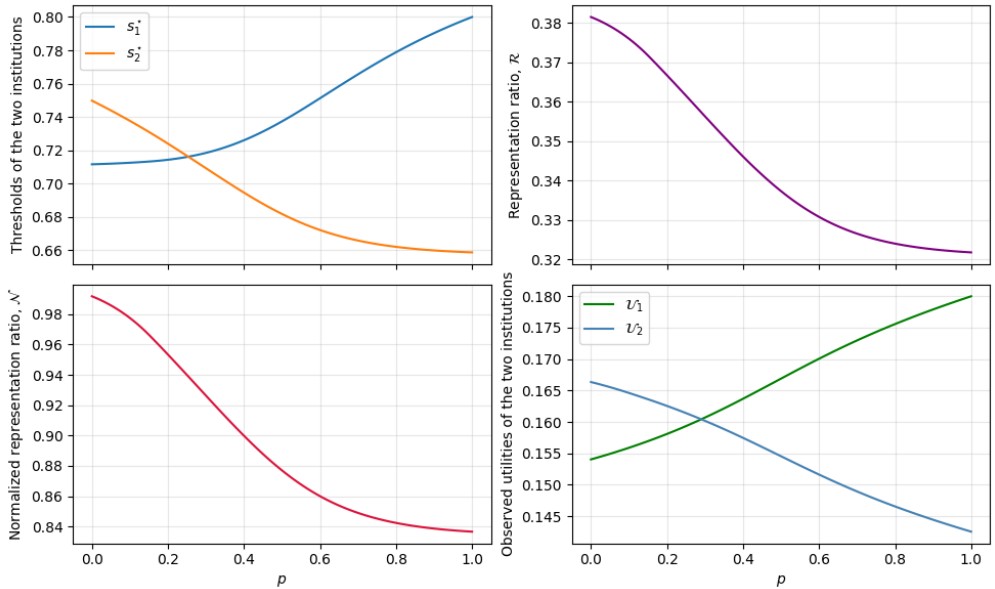

Figure 11: Variation of thresholds, representation ratio, normalized representation ratio, and utilities with $p$ for a fixed value of $c = 0.2$, $\beta = 0.8$, and $\gamma = 0.8$.

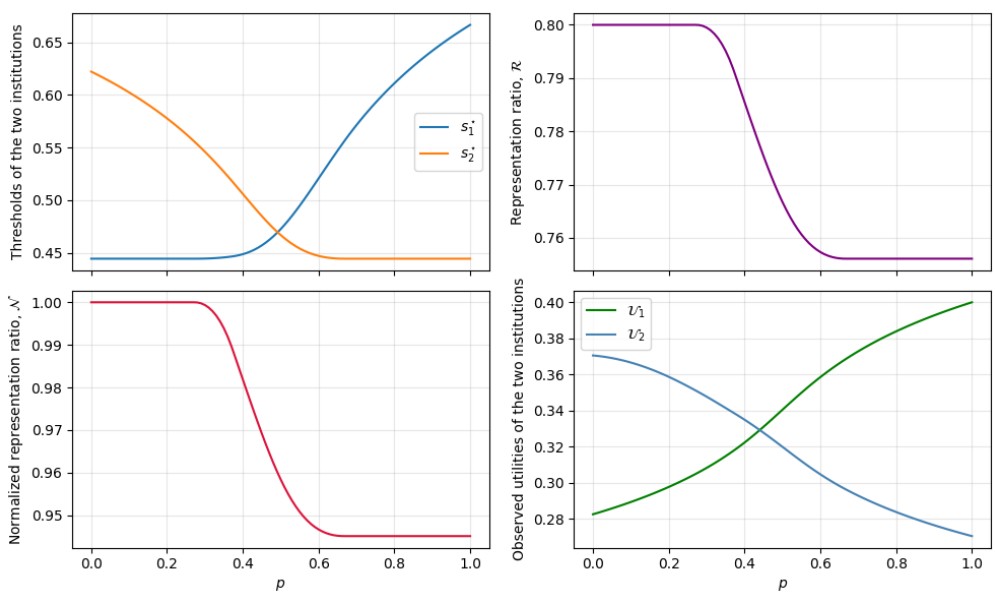

Figure 12: Variation of thresholds, representation ratio, normalized representation ratio, and utilities with $p$ for a fixed value of $c = 0.5$, $\beta = 0.8$, and $\gamma = 0.8$.

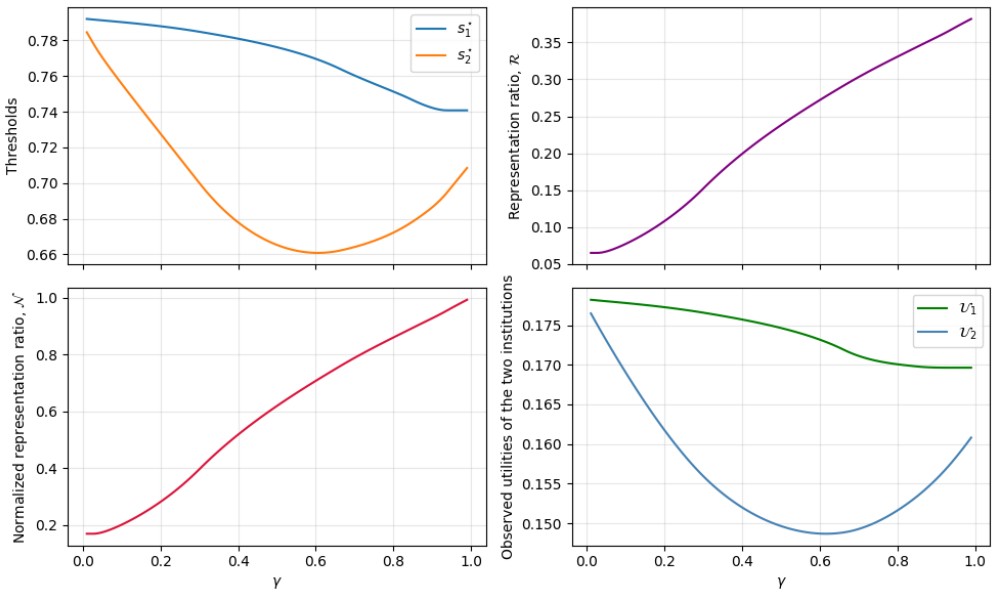

Figure 13: Variation of thresholds, representation ratio, normalized representation ratio, and utilities with $\gamma$ for a fixed value of $c = 0.2$, $\beta = 0.8$, and $p = 0.6$.

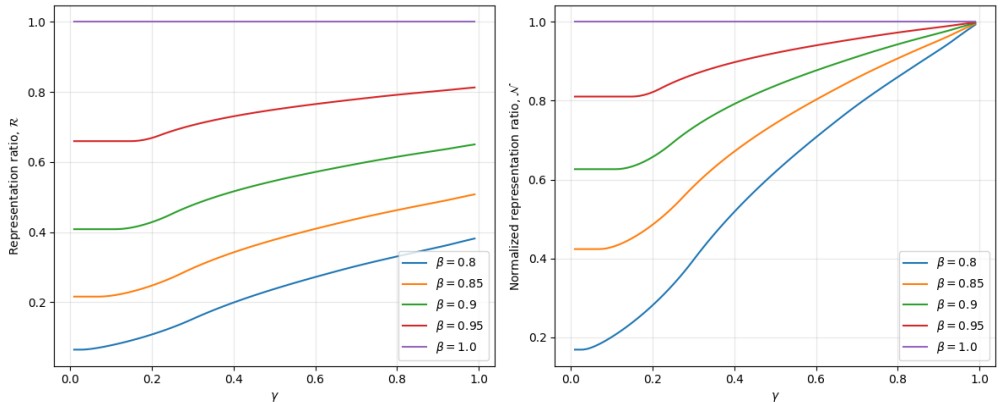

Figure 14: Variation of representation ratio $\mathcal{R}$ and normalized representation ratio $\mathcal{N}$ with $\gamma$ for different values of $\beta \in \{0.8, 0.85, 0.9, 0.95, 1.0\}$ at fixed $c = 0.2$ and $p = 0.6$.

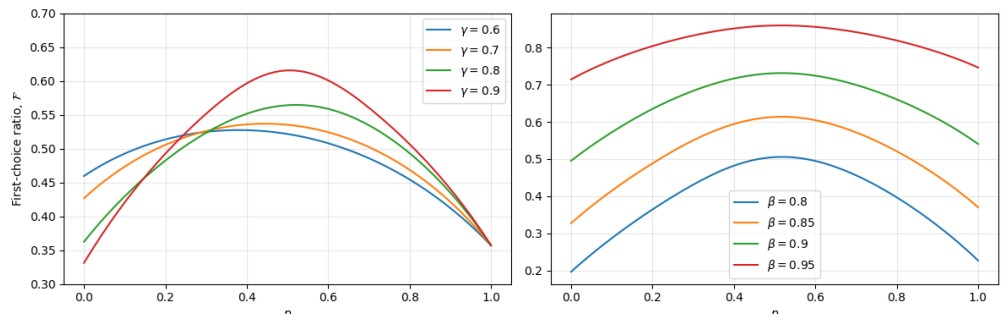

Figure 15: Variation of first choice ratio $\mathcal{F}$ with $p$ for varying values of $\beta$ and $\gamma$. The left panel uses $c = 0.2$ and $\beta = 0.9$. The right panel uses $\gamma = 0.8$ and $c = 0.3$.

