# OpenReview forum: "Matchings Under Biased and Correlated Evaluations"
_NeurIPS.cc/2025/Conference — NeurIPS 2025 poster_

### Official Review · Reviewer_H4Bb · 2025-06-18

**Clarity:** 3
**Significance:** 3
**Originality:** 4
**Rating:** 5
**Confidence:** 4

**Summary:**

The authors develop a large‐market stable‐matching model with two evaluating institutions whose signals are both group-biased (disadvantaged candidates’ scores are down-scaled by a factor $\beta\le 1$) and imperfectly aligned (correlation parameter $\gamma\in[0,1]$). They derive closed-form formulas for the pair of admission cut-offs that uniquely characterize every stable matching, reveal three critical $\gamma$ thresholds that partition the parameter space into four analytic “phases,” and obtain a piecewise expression for the disadvantaged group’s representation ratio $R(\beta,\gamma)$. Although each institution’s threshold can behave non-monotonically, $R(\beta,\gamma)$ is proven to increase with either less bias (higher $\beta$) or tighter signal alignment (higher $\gamma$). A normalized version of $R$ cleanly separates these two effects, enabling the authors to trace Pareto frontiers of bias-reduction versus alignment-improvement interventions that achieve any target fairness level—thus offering actionable guidance on when debiasing, standardization, or coordination yields the greatest gains in representation.

**Questions:**

1. Would your selection procedure change fundamentally under additive bias, or could your proofs adapt directly?

2. You state: “In the finite setting, stable matchings are characterised by threshold-based rules.” Could you cite the specific sources supporting this statement?

3. The main results in the article focus on the closed‐form solutions across different parameter phases. In practice, when $n$ is not large, how can these theoretical treshholds  guide the design of effective intervention methods?

4. How sensitive are the $\gamma$ thresholds and the monotonicity of $\mathcal{R}$ to non-uniform or correlated talent distributions?

**Ethical Concerns:**

["NO or VERY MINOR ethics concerns only"]

**Final Justification:**

**Final Justification for Recommended Score**

After considering the rebuttal, discussions with the authors, and exchanges with other reviewers and the AC, my evaluation is as follows:

**Resolved Issues**

* **Clarity of Section 3:** Authors agreed to move detailed derivations and technical case analyses to the appendix while keeping key structural insights in the main text, improving readability.
* **i.i.d. Assumption and Monotonicity:** Authors provided a rigorous explanation of why monotonicity results rely on the i.i.d. assumption, clarified the role of the continuum limit, and acknowledged limitations under strong dependence. They will add a discussion of possible relaxations and extensions.
* **Presentation of Theorems:** Authors committed to simplifying theorem statements in the main text to highlight intervention-relevant properties, retaining full proofs in the appendix.
* **Special-case Clarification:** Authors clarified the case of full admission and its implications for the representation ratio.


**Conclusion**
Given the strength of the core results, the clear motivation, the responsiveness to feedback, and the improvements promised for the final version, I increase my original score.

**Limitations:**

Yes

**Quality:**

4

**Strengths And Weaknesses:**

## Strengths

* The paper presents a complete large-market analysis, deriving closed-form formulas for both admission cut-offs *and* the resulting representation metrics; proofs are laid out carefully in the main text and extensive appendices, with clear intermediate lemmas (e.g., Theorem 3.1–3.3 and Table 1’s structural summary) that make the logical flow easy to audit .

* By turning the representation ratio into an explicit function $R(\beta,\gamma)$ and normalised variant $N(\beta,\gamma)$, the authors translate their math into concrete design tools (e.g., the Pareto-frontier plots in Fig. 3) that a practitioner could use to pick a cost-effective debiasing or standardisation strategy .

* Prior studies either assume common rankings with bias or study correlation without group-specific bias (Brilliantova & Hosseini, 2022). This paper is, to my knowledge, the first to analyse the *interaction* and to prove monotonicity of a fairness metric despite non-monotone intermediate thresholds .

* The writing is well-structured: Section 2 walks through the model step-by-step, and “Step 1 / Step 2” derivations of $s_1^\star$ and $s_2^\star$ (p. 6-7) are particularly reader-friendly .

* Summaries and visual aids (Table 1, Figures 2–3) make the complex piece-wise behaviour intuitively accessible .

## Weaknesses

* Uniform score distributions, two institutions of equal capacity, and a single prestige-dominant preference profile (all candidates prefer institution 1) abstract away many features of real markets. The authors acknowledge some of these limitations themselves , but they remain a barrier to external validity.

* All results live in an asymptotic continuum. Even a small simulation study or a synthetic-to-real case (as done by Celis et al., 2024 for the one-institution model) would have strengthened the quality argument.

* The four-case formula for $s^\star(\gamma)$ and the ensuing phases are relegated to dense algebra; newcomers to matching theory may struggle to interpret why the geometric regimes matter.

* Some notation choices (e.g., switching between raw, biased, and scaled utilities) require multiple passes to parse. A running toy example early on could improve approachability.

* Real clearinghouses (college admissions, labour platforms) routinely involve dozens of institutions and many protected attributes, so scaling of results is unclear.

* In the main article the probability of each candidate prefers Institution 1 over Institution 2 is equal to 1 and the general case put on the appendic. In my opinion as far as possible it is better put main and general result in the main article not in appendix. We have the same situation fo for consider the general case of $\beta$.

* In the main article the Institution 1 selects only from group $G_1$ but it claims general case is consider in the appendix.

* The definition of fairness criteria is defined only the fraction of selected candidates from group $G_j$. Since the main point of your model is preference of institution 1 to 2. Why this point is not consider in the definition of fairness criteria

* As my understand to run intervention planning we need to estimate $(\beta_0,\gamma_0)$, but in this work do not explain how can estimate it by emperical data.

* Without empirical or simulated finite-market evidence, practitioners can’t judge how sharp the analytic thresholds are for real-sized cohorts.


The paper is mathematically rigorous, clearly written, and addresses a problem of growing practical importance. In my view, it has substantial impact and surpasses several previously published studies on the topic. Its main drawback, however, is the overall structure. Owing to conference page limits, the manuscript exceeds 60 pages and relegates many key discussions and facts to the appendix, which diminishes the work’s immediate value. Although I am not a specialist in this field, I believe that submitting a journal version—where the principal results are included in the main text rather than the appendix—would give the paper a strong chance of acceptance and make it better suited for a journal than for a conference.

---

> ### Author Rebuttal · Authors · 2025-07-30
>
> Thank you for your thoughtful and detailed review. While our model builds on earlier formulations, we believe the structural classification into correlation regimes, the closed-form computation of thresholds, and the proof techniques for monotonicity provide nontrivial analytical advances that are central to enabling the intervention strategies in Section 4.
>
> Below, we respond to your comments and questions.
>
> >...  put main and general result in the main article not in appendix. … the general case of $\beta$.
>
> Thank you for the suggestion. We agree that the $p < 1$ and $\beta < 1 - c$ cases merit more visibility. As the analysis in both settings is more intricate than in the $p = 1$ and $\beta \geq 1 - c$ case, it currently appears in the appendix. We will include a concise summary in the main body to highlight the broader applicability of our framework.
>
> > In the main article the Institution 1 selects only from group $G_1$...
>
> Thank you for the comment. We believe there may be a misunderstanding: in the main body, Institution 1 does select candidates from both groups whenever $\beta\geq 1 - c$, which is the regime analyzed in detail. Line 65 mentions that candidates from $G_2$ are "not automatically excluded" refers to the case $\beta < 1 - c$, where no $G_2$ candidate can meet the threshold. We will revise this line for clarity.
>
> > The definition of fairness criteria is defined only the fraction of selected candidates from group $G_j$. Since the main point of your model is preference of institution 1 to 2. Why this point is not consider in the definition of fairness criteria.
>
> Thank you for this suggestion. In the $p = 1$ case (main body), where all candidates prefer Institution 1, group-level selection rate remains a meaningful fairness metric. However, your point is well-taken for the general $p$ case (in the Appendix), where preferences vary. We will expand the discussion of fairness notions for $p < 1$ in the final version.
>
>
> > … estimate $\beta_0,\gamma_0$ …
>
> Thanks for the question. While we treat $\beta$ and $\gamma$ as system-level parameters, they can be estimated from data. If Institution 1 uses threshold $ŝ_1^\star$ for Group 1, we can invert the formula $s_1^\star = \frac{2 - c}{1 + 1/\beta}$ to estimate: $\beta_0 \approx \frac{ŝ_1^\star}{2 - c - ŝ_1^\star}$.
>
> Similarly, if the empirical representation ratio $\hat{\mathcal{R}}$ and Institution 2’s threshold $ŝ_2^\star$ are known, one can numerically invert $\mathcal{R}(\beta_0, \gamma)$ to estimate $\gamma_0$. We will add a brief discussion of this in the final version.
>
>
> > 1. Would your selection procedure change fundamentally under additive bias…
>
> Thank you for this question. Our model and analysis extend naturally to settings involving additive bias. For instance, suppose the observed utility $û_{ij}$ of a candidate $i \in G_2$ for institution $j$ is given by $u_{ij} - \beta$, where $u_{ij}$ is the true utility and $\beta \in [0,1]$ represents the additive bias penalty.
>
> Under a natural assumption $\beta \leq c$ (analogous to the multiplicative case where we assume $\beta \geq 1 - c$) which ensures that Institution 1 admits candidates from group $G_2$, the threshold for Institution 1 becomes:
>
> $$ s_1^\star = 1 - \frac{c + \beta}{2}.$$
>
> The existence of $\gamma$-thresholds continues to hold: we show an analogue of Theorem 3.1:
>
> **Theorem.** *Fix $\beta < c$. Then there exist thresholds $\gamma_1, ..., \gamma_4 \in [0,1]$ such that:*
>
> 1. $s_2^\star + \beta \leq 1 - \gamma$ iff $\gamma \leq \gamma_1$
> 2. $s_2^\star \leq 1 - \gamma$ iff $\gamma \leq \gamma_2$
> 3. $s_2^\star \geq s_1^\star \gamma$ iff $\gamma \leq \gamma_3$
> 4. $s_2^\star + \beta \geq (s_1^\star + \beta)\gamma$ iff $\gamma \leq \gamma_4$.
>
> Using these threshold conditions, along with the probability expressions developed in Section 3 for $\text{Pr}[\gamma v_{i1} + (1 - \gamma) v_{i2} \geq s_2 \,\wedge\, v_{i1} < s_1],$ we can establish an analog of Theorem 3.2 to compute $s_2^\star$ under additive bias. For example, in the regime $\gamma < \gamma_1$, the threshold equation becomes:
>
> $$ s_1^\star - \frac{2s_1^\star s_2^\star - \gamma (s_1^\star)^2}{2(1 - \gamma)} + (s_1^\star + \beta) - \frac{2(s_1^\star + \beta)(s_2^\star + \beta) - \gamma (s_1^\star + \beta)^2}{2(1 - \gamma)} = c. $$
>
> We will include a brief discussion of this generalization in the final version of the paper.
>
> > 2. You state: "In the finite setting, stable matchings are characterised by threshold-based rules." Could you cite the specific sources supporting this statement?
>
> Thank you for pointing this out. This result appears in Appendix C (Prop. C.1). We provide a brief sketch of the argument here and will add a short summary to the main text in the final version.
>
> Let $M$ be any stable assignment, and let $m_1$ and $m_2$ denote the minimum observed utility among candidates assigned to the two institutions respectively. Then, the stability of the assignment implies that Institution 1 must select all candidates $i$ with $û_{i1} \geq m_1$. A similar argument applies for Institution 2. Thus, the assignment $M$ is fully characterized by the thresholds $s_1^* = m_1$ and $s_2^* = m_2$.
>
> > 3. ... In practice, when $n$ is not large, how can these theoretical thresholds guide the design of effective intervention methods?
>
> Thank you for this question. We agree that understanding how theoretical thresholds translate into practical guidance is essential. To address this, we build on a result by Azevedo and Leshno [7] which shows that the thresholds governing stable assignments converge to deterministic values $s_1^\star$ and $s_2^\star$ defined by a continuum limit. We can quantify this convergence by showing that deviations are at most $O(\sqrt{\log n / n})$ with high probability. Since $\mathcal{R}$ is a smooth function of the thresholds, it inherits their concentration properties—justifying the use of continuum analysis. (See our response to reviewer #4qyh for full details.)
>
> To corroborate this theoretical result and illustrate its practical utility, we implemented our model for finite values of $n$ and ran 30 independent trials over randomly generated candidate profiles for each value of $\gamma$. We report results for $n = 100$ below. The empirical means closely match the continuum predictions. For $n = 250$, we observe a further reduction in standard deviation, indicating the stability and robustness of our theoretical thresholds.
>
> ---
> **$n = 100$, $c = 0.2$, $\beta = 0.9$, $|G_1| = |G_2| = 100$**
> | $\gamma$ | Representation Ratio (mean ± std) |
> | :---: | :----------------------------------: |
> |  0.0  |          0.48 ± 0.16           |
> |  0.1  |          0.44 ± 0.14           |
> |  0.2  |          0.44 ± 0.15           |
> |  0.3  |          0.49 ± 0.16           |
> |  0.4  |          0.53 ± 0.25           |
> |  0.5  |          0.55 ± 0.19           |
> |  0.6  |          0.61 ± 0.19           |
> |  0.7  |          0.64 ± 0.28           |
> |  0.8  |          0.64 ± 0.16           |
> |  0.9  |          0.62 ± 0.17           |
> |  1.0  |          0.73 ± 0.42           |
>
> ---
> These results demonstrate that even in moderately sized systems, the continuum thresholds closely approximate the empirical representation ratios. This suggests that our theoretical framework can meaningfully inform policy decisions in realistic settings. We believe this also responds to the reviewer’s suggestion for a simulation-based validation, similar in spirit to the empirical component of Celis et al. (2024). We will include detailed plots and empirical results in the final version.
>
> > 4. How sensitive are the $\gamma$-thresholds and the monotonicity of $\mathcal{R}$ to non-uniform or correlated talent distributions?
>
> Thank you for this important question. While our main analysis assumes i.i.d. uniform attributes for analytical tractability, our framework extends naturally to more general settings—including non-uniform distributions and correlated attributes.
>
> **Non-uniform distributions:** For independent attributes drawn from a smooth distribution with CDF $\Phi$, the equation for the threshold $s_1^\star$ becomes:
>
> $$
> \Phi(s_1^\star) + \Phi\left( \frac{s_1^\star}{\beta} \right) = 2(1 - c_1).
> $$
>
> The computation of representation probabilities also adapts using $\Phi$ and its density $\phi$. For example, in Case I when $\gamma \leq \gamma_1$:
>
> $$
> \Pr[u_{i1} < s_1, u_{i2} \geq s_2] = \Phi(s_1) - \int_{0}^{s_2/\gamma} \Phi\left(\frac{s_2 - \gamma x}{1 - \gamma}\right) \phi(x) \, dx.
> $$
>
> While closed-form expressions may not always be available, the thresholds and representation metrics remain numerically tractable, and the threshold regime structure (Theorem 3.1) continues to hold. We did simulations using truncated Gaussians on $[0,1]$ to confirm the monotonicity of $\mathcal{R}(\beta, \gamma)$ across a wide range of variances.
>
> **Correlated attributes:** When $v_{i1}$ and $v_{i2}$ are correlated, the sensitivity of $\mathcal{R}$ to $\gamma$ depends on the strength of the correlation. In the extreme case where $v_{i2} = \lambda v_{i1}$, the utility $u_{i2}$ becomes a scaled version of $v_{i1}$, rendering the representation ratio invariant to $\gamma$.
>
> More generally, correlated attributes can often be expressed as functions of independent variables (e.g., via whitening in the Gaussian case). The observed utilities $û_{ij}$ then inherit this structure, allowing our framework to remain applicable. However, establishing monotonicity of $\mathcal{R}$ analytically in such settings remains challenging and is a promising direction for future work. We will include a discussion of these generalizations in the final version.

---

> > ### Comment · Reviewer_H4Bb · 2025-08-01
> >
> > Thank you for the thorough and thoughtful rebuttal. Your clarifications address my concerns. I am satisfied with the revisions you plan to incorporate, and I am increasing my overall score.

---

> > > ### Author Response · Authors · 2025-08-07
> > >
> > > Thank you again for your thoughtful engagement and kind words. We’re glad the clarifications addressed your concerns, and we truly appreciate your note about increasing the score. Your comments have helped us improve the paper, and we will incorporate all the discussed points in the final version. We just wanted to gently mention that the system still reflects the earlier score on our end.

---

### Official Review · Reviewer_4qyh · 2025-06-29

**Clarity:** 3
**Significance:** 4
**Originality:** 4
**Rating:** 5
**Confidence:** 4

**Summary:**

In this work, authors study how institutional bias and inter-institutional correlations in evaluation affect the ratio of advantaged and disadvantaged populations under matching of candidates and institutions. Here, disadvantaged means that the quality may be underestimated multiplicatively by a bias factor. The main claim is that both reducing bias and increasing correlation between evaluations of institutions can improve the representation ratio of disadvantaged groups. The authors support this claim with detailed analytical results of how the bias and correlation parameters affect the representation ratio at equilibrium. Finally, their framework can extend to many natural generalizations which makes this framework nice for evaluating decentralized selection decisions on the demographics of selected candidates.

**Questions:**

- Is it possible to simplify the theorem statements in the main text (and keep full statements in the appendix) to more clearly show that $s_2^*$ and $\mathcal{R}$ have the properties necessary for intervention as explored in Section 4?
- (Extension question) How might $\mathcal{R}$ change over time if this model is extended over time, especially if previous admission decisions influence candidate preferences in future time steps?
- Small question, but what happens if $c= \frac{1}{2}$, i.e., is there a special case when both institutions can admit everyone?

**Ethical Concerns:**

["NO or VERY MINOR ethics concerns only"]

**Final Justification:**

My confidence has increased since the authors addressed my questions of model assumptions and extensions in detail and I have become more convinced of the strength of the results they present in this work.

**Limitations:**

Yes.

**Paper Formatting Concerns:**

None.

**Quality:**

4

**Strengths And Weaknesses:**

$\textit{Strengths}$: The authors present a well-motivated problem and well-motivated model parameters that correspond to key intervention points of evaluating decentralized admission procedures. Additionally, their model is very clearly explained, assumptions are clearly stated, and the authors highlight possible generalizations whenever they state simplifying assumptions (such as uniform candidate preference for one institution over another and sufficiently unbiased institutions). The organization of Section 3 and the supporting figure and table create nice intuition which makes it easier to follow along and understand their main conclusions, even though their analysis admits somewhat messy closed form solutions. Finally, Section 4 has a nice discussion of example interventions for both $\beta$ and $\gamma$ and the numerical experiments support the extent to which the interventions can improve $\mathcal{R}$ (and $\mathcal{N}$).

$\textit{Weaknesses}$: The complex analysis of the equilibrium strategy of the second institution reduces the clarity of section 3 and some details of their analysis could be deferred to the appendix for the sake of clarity. Additionally, I was interested in more discussion on the extent to which the monotonicity of $\mathcal{R}$ with respect to $\gamma$ depends on the assumption that attribute 1 is drawn iid for each candidate. That is, should we expect correlation of evaluation strategy to only improve representation if the institutions are correlating on an attribute where all candidates have the same distribution over values and each are independently assigned a value?

---

> ### Author Rebuttal · Authors · 2025-07-30
>
> Thank you for your thoughtful and constructive comments. We appreciate your careful reading of the paper and the insightful questions raised. Below, we respond to each point in detail, clarifying our assumptions, highlighting key contributions, and discussing extensions and open directions suggested by your feedback.
>
> > The complex analysis of the equilibrium strategy of the second institution reduces the clarity of section 3 and some details of their analysis could be deferred to the appendix for the sake of clarity.
>
> Thank you for this helpful suggestion. We agree that the derivation of the equilibrium strategy for the second institution is a technically involved part of Section 3 and may affect the overall readability. In the final version, we will move some of the more intricate case analyses and algebraic derivations to the appendix, while preserving the key insights and structural results (e.g., unimodality and threshold characterization) in the main text. This should help streamline the exposition without sacrificing the conceptual contributions.
>
> > Additionally, I was interested in more discussion on the extent to which the monotonicity of $\mathcal{R}$
>  with respect to $\gamma$ depends on the assumption that attribute 1 is drawn iid for each candidate. That is, should we expect correlation of evaluation strategy to only improve representation if the institutions are correlating on an attribute where all candidates have the same distribution over values and each are independently assigned a value?
>
> We thank the reviewer for raising this insightful question about the role of the i.i.d. assumption in our analysis of the monotonicity of the representation ratio $\mathcal{R}(\beta, \gamma)$ with respect to the correlation parameter $\gamma$.
>
> To rigorously establish monotonicity, our analysis first reduces the finite-population model to a continuum limit. This reduction leverages the result of Azevedo and Leshno [7], which shows that as the number of candidates $n$ grows, the outcome of the stochastic finite model concentrates around a deterministic infinite-population limit.
>
> To be more precise, in the finite model, each candidate $i$ in group $G_j$ has latent attributes $v_{i1}, v_{i2}$, drawn independently from a distribution over $[0,1]^2$ (uniform in the baseline case). These attributes are transformed into observed or *biased* utilities $û_{ij}$, which determine candidate preferences and institutional choices. A **realization** $\omega$ corresponds to a specific draw of all these $(v_{i1}, v_{i2})$ values for each candidate in the population. Once $\omega$ is fixed, the utilities $û_{ij}$ are also fixed, and the stable matching is deterministic.
>
> In this setting, **Proposition C.1** (Appendix) establishes that for every realization $\omega$, the stable assignment is governed by thresholds $S_1^\star(\omega)$ and $S_2^\star(\omega)$, such that Institution 1 selects candidates with $û_{i1} \geq S_1^\star(\omega)$ and Institution 2 selects those with $û_{i2} \geq S_2^\star(\omega)$.
>
> Now, under the assumption that candidate attributes are i.i.d., the fraction of candidates in each group satisfying these threshold conditions concentrates around their expected values due to standard probabilistic concentration (e.g., Chernoff bounds or the central limit theorem). This implies that the empirical thresholds $S_1^\star(\omega)$ and $S_2^\star(\omega)$ converge to deterministic values $s_1^\star$ and $s_2^\star$ as $n \to \infty$, which satisfy explicit equations derived in the infinite-population model.
>
> We restate the formal convergence result below:
>
> **Proposition.** *Given parameters $c$, $\beta$, and $\gamma$, there exist unique values $s_1^\star$ and $s_2^\star$ such that the following holds. Consider a finite stochastic instance $\mathcal{I}$ with $|G_1| = |G_2| = n$, and two institutions with capacities $cn$ each, bias parameter $\beta$, and correlation parameter $\gamma$. Then with high probability, there exist thresholds $s_1, s_2$ such that the stable assignment in $\mathcal{I}$ is governed by these thresholds and satisfies $|s_\ell - s_\ell^\star| = O\left(\sqrt{\frac{\log n}{n}}\right)$ for each $\ell \in \{1, 2\}$.*
>
> This proposition justifies the use of the continuum model to analyze properties such as the monotonicity of $\mathcal{R}$ with respect to $\gamma$, as the finite model increasingly resembles the limit as $n$ grows.
>
> That said, if candidate attributes exhibit strong dependence—for example, if all candidates have identical or highly similar attributes—the law of large numbers and related concentration results no longer hold. In such cases, the convergence to a deterministic limit may fail, and our monotonicity results may not apply.
>
> Nonetheless, we believe the qualitative trend—that increasing $\gamma$ improves $\mathcal{R}$—likely persists under weaker assumptions, such as mild correlations or clustered structures among candidate attributes. Extending the convergence analysis to such settings remains an open and valuable direction for future research.
>
> We will add a discussion of this modeling assumption and possible extensions in the final version. We thank the reviewer again for encouraging us to clarify this aspect.
>
> > Is it possible to simplify the theorem statements in the main text (and keep full statements in the appendix) to more clearly show that and have the properties necessary for intervention as explored in Section 4?
>
> Thank you for the helpful suggestion. We agree that clarifying the key properties of the main results could improve the flow into the analysis in Section 4. In the final version, we plan to simplify the presentation of the main theorems—focusing on the properties most relevant for intervention design—while retaining full technical statements in the appendix. This should make the analysis section more accessible without requiring substantial changes to the structure.
>
>
> > (Extension question) How might $\mathcal{R}$ change over time if this model is extended over time, especially if previous admission decisions influence candidate preferences in future time steps?
>
> Thank you for this thought-provoking question. Our current model is static and analyzes a single-round matching process. That said, we agree that in settings where candidate preferences, group participation, or institutional strategies evolve over time—based on past match outcomes—the representation ratio $\mathcal{R}$ could change in nontrivial ways.
>
> Capturing such dynamics would likely require a game-theoretic or behavioral framework with endogenous feedback. For instance, if increased representation improves perceived fairness or institutional reputation, this may in turn affect future applications—potentially creating a virtuous cycle. Conversely, persistent underrepresentation could lead to disengagement and long-run disparities. Modeling such feedback loops could illuminate when correlation amplifies or corrects inequities over time.
>
> While these dynamics are outside the scope of our current analysis, we believe this is a compelling direction for future work—especially in persistent systems like recurring admissions or hiring cycles. We will mention this direction in the final version as a promising avenue for future work.
>
>
> > Small question, but what happens if $c=1/2$, i.e., is there a special case when both institutions can admit everyone?
>
> Thank you for the question. In our model, there are two groups of size $n$, so the total candidate pool has $2n$ individuals. Each institution admits a fraction $c$ of the total population, i.e., $cn$ candidates per institution. So full admission—where every candidate is matched—occurs when $2cn = 2n$, which implies $c = 1$.
>
> Therefore, the case $c = 1/2$ is still selective: only half the total population (i.e., $n$ candidates) is admitted across both institutions. In the regime $\beta \geq 1-c = 1/2$, we can compute the values of the $\gamma$-thresholds using Theorem 3.1, the thresholds $s_2^*$ using Theorem 3.2 and the representation ratio using Theorem 3.3.
>
> In the case $c = 1$, all the candidates are admitted and hence, the representation ratio is $1$. We can compute the values of the thresholds $s^*_1$ and $s_2^\star$ using equations (4) and Theorem 3.2 respectively (observe that we shall always be in the regime $\beta \geq 1-c$ here).
>
> If we have misunderstood your intent or if you're referring to a different version of the model (e.g., with institutions admitting their own groups), we would be happy to clarify further.

---

> > ### Comment · Reviewer_4qyh · 2025-08-04
> >
> > I appreciate the authors' thorough response and engagement with my questions. Their stated revisions have addressed my concerns and I appreciate the promising directions for future work. Therefore, I will maintain my score and my confidence has also increased.

---

### Official Review · Reviewer_hvVj · 2025-07-03

**Clarity:** 4
**Significance:** 3
**Originality:** 3
**Rating:** 5
**Confidence:** 3

**Summary:**

The authors study a matching problem where two sets of candidates apply to two institutions. One set of candidates is advantaged, and the other is disadvantaged. Both candidates prefer one institution over the other. Their priorities according to the institutions are randomly drawn but the priority of a candidate at one institution is correlated to their priority at the other. The disadvantaged candidates have their priorities further scaled down by a bias factor. The authors study the behavior of the representation of these students at the institutions as a function of the bias factor and the correlation factor between the priorities at different institutes. They fully characterize this behavior in a simple setting where the number of candidates is large (tending to infinity). They show that in a particular regime, the representation of the disadvantaged students increase with decreasing bias and increasing correlation.

**Questions:**

I was a bit surprised that the representation ratio was positively correlated with the correlation factor gamma, given that the tone of the introduction seemed to be making a case against positive correlation. I don't know if this was intentional or not. In hindsight, this positive correlation makes sense since the two dimensions of the utilities are independent and the bias factor applies equally to both of them.

Would this correlation flip if the \beta for u_2 was slightly higher (ie less biased) than that for u_1?

**Ethical Concerns:**

["NO or VERY MINOR ethics concerns only"]

**Final Justification:**

The authors addressed my comment on the novelty of their technical contribution

**Limitations:**

yes

**Quality:**

4

**Strengths And Weaknesses:**

The paper is very well-written and studies an interesting problem. Their problem, even though it uses a simple setting, seems to require a tedious analysis but they do a good job of highlighting the more interesting parts. Their interpretation and discussion of their analytical results is also very thorough and they also discuss useful simplified examples where appropriate.

My only complaint would be that the techniques don't seem very novel or if they are, they haven't highlighted it in the main body of the paper. But that is minor and is to be expected with these kind of problems.

---

> ### Author Rebuttal · Authors · 2025-07-30
>
> We thank you for your thoughtful comments and encouraging overall assessment. We're especially glad that the paper’s analysis and framing were found clear and of high quality. Below, we respond to your questions and comments.
>
> > My only complaint would be that the techniques don't seem very novel or if they are, they haven't highlighted it in the main body of the paper. But that is minor and is to be expected with these kind of problems.
>
> We appreciate the reviewer’s comment regarding the novelty of techniques and agree that clarifying this in the main body would strengthen the paper. While our model builds on established components in matching theory and bias modeling, we believe the technical contributions are substantial and novel in several directions:
>
> 1. **Regime reduction via $\gamma$-thresholds.** We introduce a new classification of parameter regimes that collapses 16 case distinctions into 3 interpretable ones using carefully identified $\gamma$-thresholds. This structure emerges from non-trivial comparisons between institution-specific thresholds and cross-group rank orderings, and allows us to express key outcome metrics in unified forms across the regimes.
>
> 2. **Characterization and unimodality of $s_2^\star$.** We provide a closed-form description of the second institution’s optimal threshold $s_2^\star$ in terms of $\beta$, $\gamma$, and $c$, and rigorously prove that $s_2^\star$ behaves in a unimodal fashion with respect to $\gamma$. This behavior reflects a non-obvious trade-off: while higher correlation can align evaluations and boost the expected quality of selected candidates, it can also lead to reduced access to high-scoring candidates due to competition with Institution 1. Capturing this crossover requires careful comparative statics and monotonicity arguments.
>
> 3. **Implicit monotonicity of $\mathcal{R}(\gamma)$.** Although $\mathcal{R}$ can be written in terms of the parameters $\beta$ and $\gamma$, its piecewise definition and dependence on regime-specific thresholds make direct analysis challenging. We establish its monotonicity not by direct computation, but by examining the properties of the equilibrium-defining equations that $\mathcal{R}$ satisfies in each regime. This strategy leverages structure in the stable matching mechanism and can generalize to other fairness metrics where explicit formulas are difficult to analyze.
>
> We will revise the final version to highlight these contributions more clearly in the main text.
>
>
> > I was a bit surprised that the representation ratio was positively correlated with the correlation factor gamma, given that the tone of the introduction seemed to be making a case against positive correlation. I don't know if this was intentional or not. In hindsight, this positive correlation makes sense since the two dimensions of the utilities are independent and the bias factor applies equally to both of them. Would this correlation flip if the \beta for u_2 was slightly higher (ie less biased) than that for u_1
>
>
> Thank you for this thoughtful observation. We understand the reviewer’s surprise and agree that the positive correlation between the representation ratio $\mathcal{R}$ and the correlation parameter $\gamma$ might initially appear to contradict the tone of the introduction. In the introduction, our concern with correlation stems from its potential to entrench shared biases across institutions. However, as our results show, the effect of increasing $\gamma$ on $\mathcal{R}$ is more nuanced and depends critically on the bias parameter $\beta$.
>
> In particular, we prove that $\mathcal{R}$ is monotonically increasing in $\gamma$ **only** when $\beta \geq 1 - c$. Below this regime, monotonicity can fail, and the representation ratio may behave non-monotonically; see Table 2 in Section I for a comparison of these cases. We agree that this distinction could be made more explicit in the paper and appreciate the opportunity to clarify it.
>
> Regarding asymmetric bias across institutions (i.e., $\beta_1 < \beta_2$), our findings suggest that increasing $\gamma$ can still improve representation, because higher correlation allows the less-biased institution to better “rescue” disadvantaged candidates who narrowly miss the threshold at the more-biased institution. This effect persists even when the bias levels differ.
>
> To illustrate, consider the following extreme case:
>
> **Propsition.** *Let Institution 1 apply bias with parameter $\beta_1$, where $1 - c < \beta_1 < 1$, and let Institution 2 be unbiased, i.e., $\beta_2 = 1$. Then:*
> - *When $\gamma = 1$, $\mathcal{R}(\gamma) = 1$,*
> - *When $\gamma = 0$, $\mathcal{R}(\gamma) = \frac{2c + \beta_1 - 1}{2c\beta_1 - \beta_1 + 1} < 1$.*
>
> **Proof idea.** When $\gamma = 1$, the institutions are fully aligned, and any candidate with $u_{i1} \geq s_2^\star$ is selected by at least one institution, regardless of group. In this case, the threshold $s_2^\star$ becomes $1 - c$, and each group contributes a fraction $c$ to the total admits—yielding a representation ratio of 1 despite the asymmetry in bias. In contrast, when $\gamma = 0$, the first institution selects candidates based on a threshold $s_1^\star = \frac{2-c}{1+1/\beta_1}$. For the  remaining candidates, the distribution of $v_{i2}$ is i.i.d. and hence, the fraction of candidates from each group that get selected to Institution 2 depends on their group-dependent acceptance probabilities.
>
> **Additional simulations.** To further support this point, we have extended our implementation to allow different bias parameters $\beta_1$ and $\beta_2$ for Institutions 1 and 2, respectively. Our numerical experiments indicate that the monotonicity of $\mathcal{R}(\beta_1, \beta_2, \gamma)$ with respect to $\gamma$ continues to hold across a range of asymmetric settings. For instance, we have verified this property for $c = 0.3$, $\beta_1 = 0.8$, and $\beta_2$ varying from $0.8$ to $1$ (where $\beta_1$ and $\beta_2$ are the bias parameters for the two institutions, respectively). Similar trends hold for other values of $c$ and for $\beta_1 \geq 1 - c$.
>
> Due to constraints of the rebuttal process, we are unable to attach plots, but we will include this discussion—along with illustrations—in the final version.

---

> > ### Comment · Reviewer_hvVj · 2025-08-07
> >
> > Thank you for the response. I will update my score on the originality.

---

### Decision · Program_Chairs · 2025-09-17

**Decision:**

Accept (poster)

**Comment:**

This paper studies stable matching with group-dependent bias and correlated evaluations across institutions, deriving closed-form expressions for representation ratios and characterizing equilibrium thresholds that enable fairness-aware interventions in decentralized selection systems. All reviewers provided positive assessments regarding the paper's technical rigor, clarity, and practical relevance for fairness-aware system design. Reviewers appreciated the well-motivated problem formulation, thorough analytical results, and the framework's potential for real-world interventions. The authors provided comprehensive rebuttals addressing reviewer concerns and commitment to revisions including making improvements to Section 3 clarity, theorem presentation, discussion of model assumptions and extensions, etc. This paper merits acceptance and authors are encouraged to incorporate all the reviewer suggestions in the final version.